# Efficient $k$-Sparse Band–Limited Interpolation with Improved Approximation Ratio

**Yang Cao**
Wyoming Seminary
ycao4@wyomingseminary.org

**Xiaoyu Li**
University of New South Wales
7.xiaoyu.li@gmail.com

**Zhao Song**
University of California, Berkeley
magic.linuxkde@gmail.com

**Chiwun Yang**
Sun Yat-sen University
christiannyang37@gmail.com

## Abstract

We consider the task of interpolating a $k$-sparse band–limited signal from a small collection of noisy time-domain samples. Exploiting a new analytic framework for hierarchical frequency decomposition that performs systematic noise cancellation, we give the first polynomial-time algorithm with a provable $(3 + \sqrt{2} + \varepsilon)$-approximation guarantee for continuous interpolation. Our method breaks the long-standing $C > 100$ barrier set by the best previous algorithms, sharply reducing the gap to optimal recovery and establishing a new state of the art for high-accuracy band–limited interpolation. We also give a refined "shrinking-range" variant that achieves a $(\sqrt{2} + \varepsilon + c)$-approximation on any sub-interval $(1 - c)T$ for some $c \in (0, 1)$, which gives even higher interpolation accuracy.

## 1 Introduction

The fast Fourier transform (FFT) (Cooley and Tukey, 1965) stands as a cornerstone across engineering, signal processing, mathematics, and theoretical computer science, underpinning both theoretical advances and practical applications. Over time, numerous FFT variants have been proposed, tailored to different signal domains and time-invariance assumptions (Oppenheim et al., 1997; Osgood, 2002; Oppenheim, 2011). In this work we focus on the *sparse Fourier transform* (SFT), which assumes the signal is band-limited (or Fourier-sparse), i.e., it is observed in the time domain (either discrete or continuous) but is $k$-*sparse* in the frequency domain, i.e., its spectrum $\widehat{x}$ contains only $k$ non-zero components. Our main results concern *one-dimensional continuous* signals, though we discuss extensions to higher dimensions and the discrete setting. Formally, the band-limited signal is defined as follows.

**Definition 1.1** (Band-limited signal). *Let $k \in \mathbb{Z}_{>0}$. Let $\delta_{f_i}(f)$ denote the Dirac function centered at $f_i \in \mathbb{R}$. We define the $k$-sparse band-limited signal $\widehat{x}^*(f)$ to be as follows:*

$$x^*(t) = \sum_{j=1}^{k} v_j \cdot e^{2\pi \mathbf{i} f_j t} \xrightarrow{\text{CFT}} \widehat{x}^*(f) = \sum_{j=1}^{k} v_j \cdot \delta_{f_j}(f)$$

*where $v_j \in \mathbb{C}$ is the coefficient and $f_j \in \mathcal{F}$ is the frequency contained in the frequency range $\mathcal{F} \subset \mathbb{R}$ for each $j \in [k]$. We use $\mathcal{K}$ to denote the set of $f_j$'s.*

Band-limited signals are ubiquitous in practice, underpinning tasks such as image compression and analysis (Watson, 1994), compressed sensing (Donoho, 2006), and (deep) learning with frequency-invariant kernels (Mei et al., 2021). Fourier methods have also emerged as a powerful tool in machine

learning, inspiring diverse models such as random features for kernel approximation (Rahimi and Recht, 2007), Fourier neural operators for parametric PDEs (Li et al., 2021b), and spectral methods for temporal domain generalization (Yu et al., 2025). Efficient algorithms for computing sparse Fourier transforms are thus of fundamental importance to both signal processing and modern machine learning pipelines. A canonical challenge in this context is *band-limited signal interpolation* (Chen et al., 2016)—closely related to the[1] *Fourier Set-Query* problem (Price, 2011)—which seeks to reconstruct (part of) a signal from only a handful of *noisy* samples (observations $x^*(t_i) + g(t_i)$ at chosen time points $t_i \in [0, T]$), ideally on the order of $k$, taken from the time domain $[0, T]^d$. The band-limit $F$ constrains all frequencies to lie in $[-F, F]$. We give the formal definition as follows.

**Definition 1.2** (Band-limited signal interpolation). *Assume that the orignal signal $x^*(t)$ is $k$-sparse band-limited. Given the observations of the form $x^*(t) + g(t)$ where $g$ is an arbitrary noisy function, with the signal-to-noise ratio bounded below by a constant (e.g., $\|x^*\|_T \gtrsim \|g\|_T$, where $\|x\|_T^2 := T^{-d} \int_{[0,T]^d} |x(t)|^2 \mathrm{d}t$ denotes the signal's* energy*), the goal is to output the reconstructed signal $y(t)$ such that*

$$\|y - x^*\|_T \leq C\|g\|_T + \delta\|x^*\|_T$$

*where $C > 0$ is constant and $\delta \in (0, 1)$ is an accuracy parameter.*

In general, this kind of sparse recovery problems has a long history in signal-processing and computer science (Cooley and Tukey, 1965; Reynolds, 1989; Aibinu et al., 2008; Voelz, 2011; Hassanieh et al., 2012a,b; Ghazi et al., 2013; Indyk and Kapralov, 2014; Indyk et al., 2014; Boashash, 2015; Kapralov, 2016, 2017; Kapralov et al., 2019; Nakos et al., 2019; Jin et al., 2023; Song et al., 2023). A fundamental fact, pointed out in Moitra (2015), is that when the frequency gap is small ($\eta := \min_{i \neq j \in [k]} |f_i - f_j| < 1/T$), exact recovery of the signal is informational-theoretically impossible. Complementing this negative result, Price and Song (2015) gave a $k \cdot \mathrm{polylog}(k, FT/\delta)$-time $\delta$-error reconstruction algorithm for one-dimensional signals where $F$ is the band-limit, assuming the time domain satisfies $T > \Omega(\log^2(k/\delta)/\eta)$, and that the frequency gap $\eta$ is known. Chen et al. (2016) strengthened this result by showing that even if the frequency gap is unknown, *approximate* reconstruction of one-dimensional signals in $\mathrm{poly}(k, \log(FT))$-samples and time is possible, in the sense that the output signal is close to the original signal in the time domain albeit with worse sparsity in the frequency domain[2]. Subsequent works (Chen and Price, 2019b,a; Li et al., 2021a) have improved this result, both in sample-complexity and decoding time. Recently, Li et al. (2021a) improved the sparsity of the output signal from $\mathrm{poly}(k)$ to $k \cdot \mathrm{poly}\log(k)$, settling for a somewhat weaker notion of approximation[3] than that of Chen et al. (2016).

Unfortunately, despite this steady algorithmic progress, the approximation ratio achieved by all prior band-limited interpolation methods has stubbornly remained above a large constant factor (around 100). This gap is not merely an artifact of loose analysis: it stems from a fundamental "triangle-inequality bottleneck"—the three dominant error sources (frequency truncation, polynomial approximation, and linear regression) accumulate additively, and each was previously controlled only up to a constant factor. Closing this gap is crucial both for theory to understand the true limits of sparse recovery under noise and for practice, where large constant blow-ups translate into prohibitively low signal-to-noise requirements.

**Our contributions.** We break this long-standing barrier and obtain the *first* high-accuracy algorithm whose approximation factor is strictly below 5. Concretely, for any $k$-sparse band-limited signal observed over $[0, T]$ and frequency domain $\mathcal{F} = [-F, F]$ with additive noise $g(t)$, our main result guarantees a reconstructed signal $y(t)$ satisfying

$$\|y - x^*\|_T \leq (3 + \sqrt{2} + \varepsilon)\|g\|_T + \delta\|x^*\|_T$$

for arbitrary accuracy parameters $\varepsilon, \delta \in (0, 1)$, using only $\mathrm{poly}(k, \varepsilon^{-1}, \log(FT/\delta))$ samples and nearly-linear time. Here, the *approximation ratio* $C = 3 + \sqrt{2} + \varepsilon$ quantifies the multiplicative

---

[1]Classical work on band-limited interpolation typically first estimates frequencies and then magnitudes. The latter step can be cast as a Set-Query problem: given a collection of locations, recover the Fourier coefficients $\hat{x}$ at those positions. When the frequencies lie on a lattice, the two formulations coincide.

[2]More precisely, the error guarantee is $\|y(t) - x^*(t)\|_T \leq O(\|g(t)\|_T + \delta\|x^*(t)\|_T)$, where $x^*(t)$ is the original signal, $y(t)$ is the reconstructed signal, and $g(t)$ is the noise distribution.

[3]$\|y(t) - x^*(t)\|_{(1-c)T} \leq \mathrm{poly}(\log(k/c\delta)) \cdot \|g(t)\|_T + \delta\|x^*\|_T$.

factor by which the reconstruction error $\|y - x^*\|_T$ can exceed the unavoidable noise floor $\|g\|_T$; it is the standard measure of solution quality in approximation algorithms for signal recovery. This sharply improves the best previous constant $C > 100$ of Chen et al. (2016), narrows the gap to the information-theoretic optimum of 1, and resolves an open question posed by Li et al. (2021a). We also give a refined "shrinking-range" variant that achieves a $(\sqrt{2} + \varepsilon + c)$-approximation on any sub-interval $(1 - c)T$ with the same sample complexity.

**Technical novelties.**  Our improvement hinges on two new analytic ingredients.

1. **Ultra-high-sensitivity frequency estimation.** We design a filter family that amplifies each true cluster's energy while *canceling* an equal-scale portion of the adversarial noise. This raises the recoverable energy threshold from $\Theta(1)$ to $(1 + \sqrt{\varepsilon})$, eliminating an entire factor of 2 in the first error component.

2. **Hierarchical noise-cancellation analysis.** We view band-limited interpolation through a unified two-step lens—frequency estimation followed by signal estimation—and track the flow of noise across levels. A refined coupling argument shows that the filtered noise passed to the second step is *correlated* with the unrecoverable signal energy; bounding them jointly yields the multiplicative factor $(3 + \sqrt{2})$ instead of the additive sum of three constants.

Beyond improving the approximation ratio, our framework is conceptually modular: swapping in any future advances in either sub-routine immediately propagates to a tighter end-to-end guarantee. We believe the tools introduced here—particularly the systematic noise-cancellation bound and the lattice-frequency viewpoint—will be valuable well beyond the specific interpolation task, offering a blueprint for pushing other sparse-recovery algorithms past long-standing constant-factor barriers.

## 1.1   Main results

Recall that all existing algorithms for band-limited interpolation achieve only coarse error bounds of the form

$$\|y - x^*\|_T \leq C\|g\|_T + \delta\|x^*\|_T,$$

with a constant $\approx 100$ in the best published result (Chen et al., 2016); repeated uses of the triangle inequality prevent $C$ from dropping below 3. We introduce three new ingredients—*sharper noise control*, an ultra-sensitive frequency estimator, and an efficient signal-estimation routine—and combine them in a refined error analysis that collapses these additive losses into a single multiplicative term. This yields the first algorithm with a provable approximation constant strictly below 5, and it remains near-optimal in both sample complexity and running time.

Our first result uses a sharper error analysis improves $C$ to $3 + \sqrt{2} + \varepsilon$ for any small $\varepsilon > 0$, which is stated as follows.

**Theorem 1.3** (High-accuracy band-limited interpolation, informal version of Theorem H.42). *Let $x^*(t)$ be a $k$-sparse band-limited signal with frequencies in $[-F, F]$. Assume the minimum frequency separation is $\eta \geq \Omega(1/T)$ and the signal-to-noise ratio satisfies $\|x^*\|_T \gtrsim \|g\|_T$. Given observations $x(t) = x^*(t) + g(t)$ in time duration $[0, T]$, where $g$ is arbitrary noise. For $\varepsilon, \delta \in (0, 1)$, there exists an algorithm that uses $\mathrm{poly}(k, \varepsilon^{-1}, \log(1/\delta)) \log(FT)$ samples and runtime, and outputs a $\mathrm{poly}(k, \varepsilon^{-1}, \log(1/\delta))$-sparse band-limited signal $y(t)$ such that, with high probability,*

$$\|y - x^*\|_T \leq (3 + \sqrt{2} + \varepsilon)\|g\|_T + \delta\|x^*\|_T.$$

Our techniques extend to a "shrinking-range" variant that attains an even tighter constant on any sub-interval $(1 - c)T$, which leverages additional "spatial slack" to lower the approximation ratio to $\sqrt{2} + \varepsilon + c$.

**Theorem 1.4** (Ultra-high-accuracy band-limited interpolation with shrinking range, informal version of Theorem I.4). *Let $x^*(t)$ be a $k$-sparse band-limited signal with frequencies in $[-F, F]$. Assume the minimum frequency separation is $\eta \geq \Omega(1/T)$ and the signal-to-noise ratio satisfies $\|x^*\|_T \gtrsim \|g\|_T$. Given observations $x(t) = x^*(t) + g(t)$ in time duration $[0, T]$, where $g$ is arbitrary noise. Let $T' = T(1 - c)$. For $\varepsilon, \delta \in (0, 1)$, there exists an algorithm that uses $\mathrm{poly}(k, \varepsilon^{-1}, \log(1/\delta)) \log(FT)$ samples and $\mathrm{poly}(k, \varepsilon^{-1}, c^{-1}, \log(1/\delta)) \cdot \log^2(FT)$ runtime, and*

*outputs a* $\mathrm{poly}(k, \varepsilon^{-1}, c^{-1}, \log(1/\delta))$*-sparse band-limited signal* $y(t)$ *such that, with high probability,*

$$\|y - x^*\|_{T'} \leq (\sqrt{2} + \varepsilon + c)\|g\|_T + \delta\|x^*\|_T.$$

We remark that Li et al. (2021a) obtain a related but incomparable guarantee. For any $\delta > 0$, their algorithm outputs a reconstruction $y(t)$ satisfying

$$\|y(t) - x^*(t)\|_{(1-c)T} \leq \alpha\|g(t)\|_T + \delta\|\widehat{x}^*(f)\|_1,$$

where $\alpha$ is the approximation ratio, where $c \in (0,1)$ is the shrinking parameter, $\alpha = \mathrm{poly}(\log(k/(\delta c)))$, and $\widehat{x}^*(f)$ is the Fourier transform of $x^*(t)$.. The procedure uses $\mathrm{poly}(k, \log(1/\delta))\log(FT)$ samples and $\mathrm{poly}(k, c^{-1}, \log(1/\delta))\log^2(FT)$ time, returning a $k, \mathrm{poly}(1/c, \log(k/\delta))$-sparse signal. Hence their result achieves near-optimal sparsity, but the approximation factor grows polylogarithmically with $k$, $1/\delta$, and $1/c$, whereas our algorithm attains a constant $(\sqrt{2} + \varepsilon + c)$ ratio.

## 1.2 Notations

For any positive integer $n$, we use $[n]$ to denote $\{1, 2, \cdots, n\}$. We use $\mathbf{i}$ to denote $\sqrt{-1}$. For a complex number $z \in \mathbb{C}$ where $z = a + \mathbf{i}b$ and $a, b \in \mathbb{R}$. We use $\overline{z}$ to denote the complex conjugate of $z$, i.e., $\overline{z} = a - \mathbf{i}b$. Then it is obvious that $|z|^2 = z \cdot \overline{z} = a^2 + b^2$.

We use $f \lesssim g$ to denote that there exists a constant $C$ such that $f \leq Cg$, and $f \eqsim g$ to denote $f \lesssim g \lesssim f$. We use $\widetilde{O}(f)$ to denote $f \log^{O(1)}(f)$. We say $x(t)$ is a $k$-sparse band-limited when $x(t) = \sum_{j=1}^{k} v_j \exp(2\pi\mathbf{i}f_j t)$. We use $\widehat{x}(f)$ to denote the Fourier transform of $x(t)$. More specifically, $\widehat{x}(f) = \int_{-\infty}^{\infty} x(t) \exp(-2\pi\mathbf{i}ft)\mathrm{d}t$.

We define our discrete norm as $\|g(t)\|_S^2 = \frac{1}{|S|}\sum_{t \in S}|g(t)|^2$ for function $g$. We define our weighted discrete norm as $\|g(t)\|_{S,w}^2 = \sum_{t \in S} w_t|g(t)|^2$ for function $g$. We define the continuous $T$-norm as $\|g(t)\|_T^2 = \frac{1}{T}\int_0^T |g(t)|^2\mathrm{d}t$ for function $g$.

In general, we assume $x^*(t)$ is our ground truth and is a $k$-sparse band-limited signal. We can observe function $x(t) = x^*(t) + g(t)$ for $g(t)$ being a noise function. We can observe $x(t)$ in duration $[0, T]$. The ground truth $x^*(t)$ has frequencies in $[-F, F]$.

## 2 Technical Overview

Section 2.1 introduces the framework of discrete Fourier set query. Section 2.2 shows how to apply our discrete Fourier set query estimation algorithm to obtain a high-accuracy band-limited interpolation algorithm as in Theorem 1.3 and Theorem 1.4.

### 2.1 Discrete Fourier Set Query

Many signal processing tasks can be phrased as set query problems in different domains. For instance, the sparse Fourier transform examines only the coefficients at a small set of frequencies—exactly a set-query problem in the frequency domain. Another example is recovering the actual Fourier coefficients when their *support* (the set of non-zero frequencies) is known.

For concreteness we restrict attention to a one-dimensional discrete signal $x(t)$ of length $n$, written as $x(t) = \sum_{j=1}^{n} \widehat{x}_j e^{2\pi\mathbf{i}jt/n}, t \in [n]$. Given a $k$-subset $S \subset [n]$, the set–query task is to recover $\widehat{x}_S$. Equivalently, define $x_S(t) = \sum_{f \in S} \widehat{x}_f e^{2\pi\mathbf{i}ft/n}$; this is a $k$-sparse signal.

**Algorithm overview.** Our algorithm can be viewed as a three-step pipeline, which collapses into three concrete stages in practice:

1. **Uniform sketching.** Any discrete $k$-sparse signal has energy bound $R = k$, meaning $\sup_t |x(t)|^2 \leq R\|x\|_T^2$. Consequently, a *uniform* sample $S_0 \subset [n]$ of size $O(k \log k)$ already preserves the signal's energy up to a constant factor and forms a faithful oblivious sketch.

2. **Sketch distillation.** We refine $S_0$ via the *Sketch Distillation* procedure: a well-balanced sampler chooses a linear-size subset $S_1 \subset S_0$ and weights $w$ such that $\|x_S\|_{S_1,w} \approx \|x_S\|_2/n$ for every signal supported on $S$, while simultaneously ensuring the weighted energy of the orthogonal component $x_{\overline{S}} + g$ is *not* amplified, i.e. $\|x_{\overline{S}} + g\|_{S_1,w} = O(\|x_{\overline{S}} + g\|_T)$.

3. **Weighted regression.** With samples $\{x(t)\}_{t \in S_1}$ and weights $w$, we solve the weighted least-squares problem $\min_{v' \in \mathbb{C}^k} \|\sqrt{w} \circ (Av' - b)\|_2$, where $A_{i,j} = e^{2\pi \mathbf{i} f_j t_i/n}$ and $b_i = x(t_i)$. The solution $\widehat{x}_S(f_j) = v'_j$ yields a reconstruction whose error obeys $\|\widehat{x}_S - x_S\|_T \leq \varepsilon \|x_{\overline{S}} + g\|_T$.

Together, these steps give a linear-sample, high-accuracy solution to the discrete Fourier set-query problem, achieving $(1 + \varepsilon)$ approximation with high probability. A routine analysis gives an $O(1)$ approximation; below we refine it to $(1 + \varepsilon)$.

---

**Algorithm 1** Discrete 1-D Fourier Set-Query

---

1: **procedure** SETQUERY($x$, $n$, $k$, $S$, $\varepsilon$)
2:     $\{f_1, \ldots, f_k\} \leftarrow S$
   /* Step 1: Uniform sketching */
3:     $S_0 \leftarrow$ Sample $O(\varepsilon^{-2} k \log k)$ points i.i.d. from $\mathrm{Uniform}([n])$
   /* Step 2: Sketch distillation */
4:     $\mathcal{F} \leftarrow \left\{ \sum_{j=1}^{k} v_j e^{2\pi \mathbf{i} f_j t/n} \mid v_j \in \mathbb{C} \right\}$
5:     $(\{t_1, \ldots, t_s\}, w) \leftarrow$ RANDBSS+$\left(k, \mathcal{F}, \mathrm{Uniform}(S_0), (\varepsilon/4)^2\right)$             ▷ Algorithm 3
   /* Step 3: Weighted regression */
6:     **for** $(i, j) \in [s] \times [k]$ **do**
7:         $A_{i,j} \leftarrow e^{2\pi \mathbf{i} f_j t_i/n}$
8:     **end for**
9:     **for** $i \in [s]$ **do**
10:        $b_i \leftarrow x(t_i)$                                      ▷ observe $x$ at $t_i$
11:     **end for**
12:     $v' \leftarrow \arg \min_{v' \in \mathbb{C}^k} \|\sqrt{w} \circ (Av' - b)\|_2$
13:     **return** $v'$
14: **end procedure**

---

**Composition of well-balanced samplers.** To obtain a $(1 + \varepsilon)$ guarantee we must show that the final sketch $(S_1, w)$ arises from a single *well-balanced sampling procedure* (WBSP). In general, composing two WBSPs may violate well-balancedness: while the first property (accurate energy estimation for every $f \in \mathcal{F}$) is preserved, the second property concerning weight sum and condition number can fail.

Our setting is special: the first sampler draws each point *uniformly* from $[n]$. We prove that, under this choice and with the tight energy bound $R = k$ for band-limited signals, each sample produced by the two-stage composition is distribution-equivalent to a fresh uniform draw. Hence the composite sampler is itself well-balanced, allowing us to invoke the sharper error analysis and conclude that

$$\|\widehat{y}_S - \widehat{x}_S\|_2 \leq \varepsilon \|\widehat{x}_{\overline{S}}\|_2^2$$

with high probability. Thus we obtain a linear-sample, high-accuracy algorithm for the discrete Fourier set-query problem.

## 2.2 High-accuracy band-limited interpolation

In this section, we introduce how to obtain a high-accuracy one-dimensional band-limited interpolation algorithm (Theorem 1.3), which improves the constant-accuracy algorithm by Chen et al. (2016).

Let us briefly summarize the previous algorithm in Chen et al. (2016). The high-level idea is to first find some small intervals in the frequency domain such that each contains some significant frequencies of the signal $x^*$. (These intervals are called "heavy-clusters" in their paper.) Then, they use some filter techniques (also used in Price and Song (2015)) to reduce the problem of reconstructing $x^*$, a signal with multiple heavy-clusters to several single heavy-cluster signals. Then, for each single

heavy-cluster signal, since the band-limit is small, they can efficiently estimate its frequencies. Finally, they reconstruct a $\mathrm{poly}(k)$-sparse signal that is close to $x^*$ via a robust polynomial learning algorithm. More specifically, their algorithm consists of the following steps:

1. They show that the ground-truth signal $x^*(t) = \sum_{j=1}^k v_j e^{2\pi \mathbf{i} f_j t}$ can be approximated by $x_S(t) = \sum_{j \in S} v_j e^{2\pi \mathbf{i} f_j t}$, where $S := \{ j \in [k] : f_j \in \text{some heavy-cluster } C_i \}$ is the set of frequencies in the heavy-clusters. This step will cause an approximation error $E_1 := \|x^* - x_S\|_T \leq 1.2\mathcal{N}$[4], where $\mathcal{N}^2 := \|g\|_T^2 + \delta \|x^*\|_T^2$ appears in the approximation error of the band-limited interpolation problem.

2. They solve a Frequency Estimation problem for $x_S$ using the filter techniques and multiple-to-one heavy-cluster reduction, and get a list $L$ of $O(k)$ candidate frequencies so that for each $j \in S$, $f_j$ is close to some $\widetilde{f}_{p_j} \in L$.

3. The signal $x_S(t)$ can be decomposed into $\sum_{i=1}^{|L|} e^{2\pi \mathbf{i} \widetilde{f}_i t} \cdot x_i^*(t)$, where $x_i^*(t) := \sum_{j:p_j=i} e^{2\pi \mathbf{i}(f_j - \widetilde{f}_i)}$ is a one-cluster signal with small band-limit. For each $x_i^*(t)$, they prove that there exists a low-degree polynomial $P_i(t)$ that can approximate it. Let's denote the polynomial-approximated signal $\sum_{i=1}^{|L|} e^{2\pi \widetilde{f}_i t} \cdot P_i(t)$ by $x_{S,\mathsf{poly}}(t)$, which has an approximation error $E_2 := \|x_S - x_{S,\mathsf{poly}}\|_T \leq \delta \|x_S\|_T$.[5]

4. It remains to reconstruct $x_{S,\mathsf{poly}}(t)$, which is a variant of Signal Estimation problem. They use a sampling-and-regression approach to obtain a $\mathrm{poly}(k)$-sparse signal $y(t)$ with an approximation error $E_3 := \|y - x_{S,\mathsf{poly}}\|_T \leq 2200\mathcal{N}$.

By triangle inequality, the total approximation error is $\|y - x^*\|_T \leq E_1 + E_2 + E_3 \leq C \cdot \mathcal{N}$, where $C > 1000$ is an absolute constant.

**3-approximation barrier** To achieve high-accuracy band-limited interpolation, we need to control the errors $E_1$, $E_2$, and $E_3$.

- For $E_1$, it is coupled with the second step, since the approximation error of $x_S$ is connected to the significance of each heavy-cluster. If we choose a too-small $E_1$, $x_S$ will contain some not-so-significant frequencies, and the Frequency Estimation algorithm may not be able to find them. Thus, with the techniques in Chen et al. (2016), we cannot make $E_1$ to be less than $\mathcal{N}$. Even worse, due to the noise $g$ in the observed signal, the error will be at least $2\mathcal{N}$.

- For $E_2$, it only depends on the error parameter $\delta$. Since the sample and time complexities of the algorithm only depend logarithmically on $1/\delta$, it allows us to re-scale $\delta$ and make $E_2$ very small.

- For $E_3$, where we lose a big constant, we need a high-accuracy Signal Estimation algorithm to recover the polynomial-approximated signal $x_{S,\mathsf{poly}}(t)$. However, as we discussed in the previous sections, the error of the Signal Estimation will be at least $\mathcal{N}$.

Therefore, there is a 3-approximation barrier in Chen et al. (2016)'s approach due to the triangle inequality.

In the remainder of this section, we first introduce our techniques to achieve a $(7 + \varepsilon)$-approximation error. Then, we show how to overcome the barrier and achieve a $(3 + \sqrt{2} + \varepsilon)$-approximation error.

### 2.2.1 $(7 + \varepsilon)$-approximation algorithm

**High sensitivity frequency estimation** We first improve $E_1$ from $1.2\mathcal{N}$ to $(1 + \varepsilon)\mathcal{N}$ by proposing a high sensitivity frequency estimation method. More specifically, to identify the heavy-clusters of

---

[4]Due to the noisy observations, not every frequency in the heavy-clusters is recoverable. This gap causes an extra implicit error term in Chen et al. (2016), which is about $12\mathcal{N}$. Section 2.2.1 has a more detailed discussion.

[5]We remark that even if the ground-truth signal $x^*(t)$ can be well-approximated by a mixed Fourier-polynomial signal $\widetilde{x}(t)$, we are unable to recover *every* basis of $\widetilde{x}(t)$ due to the limitation of the frequency estimation procedure. Thus, directly applying linear regression to *partially* reconstruct $\widetilde{x}(t)$ will not guarantee to be a $(1 + \varepsilon)$-approximation of $x^*(t)$.

the signal $x^*$, Chen et al. (2016) designed a filter function $H$ such that $H \cdot x^*$ has high energy in each heavy cluster $C_i$; that is,

$$\int_{C_i} |\widehat{H \cdot x^*}(f)|^2 \mathrm{d}f \geq \frac{T}{k} \mathcal{N}^2. \tag{1}$$

Moreover, $H$'s frequencies are contained in a small interval of length $\Delta$. These two properties imply that for any true frequency $f_i \in C_i$, the signal $H \cdot x^*$ with frequency domain restricted to $[f_i - \Delta, f_i + \Delta]$ is a one heavy-cluster signal with small band-limit, which allows us to use Price and Song (2015)'s approach to estimate $f_i$. The filter function $H$ in Chen et al. (2016) is only $O(1)$-sensitive, which means it can concentrate a constant fraction of the signal's energy. And for those less important frequencies, they cannot be clustered by $H$ and will be lost in the frequency estimation procedure.

We manage to modify their filter construction and obtain a $(1 - \varepsilon)$-sensitive filter function $H$ such that the signal $x_{S^*}$ consisted of the frequencies in the new heavy-clusters has about $(1 - \varepsilon)$-fraction of energy of $x^*$. More specifically, we have

$$E_1^{\mathsf{new}} = \|x^* - x_{S^*}\|_T \leq (1 + \varepsilon)\mathcal{N}.$$

To prove that we can actually estimate the frequencies in the new heavy-clusters, we observe a subtle point: the energy condition of heavy-cluster and the energy condition of frequency estimation are inconsistent due to the *noise* in observations. To be able to estimate the one heavy-cluster signal's frequency, it is required that

$$\int_{C_i} |\widehat{H \cdot x}(f)|^2 \mathrm{d}f \geq \frac{T}{k} \mathcal{N}^2, \tag{2}$$

which is different from Eq. (1). In other words, not all frequencies in $S^*$ are recoverable, but only most of them. Since Chen et al. (2016) only wants a constant approximation error, they may simply ignore this difference by losing a constant factor in accuracy. For us, however, we need to make it precise. We define $S$ to be a subset of $S^*$ containing the frequencies in the heavy-clusters satisfying Eq. (2). We analyze the effect of $H \cdot g$ and show that by strengthening the RHS of heavy-cluster's energy condition (Eq. (1)) to $\frac{4T}{k} \mathcal{N}^2$, we can bound the unrecoverable part's energy by

$$E_{1.5} := \|x_{S^*} - x_S\|_T \leq (2 + \varepsilon)\mathcal{N}.$$

For the recoverable part $x_S$, we can just follow Chen et al. (2016)'s approach to estimating the frequencies in each heavy-cluster.

**Generalized high-accuracy signal estimation**   We apply our three-step Fourier set-query framework to solve the signal estimation problem in the forth step of Chen et al. (2016)'s algorithm and improve $E_3$ from $2200\mathcal{N}$ to $(4 + \varepsilon)\mathcal{N}$. We first define the problem more formally. By frequency estimation, we obtain a list of candidate frequencies of $x_S$ and in the third step, we know that it can be approximated by $x_{S,\mathsf{poly}}(t) := \sum_{i=1}^{|L|} e^{2\pi \mathbf{i} \widetilde{f}_i t} \cdot P_i(t)$ with very tiny error $E_2$, where $P_i(t)$ are some degree-$d$ polynomials. We can rewrite $x_{S,\mathsf{poly}}$ in the *Fourier-monomial mixed basis*:

$$x_{S,\mathsf{poly}}(t) = \sum_{i=1}^{|L|} \sum_{j=0}^{d} v_{i,j} \cdot e^{2\pi \mathbf{i} \widetilde{f}_i t} t^j,$$

where $v_{i,j} \in \mathbb{C}$ and $\widetilde{f}_i \in L$ are known. That is, we need to learn $\{v_{i,j}\}$ given noisy observations $x_{S,\mathsf{poly}}(t) + g'(t)$, which is a Signal Estimation problem for the following family of signals:

$$\mathcal{F}_{\mathsf{mix}} := \mathrm{span} \left\{ e^{2\pi \mathbf{i} \widetilde{f}_i t} \cdot t^j \;\middle|\; i \in [|L|], j \in \{0, \cdots, d\} \right\}.$$

We apply our three-step framework as follows.

- In Step 1, we first need an energy bound $\mathcal{F}_{\mathsf{mix}}$. Chen et al. (2016) showed that

$$R_{\mathsf{mix}} := \sup_{u(t) \in \mathcal{F}_{\mathsf{mix}}} \frac{\sup_t |u(t)|^2}{\|u\|_T^2} \leq \widetilde{O}(|L|^4 d^4).$$

  Then we can get that uniformly sample $\widetilde{O}(|L|^4 d^4 \varepsilon^{-1})$ points in $[0, T]$ gives an oblivious sketching for $x_{S,\mathsf{poly}}$. Furthermore, we can show that this sampler is $\varepsilon$-well-balanced.

- In Step 2, since we aim at achieving high-accuracy, we do not distill the sketch but directly apply the sharper error analysis to control the energy of the orthogonal part of noise.
- In Step 3, we solve a weighted linear regression to estimate the coefficients and obtain a signal $y'(t) \in \mathcal{F}_{\mathsf{mix}}$ such that

$$E_3^{\mathsf{new}} := \|y' - x_{S,\mathsf{poly}}\|_T \leq (1+\varepsilon)\|x - x_{S,\mathsf{poly}}\|_T \leq (4+\varepsilon)\mathcal{N}.$$

Then, we can transform $y$ back to a $\mathrm{poly}(k)$-sparse signal with error $\|y - y'\|_T \leq E_2$.

Combining them together and re-scaling $\varepsilon$ and $\delta$, we get that

$$\begin{aligned}
\|y - x^*\|_T &\leq \|y - y'\|_T + \|y - x_{S,\mathsf{poly}}\|_T + \|x_{S,\mathsf{poly}} - x_S\|_T + \|x_S - x_{S^*}\|_T + \|x_{S^*} - x^*\|_T \\
&\leq E_2 + E_3^{\mathsf{new}} + E_2 + E_{1.5} + E_1^{\mathsf{new}} \\
&\leq (7+\varepsilon)\|g\|_T + \delta\|x^*\|_T.
\end{aligned}$$

Therefore, we obtain a band-limited interpolation algorithm with $(7+\varepsilon)$-approximation error.

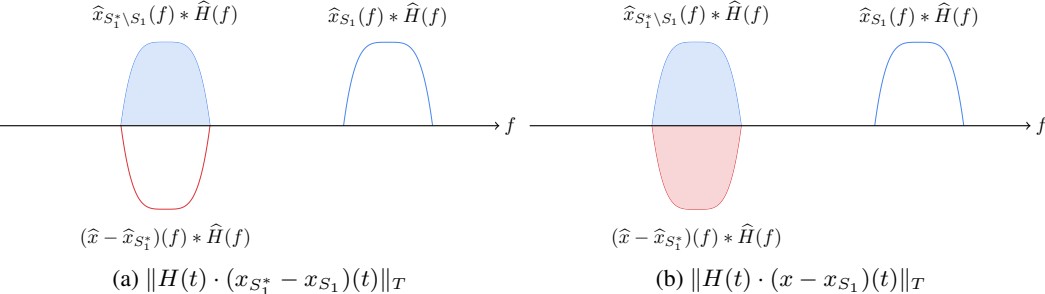

(a) $\|H(t) \cdot (x_{S_1^*} - x_{S_1})(t)\|_T$          (b) $\|H(t) \cdot (x - x_{S_1})(t)\|_T$

Figure 1: An illustration of the signal-noise cancellation effect (Eq. (3)). In (a), the blue region corresponds to the first term of Eq. (3), which roughly equals the energy of $x - x_{S_1^*}$. In (b), the red and blue regions correspond to the second term, where most of the energy is canceled. Thus, their total energy is very close to $\|x - x_{S_1^*}\|_T$.

### 2.2.2   $(3 + \sqrt{2} + \varepsilon)$-approximation algorithm

How can we further improve this algorithm? We observe that $E_3^{\mathsf{new}}$ can be written more precisely as $(1+\varepsilon)\|g\|_T + (3+\varepsilon)\mathcal{N}$. On the other hand, $E_1^{\mathsf{new}}$ and $E_{1.5}$ only depend on $\mathcal{N}$. If we can take a smaller value for $\mathcal{N}^2$, i.e., $\varepsilon(\|g\|_T^2 + \delta\|x\|_T^2)$, then we will improve approximation error. We show that it is possible via an *ultra-high sensitivity frequency estimation method*.

**Ultra-high sensitivity frequency estimation**   To improve the sensitivity of the frequency estimation method, let $\mathcal{N}_1^2 := \varepsilon\mathcal{N}^2$ and consider the heavy-clusters with parameter $\mathcal{N}_1$. Let $S_1^*$ denote the set of frequencies of $x^*(t)$ in the $\mathcal{N}_1$-heavy-clusters. By the same analysis as in our previous frequency estimation approach, we have $E_1^{\mathsf{new}+} := \|x^* - x_{S_1^*}\|_T \leq (1+\varepsilon)\mathcal{N}_1$.

However, due to the inconsistent energy conditions, only those frequencies in the heavy-clusters satisfying Eq. (2) are recoverable. Let $S_1$ denote the set of such frequencies, and we need to upper bound $\|x_{S_1^*} - x_{S_1}\|_T$. Previously, we strengthen the heavy-cluster's condition (Eq. (1)) and get a $E_{1.5} \leq (2+\varepsilon)\|g\|_T$ bound. Here, instead, we relax the RHS of Eq. (2) to $\varepsilon \cdot \frac{T}{k}\mathcal{N}_1^2$. Intuitively, more frequencies will satisfy the new frequency estimation condition; and if there is an unrecoverable frequency $f^* \in S_1^* \setminus S_1$, it indicates that its contribution in filtered signal $H \cdot x^*$ is cancelled out by the filtered noise $H \cdot g$. Using this *signal-noise cancellation effect*, we prove that:

$$\|H(x_{S_1} - x_{S_1^*})\|_T^2 + \|H(x - x_{S_1})\|_T^2 \leq (1+\sqrt{\varepsilon})\|x - x_{S_1^*}\|_T^2, \tag{3}$$

which saves a factor of 2 from $E_{1.5}$ by introducing an extra term $\|H(x - x_{S_1})\|_T$. Recall $\|x - x_{S_1}\|_T$ is related to $E_3^{\mathsf{new}}$, the error of the signal estimation procedure. We can decompose it into the "passing energy" $\|H(x - x_{S_1})\|_T$ and "filtered energy" $\|(\mathrm{Id} - H)(x - x_{S_1})\|_T$ and bound them by:

$$\|x - x_{S_1}\|_T \leq \|H(x - x_{S_1})\|_T + \|g\|_T + O(\varepsilon)\|x^* - x_{S_1}\|_T.$$

Thus, Eq. (3) can be considered as upper-bounding $E_{1.5}$ and $E_3^{\text{new}}$ simultaneously. Combining them together, we get the following error guarantee for the frequency recoverable signal $x_{S_1}$:

$$\|x - x_{S_1}\|_T + \|x^* - x_{S_1}\|_T \le (1 + \sqrt{2} + O(\sqrt{\varepsilon}))\|g\|_T + O(\sqrt{\delta})\|x^*\|_T. \tag{4}$$

Then, by a more careful analysis of the HASHTOBINS approach used by Chen et al. (2016) for Frequency Estimation, we show that $x_{S_1}$'s frequencies can be efficiently approximated, which gives an ultra-high sensitivity frequency estimation method.

The remaining part of the algorithm is almost identical to the previous one. We run the high-accuracy signal estimation algorithm to reconstruct $x_{S_1}$. Let $y(t)$ denote the output band-limited signal. By Eq. (4) and re-scaling $\varepsilon$ and $\delta$, we have

$$\|y - x^*\|_T \le (3 + \sqrt{2} + \varepsilon)\|g\|_T + \delta\|x^*\|_T.$$

Therefore, we achieve a $(3 + \sqrt{2} + \varepsilon)$-approximation error for the band-limited interpolation.

### 2.2.3 $(\sqrt{2} + \varepsilon + c)$-approximation algorithm with shrinking range

When we only care about the signal on the interior window $[0, (1 - c)T]$, the two "edge strips" of length $cT/2$ at the beginning and end become disposable budget. We exploit this slack with a *shrinking-range filter* $H$ that (i) leaves the interior almost unchanged and (ii) suppresses the contribution of every frequency band whose energy is already comparable to the noise. This ensures that, after filtering, *all* irrecoverable energy cancels with the adversarial noise, so the residual we must approximate is strictly smaller than in the full-range setting.

Using the same high-sensitivity filter construction, but tuned to the relaxed threshold $\varepsilon_1\|g\|_T^2$, we identify a set $S$ of "truly heavy" frequencies inside each cluster. We show that (see Lemma I.1)

$$\|x - x_S\|_{T'} + \|x_S - x^*\|_{T'} \le (\sqrt{2} + O(\sqrt{\varepsilon} + c))\|g\|_T + O(\sqrt{\delta})\|x^*\|_T.$$

Then we can recovers, for every $f_j \in S$, an approximation $\widetilde{f}_j$ with resolution $O(\Delta_0\sqrt{\Delta_0 T})$ using only $\text{poly}(k, \varepsilon^{-1}, c^{-1})\log(FT)$ samples. See Corollary I.2 for more details.

Then we can replace each narrow band around $\widetilde{f}_j$ by a low-degree polynomial $P_j(t)$ (degree $d = O(T\Delta_0^{3/2} + k^3 \log k)$) whose Fourier support stays inside the band and whose time-domain error is $O(\delta)\|x_S\|_T$.

On $T' = (1 - c)T$ we run the high-accuracy signal-estimation routine (Section G.2), but now with the *uniform* sampler restricted to $T'$. Because the filter already tames boundary energy, a linear-sized well-balanced sample suffices to learn the coefficients with relative error $1 + \varepsilon$.

**Putting it together.** Summing the four error sources—cluster trimming, frequency rounding, polynomial patching, and weighted regression—and rescaling $\varepsilon, \delta$ yields Theorem 1.4: for any $c \in (0, 1)$,

$$\|y - x^*\|_{T'} \le (\sqrt{2} + \varepsilon + c)\|g\|_T + \delta\|x^*\|_T,$$

with sample complexity and runtime $\text{poly}(k, \varepsilon^{-1}, c^{-1}, \log(1/\delta))\log^{O(1)}(FT)$, and output sparsity $\text{poly}(k, \varepsilon^{-1}, c^{-1}, \log(1/\delta))$. Thus, shrinking the range lets us push the approximation constant all the way down to $\sqrt{2} + \varepsilon + c$.

## 3 Conclusion

In this work, we break the long-standing constant-factor barrier for noisy band-limited interpolation. Our primary algorithm delivers a $(3 + \sqrt{2} + \varepsilon)$-approximation with $\text{poly}(k, \varepsilon^{-1}, \log(1/\delta))$ sample and time complexity. A refined "shrinking-range" variant pushes the constant down to $(\sqrt{2} + \varepsilon + c)$ on any interior window $(1 - c)T$, demonstrating that additional spatial slack can be traded directly for reconstruction accuracy. Two technical ingredients drive these improvements: We introduce a new filter family simultaneously magnifies cluster energy and cancels adversarial noise, allowing

reliable recovery at an energy threshold arbitrarily close to the information-theoretic optimum. We also proved a unified view of frequency and signal estimation tracks how residual noise propagates through each stage, replacing three additive error terms with a single multiplicative bound and collapsing the approximation constant. These techniques are modular and extend naturally to broader sparse-recovery settings, opening avenues for even tighter guarantees in higher dimensions, alternate transform domains, and streaming environments.

Our guarantees rely on several idealised assumptions: the signal must be exactly $k$-sparse in the frequency domain (or perfectly lattice-aligned), the signal-to-noise ratio must exceed a constant threshold, and the analysis is fully worked out only in one dimension; relaxing any of these conditions can inflate the hidden polynomial factors in our sample and runtime bounds or even invalidate recovery. Moreover, the ultra-sensitive filters we employ require high numerical precision—round-off error or model mismatch may erode the stated constants in practical deployments. It remains the open whether these assumptions can be relaxed and further improve the approximation ratio.

## Acknowledgment

We thank anonymous NeurIPS reviewers for their constructive comments.

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

# Appendix

## A  Preliminaries

This section is organized as follows. In Section A.1, we provide some technical tools in probability theory and linear algebra. In Section A.2, we review the Fourier transformation for different types of signals.

### A.1  Tools and inequalities

**Lemma A.1** (Chernoff Bound Chernoff (1952)). *Let $X_1, X_2, \cdots, X_n$ be independent random variables. Assume that $0 \leq X_i \leq 1$ always, for each $i \in [n]$. Let $X = X_1 + X_2 + \cdots + X_n$ and $\mu = \mathbb{E}[X] = \sum_{i=1}^{n} \mathbb{E}[X_i]$. Then for any $\varepsilon > 0$,*

$$\Pr[X \geq (1+\varepsilon)\mu] \leq \exp(-\frac{\varepsilon^2}{2+\varepsilon}\mu) \text{ and } \Pr[X \leq (1-\varepsilon)\mu] \leq \exp(-\frac{\varepsilon^2}{2}\mu).$$

**Definition A.2** ($\varepsilon$-net). *Let $T$ be a metric space with distance measure $d$. Consider a subset $K \subset T$ and let $\varepsilon > 0$. A subset $\mathcal{N} \subseteq K$ is called an $\varepsilon$-net of $K$ if every point in $K$ is within distance $\varepsilon$ of some point of $\mathcal{N}$, i.e.*

$$\forall x \in K, \exists y \in \mathcal{N} \text{ s.t. } d(x,y) \leq \varepsilon.$$

**Fact A.3** (Fast matrix multiplication). *We use $\mathcal{T}_{\mathrm{mat}}(a,b,c)$ to denote the time of multiplying an $a \times b$ matrix with another $b \times c$ matrix.*

*We use $\omega$ to denote the exponent of matrix multiplication, i.e., $\mathcal{T}_{\mathrm{mat}}(n,n,n) = n^\omega$. Currently $\omega \approx 2.373$ Williams (2012); Le Gall (2014); Alman and Williams (2021).*

**Fact A.4** (Weighted linear regression). *Given a matrix $A \in \mathbb{C}^{n \times d}$, a vector $b \in \mathbb{C}^n$ and a weight vector $w \in \mathbb{R}_{>0}^n$, it takes $O(nd^{\omega-1})$ time to output an $x'$ such that*

$$x' = \arg\min_x \|\sqrt{W}(Ax - b)\|_2 = (A^*WA)^{-1}A^*Wb.$$

*where $\sqrt{W} := \mathrm{diag}(\sqrt{w_1}, \ldots, \sqrt{w_n}) \in \mathbb{R}^{n \times n}$, and $\omega \approx 2.373$ is the exponent of matrix multiplication Williams (2012); Le Gall (2014); Alman and Williams (2021).*

**Fact A.5.** *For any $x \in (0,1)$, we have $\cos(x) \leq \exp(-x^2/2)$.*

### A.2  Basics of Fourier transformation

The definition of high dimensional Fourier transform is as follows:

$$\widehat{x}(f) = \int_{(-\infty,\infty)^d} x(t) \exp(-2\pi\mathbf{i}\langle f, t\rangle) \mathrm{d}t, \text{ where } f \in \mathbb{R}^d,$$

and the definition of high dimensional inverse Fourier transform is as follows:

$$x(t) = \int_{(-\infty,\infty)^d} \widehat{x}(f) \exp(2\pi\mathbf{i}\langle f, t\rangle) \mathrm{d}f, \text{ where } t \in \mathbb{R}^d.$$

Note that when we replace $d = 1$ in the definition of high dimensional Fourier transform and inverse Fourier transform above, we get the definition of one-dimensional Fourier transform and inverse Fourier transform.

The definition of discrete Fourier transform is as follows:

$$\widehat{x}_f = \sum_{t=1}^{n} x_t \exp(-2\pi\mathbf{i}ft/n), \text{ where } f \in [n],$$

and the definition of discrete inverse Fourier transform is as follows:

$$x_t = \frac{1}{n}\sum_{f=1}^{n} \widehat{x}_f \exp(2\pi\mathbf{i}ft/n), \text{ where } t \in [n].$$

A continuous $k$-Fourier sparse signal $x(t) : \mathbb{R}^d \to \mathbb{C}$ can be represented as follows:

$$x(t) = \sum_{j=1}^{k} v_j \exp(2\pi \mathbf{i} \langle f_j, t \rangle), \ v_j \in \mathbb{C}, f_j \in \mathbb{R}^d, \ \forall j \in [k].$$

Thus, $\widehat{x}(f)$ is:

$$\widehat{x}(f) = \sum_{j=1}^{k} v_j \delta(t - f_j).$$

A discrete $k$-Fourier sparse signal $x \in \mathbb{C}^n$ can be represented as follows:

$$x_t = \sum_{j \in S} v_j \exp(2\pi \mathbf{i} jt/n), \ S \subseteq [n], |S| = k, \ v_j \in \mathbb{C}, \forall j \in S.$$

So, $\widehat{x}_f$ is:

$$\widehat{x}_f = \begin{cases} v_j & , j \in S \\ 0 & , \text{o.w.} \end{cases}$$

# B  Definitions of Semi-Continuous Fourier Set Query and Interpolation

In this section, we give the formal definitions of the problems studied in this paper. In Section B.1, we define the Fourier set query for discrete and continuous signals. In Section B.2, we define the band-limited interpolation problem and its two sub-problems: frequency estimation and signal estimation. And in Section B.3, we discuss the importance sampling method.

## B.1  Formal definitions of Fourier set query

The discrete Fourier set query problem is defined as follows:

**Definition B.1** (Discrete Fourier set query problem). *Let $x \in \mathbb{C}^n$ and $\widehat{x}$ be its discrete Fourier transformation. Let $\varepsilon > 0$. Given a set $S \subseteq [n]$ and query access to $x$, the goal is to use a few queries to compute a vector $x'$ with support $\text{supp}(x') \subseteq S$ such that*

$$\|(x' - \widehat{x})_S\|_2^2 \leq \varepsilon \cdot \|\widehat{x}_{[n] \setminus S}\|_2^2.$$

We also define the continuous Fourier set query problem as follows:

**Definition B.2** (Continuous Fourier set query problem). *For $d \geq 1$, let $x^*(t)$ be a signal in time duration $[0, T]^d$. Let $\widehat{x^*}(f)$ denote the continuous Fourier transformation of $x^*(t)$. Let $\varepsilon > 0$. Given a set $S \subseteq \mathbb{R}^d$ of frequencies such that $\text{supp}(\widehat{x^*}) \subseteq S$, and observations of the form $x(t) = x^*(t) + g(t)$, where $g(t)$ denotes the noise. The goal is to output a Fourier-sparse signal $x'(t)$ with support $\text{supp}(\widehat{x'}) \subseteq S$ such that*

$$\|x' - x^*\|_T^2 \leq (1 + \varepsilon) \cdot \|g\|_T^2.$$

## B.2  Formal definitions of semi-continuous band-limited interpolation

In this section, we provide the following formal definition of the semi-continuous band-limited interpolation problem, where we assume that the frequencies of the signal are contained in a lattice.

**Problem B.3** (Semi-continuous band-limited interpolation problem). *Given a basis $\mathcal{B}$ of $m$ known vectors $b_1, b_2, \cdots b_m \in \mathbb{R}^d$, let $\Lambda(\mathcal{B}) \subset \mathbb{R}^d$ denote the lattice*

$$\Lambda(\mathcal{B}) = \left\{ z \in \mathbb{R}^d : z = \sum_{i=1}^{m} c_i b_i, c_i \in \mathbb{Z}, \forall i \in [m] \right\}$$

*Suppose that $f_1, f_2, \cdots, f_k \in \Lambda(\mathcal{B})$, $\forall i \in [k], |f_i| \leq F$. Let $x^*(t) = \sum_{j=1}^{k} v_j e^{2\pi \mathbf{i} \langle f_j, t \rangle}$, and let $g(t)$ denote the noise. Given observations of the form $x(t) = x^*(t) + g(t)$, $t \in [0, T]^d$. Let $\eta = \min_{i \neq j} \|f_j - f_i\|_\infty$. There are three goals:*

1. *The first goal is to design an algorithm that output $f_1, f_2, \cdots, f_k$ exactly given query access to the signal $x(t)$ for $t \in [0, T]^d$.*

2. *The second goal is to design an algorithm that output a set $L$ of frequencies such that, for each $f_i$, there is $f_i' \in L$, $\|f_i - f_i'\|_2 \le D/T$.*

3. *The third goal is to design an algorithm that output $y(t) = \sum_{j=1}^{\widetilde{k}} v_j' \cdot e^{2\pi i f_j' t}$ such that $\int_{[0,T]^d} |y(t) - x(t)|^2 \mathrm{d}t \lesssim \int_{[0,T]^d} |g(t)|^2 \mathrm{d}t$.*

Then, we extract two sub-problems from Problem B.3: Frequency Estimation and Signal Estimation. We give their definitions below.

We first define the $d$-dimensional frequency estimation under the semi-continuous as follows. In this problem, we want to recover each frequencies in a small range.

**Problem B.4** (Frequency estimation). *Given a basis $\mathcal{B}$ of $m$ known vectors $b_1, b_2, \cdots b_m \in \mathbb{R}^d$, let $\Lambda(\mathcal{B}) \subset \mathbb{R}^d$ denote the lattice*

$$\Lambda(\mathcal{B}) = \left\{ z \in \mathbb{R}^d : z = \sum_{i=1}^m c_i b_i, c_i \in \mathbb{Z}, \forall i \in [m] \right\}$$

*Suppose that $f_1, f_2, \cdots, f_k \in \Lambda(\mathcal{B})$. Let $x^*(t) = \sum_{j=1}^k v_j e^{2\pi i \langle f_j, t \rangle}$, and let $g(t)$ denote the noise. Given observations of the form $x(t) = x^*(t) + g(t)$, $t \in [0, T]^d$. Let $\eta = \min_{i \ne j} \|f_j - f_i\|_\infty$.*

*The goal is to design an algorithm that output a set $L$ of frequencies such that, for each $f_i$, there is $f_i' \in L$, $\|f_i - f_i'\|_2 \le D/T$.*

We remark that the recovered frequencies in $L$ are not necessary to be in $\Lambda(\mathcal{B})$, and $D$ is a parameter that can depend on $k$.

Next, we define the $d$-dimensional Signal Estimation under the semi-continuous setting as follows. In this problem, we want to recover a signal that can approximate the ground-truth signal in the time domain.

**Problem B.5** (Signal Estimation problem). *Given a basis $\mathcal{B}$ of $m$ known vectors $b_1, b_2, \cdots b_m \in \mathbb{R}^d$, let $\Lambda(\mathcal{B}) \subset \mathbb{R}^d$ denote the lattice*

$$\Lambda(\mathcal{B}) = \left\{ z \in \mathbb{R}^d : z = \sum_{i=1}^m c_i b_i, c_i \in \mathbb{Z}, \forall i \in [m] \right\}$$

*Suppose that $f_1, f_2, \cdots, f_k \in \Lambda(\mathcal{B})$. Let $x^*(t) := \sum_{j=1}^k v_j e^{2\pi i \langle f_j, t \rangle}$, and let $g(t)$ denote the noise. Given observations of the form $x(t) = x^*(t) + g(t)$, $t \in [0, T]^d$. Let $\eta = \min_{i \ne j} \|f_j - f_i\|_\infty$.*

*The goal is to design an algorithm that outputs $y(t) = \sum_{j=1}^{\widetilde{k}} v_j' \cdot e^{2\pi i f_j' t}$ such that*

$$\int_{[0,T]^d} |y(t) - x(t)|^2 \mathrm{d}t \lesssim \int_{[0,T]^d} |g(t)|^2 \mathrm{d}t.$$

*Note that outputting $y(t) = \sum_{j=1}^{\widetilde{k}} v_j' \cdot e^{2\pi i f_j' t}$ means outputting $\{v_j', f_j'\}_{j \in [\widetilde{k}]}$.*

**Remark B.6.** *We note that given the solution of Frequency Estimation (Problem B.4), Signal Estimation (Problem B.5) can be formulated as a Fourier set query problem (Problem B.2). More specifically, by Frequency Estimation, we will find a set that contains all frequencies of the ground truth signal $x^*(t)$. Then, we only need to recover the coefficients with frequencies in this set, which is equivalent to a set query problem.*

## B.3 Facts about importance sampling

Important sampling try to estimate a statistic value in one distribution by taking samples in another distribution. In particular, Chen and Price (2019a) considered the importance sampling for estimating the norm of functions in a linear family $\mathcal{F}$.

In this followings, we first provide some basic definitions about linear function family.

**Definition B.7** (Condition number of sampling distribution). *Let $G$ be any domain and $\mathcal{F}$ is a linear function family from $G$ to $\mathbb{C}$. Let $D$ be an arbitrary distribution over $G$. Then the condition number of $D$ with respect to $\mathcal{F}$ is defined as follows:*

$$K_D := \sup_{t \in G} \sup_{f \in \mathcal{F}} \frac{|f(t)|^2}{\|f\|_D^2},$$

*where*

$$\|f\|_D^2 := \int_G D(t) \cdot |f(t)|^2 \mathrm{d}t.$$

**Definition B.8** (Orthonormal basis for linear function family). *Let $G$ be any domain. Given a linear function family $\mathcal{F}$ from $G$ to $C$, and a probability distribution $D$ over $G$. We say $\{v_1, \ldots, v_d\}$ form an orthonormal basis of $\mathcal{F}$ with respect to $D$, if they satisfy the following properties:*

- *for any $i, j \in [d]$, $\int_G D(t) v_i(t) \overline{v_j(t)} \mathrm{d}t = \mathbf{1}_{i=j}$, and*

- *for any $f \in \mathcal{F}$, $f \in \mathrm{span}\{v_1, \ldots, v_d\}$.*

**Fact B.9.** *Let $\{v_1, \ldots, v_k\}$ be an orthonormal basis of $\mathcal{F}$ with respect to $D$. For any function $f \in \mathcal{F}$, let $\alpha(f)$ denote the coefficients under the basis $\{v_1, \ldots, v_d\}$, i.e., $h = \sum_{i=1}^d \alpha(h)_i \cdot v_i$. Then,*
$$\|\alpha(h)\|_2 = \|h\|_D.$$

For an unknown function $f \in \mathcal{F}$, the goal of importance sampling is to estimate $\|f\|_D$, given samples from another distribution $D'$. The following definition introduces the importance sampling procedure and condition number of the importance sampling distribution.

**Definition B.10** (Definition 3.1 of Chen and Price (2019a)). *For any unknown distribution $D'$ over the domain $G$ and any function $f \in \mathcal{F}$, let $f^{(D')}(t) := \sqrt{\frac{D(t)}{D'(t)}} \cdot f(t)$ be the importance sampling function for some known distribution $D$ such that*

$$\mathop{\mathbb{E}}_{t \sim D'} \left[ |f^{(D')}(t)|^2 \right] = \mathop{\mathbb{E}}_{t \sim D'} \left[ \frac{D(t)}{D'(t)} |f(t)|^2 \right] = \mathop{\mathbb{E}}_{t \sim D} \left[ |f(t)|^2 \right].$$

*Then, we can use samples from $D'$ to estimate $\|f^{(D')}\|_{D'}$, which gives an estimate of $\|f\|_D$.*

*When the family $\mathcal{F}$ and $D$ is clear, we use $K_{\mathsf{IS},D'}$ to denote the condition number of importance sampling from $D'$:*

$$K_{\mathsf{IS},D'} = \sup_t \left\{ \sup_{f \in \mathcal{F}} \left\{ \frac{|f^{(D')}(t)|^2}{\|f^{D'}\|_{D'}^2} \right\} \right\} = \sup_t \left\{ \frac{D(t)}{D'(t)} \cdot \sup_{f \in \mathcal{F}} \left\{ \frac{|f(t)|^2}{\|f\|_D^2} \right\} \right\}. \tag{5}$$

From Definition B.10, we know that the efficiency of importance sampling depends on how many samples we need to estimate $\|f^{D'}\|_{D'}$. The following lemma provide a criteria for judging whether a set of samples gives a good estimation for the norm of function.

**Lemma B.11** (Lemma 4.2 in Chen and Price (2019a)). *For any $\varepsilon \in (0, 1)$, let $S = \{t_1, \ldots, t_s\}$ and the weight vector $w \in \mathbb{R}_{>0}^s$. Define a matrix $A \in \mathbb{R}^{s \times d}$ be the $s \times d$ matrix defined as $A_{i,j} = \sqrt{w_i} \cdot v_j(t_i)$, where $\{v_1, \ldots, v_d\}$ is an orthonormal basis for $\mathcal{F}$. Then*

$$\|h\|_{S,w}^2 := \sum_{j=1}^s w_j \cdot |h(x_j)|^2 \in [1 \pm \varepsilon] \cdot \|h\|_D^2 \quad \text{for every } h \in \mathcal{F}$$

*if and only if the eigenvalues of $A^* A$ are in $[1 - \varepsilon, 1 + \varepsilon]$.*

The following lemma shows that the sample complexity depends on the condition number $K_{\mathsf{IS},D'}$:

**Lemma B.12** (Lemma 6.6 in Chen and Price (2019a)). *Let $D'$ be an arbitrary distribution over $G$ and let $K_{\mathsf{IS},D'}$ be the condition number of importance sampling from $D'$ (defined by Eq. (5)). There exists an absolute constant $C$ such that for any $\varepsilon \in (0, 1)$ and $\delta \in (0, 1)$, let $S = \{t_1, \ldots, t_s\}$ be a set of i.i.d. samples from the distribution $D'$ and let $w$ be the weight vector defined by $w_j = \frac{D(t_j)}{s \cdot D'(t_j)}$ for each $j \in [s]$. Then, as long as*

$$s \geq \frac{C}{\varepsilon^2} \cdot K_{\mathsf{IS},D'} \log \frac{d}{\delta},$$

*the $s \times d$ matrix $A_{i,j} = \sqrt{w_i} \cdot v_j(t_i)$ satisfies*
$$\|A^* A - I\|_2 \leq \varepsilon \text{ with probability at least } 1 - \delta.$$

## C  Energy Bounds for Band-limited Signals

The energy bound shows that the maximum value of a band-limited signal in a certain interval can be bounded by its energy on the interval. One interesting fact is that the approximation ratio in the energy bound is only relate to the sparsity $k$, and have no relationship with time duration $T$ and band-limit $F$. An application of energy bound is preserving the norm, that is what is the least size of set $S$, such that $\|f\|_S = \|f\|_T$, for any function $f$ in a certain function family. The relationship between energy bound and norm preserving can be build by Chernoff bound.

Borwein and Erdélyi (2006); Kós (2008); Chen et al. (2016); Chen and Price (2019b) proved energy bounds for sparse Fourier signal under one-dimensional continuous Fourier transform. We further generalize these results to discrete band-limited signal under discrete Fourier transform and high-dimensional band-limited signal under continuous Fourier transform.

This section is organized as follows:

- Section C.1 reviews previous results for one-dimensional continuous Fourier-sparse signals.
- Section C.2 builds the connection between energy bound and the concentration property.

### C.1  Energy bound for one-dimensional signals

In this section, we review the energy bound proved in prior work Borwein and Erdélyi (2006); Kós (2008); Chen et al. (2016); Chen and Price (2019b).

Kós (2008) proved the following energy bound:

**Theorem C.1** (Kós (2008); Chen et al. (2016)). *Define a family of $F$-band-limit, $k$-sparse Fourier signals:*

$$\mathcal{F} := \left\{ x(t) = \sum_{j=1}^k v_j \cdot e^{2\pi \mathbf{i} f_j t} \,\Big|\, f_j \in \mathbb{R} \cap [-F, F] \right\}$$

*Then, for any $t \in (-1, 1)$,*

$$\sup_{x \in \mathcal{F}} \frac{|x(t)|^2}{\|x\|_D^2} \lesssim k^2.$$

Borwein and Erdélyi (2006) also proved a time-dependent energy bound for one-dimensional signal:

**Theorem C.2** (Borwein and Erdélyi (2006); Chen and Price (2019a)). *Define a family of $F$-band-limit, $k$-sparse Fourier signals:*

$$\mathcal{F} := \left\{ x(t) = \sum_{j=1}^k v_j \cdot e^{2\pi \mathbf{i} f_j t} \,\Big|\, f_j \in \mathbb{R} \cap [-F, F] \right\}$$

*Then, for any $t \in (-1, 1)$,*

$$\sup_{x \in \mathcal{F}} \frac{|x(t)|^2}{\|x\|_D^2} \lesssim \frac{k}{1 - |t|}.$$

### C.2  Energy bounds imply concentrations

By using Chernoff bound, we prove the following lemma to show the performance of uniformly sampling.

#### C.2.1  Continuous case

**Lemma C.3.** *Let $d \in \mathbb{Z}_+$. Let $R$ be a parameter. Given any function $x(t) : \mathbb{R}^d \to \mathbb{C}$ with $\max_{t \in [0,T]^d} |x(t)|^2 \le R\|x(t)\|_T^2$. Let $S$ denote a set of points chosen uniformly at random from $[0, T]^d$. We have that*

$$\Pr \left[ \left| \frac{1}{|S|} \sum_{i \in S} |x(t_i)|^2 - \|x(t)\|_T^2 \right| \ge \varepsilon \|x(t)\|_T^2 \right] \le \exp(-\Omega(\varepsilon^2 |S|/R)),$$

*where* $\|x(t)\|_T^2 = \frac{1}{T^d} \int_{[0,T]^d} |x(t)|^2 \mathrm{d}t.$

*Proof.* Let $M$ denote $\max_{t \in [0,T]^d} |x(t)|^2$. Replacing $X_i$ by $\frac{|x(t_i)|^2}{M}$ and $n$ by $|S|$ in Lemma A.1, we obtain that

$$\Pr[|X - \mu| > \varepsilon \mu] \leq 2 \exp(-\frac{\varepsilon^2}{3} \mu)$$

The above equation implies

$$\Pr\left[\left|\sum_{i \in S} \frac{|x(t_i)|^2}{M} - |S| \frac{\|x(t)\|_T^2}{M}\right| > \varepsilon |S| \frac{\|x(t)\|_T^2}{M}\right] \leq 2 \exp(-\frac{\varepsilon^2}{3} \mu)$$

Multiplying $M$ on the both sides

$$\Pr\left[\left|\frac{1}{|S|} \sum_{i \in S} |x(t_i)|^2 - \|x(t)\|_T^2\right| \geq \varepsilon \|x(t)\|_T^2\right] \leq 2 \exp(-\frac{\varepsilon^2}{3} \mu)$$

Applying bound on $\mu$

$$\Pr\left[\left|\frac{1}{|S|} \sum_{i \in S} |x(t_i)|^2 - \|x(t)\|_T^2\right| \geq \varepsilon \|x(t)\|_T^2\right] \leq 2 \exp(-\frac{\varepsilon^2}{3} |S| \frac{\|x(t)\|_T^2}{M})$$

which is less than $2 \exp(-\frac{\varepsilon^2}{3} |S|/R)$, thus completes the proof. $\qquad \square$

# D Uniform Sketching Band-Limited Signals

In this section, we show an intermediate step in the reduction from Frequency estimation to Signal estimation: constructing a small sketching subset $S$ of the time domain *obliviously* (without making any query to the signal), so that the signal discretized by $S$ has norm close to the original continuous signal. More formally, we define the *uniform sketching Fourier signal problem* as follows:

**Problem D.1** (Uniform sketching band-limited signal problem). *Suppose $f_1, f_2, \cdots, f_k \in \mathbb{R}^d$, and $v_1, \ldots, v_k \in \mathbb{C}$. Define the continuous signal $x(t) = \sum_{j=1}^{k} v_j e^{2\pi \mathbf{i} \langle f_j, t \rangle}$. Let $\eta = \min_{i \neq j} \|f_j - f_i\|_\infty$.*

*Let $\varepsilon \in (0, 0.1)$ denote the accuracy parameter. Find a set $S = \{t_1, \ldots, t_s\} \subseteq [0,T]^d$ of size $s$ such that*

$$(1 - \varepsilon)\|x\|_T \leq \|x\|_S \leq (1 + \varepsilon)\|x\|_T,$$

*where*

$$\|x\|_T^2 := \frac{1}{T^d} \int_{[0,T]^d} |x(t)|^2 \mathrm{d}t, \text{ and } \|x\|_S^2 := \frac{1}{|S|} \sum_{i \in [s]} |x(t_i)|^2.$$

In Section D.1, we show how to sketch one-dimensional signals with nearly-optimal weighted sketching.

## D.1 Weighted uniform sketching one-dimensional signals

For one-dimensional signals, the most natural approach to uniform sketching is to uniformly sample some points in the time domain. However, by a standard concentration argument, we know that the sample complexity is $\mathrm{poly}(k)$, which is not time-efficient for our task. In this section, we show a more efficient sketching method for one-dimensional band-limited signals by assigning different weights to each sample point. More precisely, let $S = \{t_1, \ldots, t_s\} \subseteq [0,T]$ be a discrete sketching set and let $w \in \mathbb{R}_{\geq 0}^s$ be the weight vector. We define the weighted sketching norm of the signal as follows:

$$\|x\|_{S,w} := \sum_{i \in [s]} w_i \cdot |x(t_i)|^2.$$

And the goal of weighted uniform sketching is to find a small set $S$ and a weight vector $w$ such that $\|x\|_{S,w} \approx \|x\|_T$.

In the following lemma, we give a sketch for any one-dimensional band-limited signal with nearly-optimal size:

**Lemma D.2** (Nearly-optimal weighted sketch for one-dimensional signals). *For $k \in \mathbb{N}_+$, define a probability distribution $D(t)$ as follows:*

$$D(t) := \begin{cases} c/(1 - |t/T|), & \text{for } |t| \leq T(1 - 1/k) \\ c \cdot k, & \text{for } |t| \in [T(1 - 1/k), T] \end{cases} \tag{6}$$

*where $c = \Theta(T^{-1} \log^{-1}(k))$ is a normalization factor such that $\int_{-T}^{T} D(t) \mathrm{d}t = 1$.*

*For any $f_1, \cdots, f_k \in [-F, F]$ and $v_1, \cdots, v_k \in \mathbb{C}$, let the continuous signal $x(t) = \sum_{j=1}^{k} v_j \exp(2\pi \mathbf{i} f_j t)$. For any $\varepsilon, \rho \in (0, 1)$, let $S_D = \{t_1, \cdots, t_s\}$ be a set of i.i.d. samples from $D(t)$ of size $s \geq O(\varepsilon^{-2} k \log(k) \log(1/\rho))$. Let the weight vector $w \in \mathbb{R}^s$ be defined by $w_i := 2/(TsD(t_i))$ for $i \in [s]$. Then with probability at least $1 - \rho$, we have*

$$(1 - \varepsilon)\|x\|_T \leq \|x\|_{S_D, w} \leq (1 + \varepsilon)\|x\|_T,$$

*where $\|x\|_T^2 := \frac{1}{2T} \int_{-T}^{T} |x(t)|^2 \mathrm{d}t$.*

*Proof.* For the convenient, in the proof, we use time duration $[-T, T]$. Let $\mathcal{F}$ be defined as:

$$\mathcal{F} := \left\{ x(t) = \sum_{j=1}^{k} v_j \cdot e^{2\pi \mathbf{i} f_j t} | f_j \in \mathbb{R} \cap [-F, F], v_j \in \mathbb{C} \right\}$$

Let $\{v_1(t), v_2(t), \cdots, v_k(t)\}$ be an orthonormal basis for $\mathcal{F}$ with respect to the distribution $D$, i.e.,

$$\int_0^T D(t) \cdot v_i(t)\overline{v_j(t)} \mathrm{d}t = \mathbf{1}_{i=j}, \quad \forall i, j \in [k].$$

We first prove that the distribution $D$ is well-defined. By the condition that $\int_{-T}^{T} D(t) \mathrm{d}t = 1$, we have

$$2 \int_0^{T(1-1/(k))} \frac{c}{(1 - |t/T|)} \mathrm{d}t + 2 \int_{T(1-1/(k))}^{T} c \cdot k^2 k \mathrm{d}t = 1,$$

which implies that

$$\begin{aligned} c^{-1} &= 2 \int_0^{T(1-1/k)} \frac{1}{(1 - |t/T|)} \mathrm{d}t + 2 \int_{T(1-1/k)}^{T} k^2 \mathrm{d}t \\ &= 2T \log k + 2T \\ &= \Theta(T \log(k)). \end{aligned}$$

Thus, we get that $c = \Theta(T^{-1} \log^{-1}(k))$.

To show that sampling from distribution $D$ give a good weighted sketch, we will use some technical tools in Section B.3. Applying Lemma B.12 with $D' = D$, $D = \mathrm{Uniform}([-T, T])$, $d = k$, $\delta = \rho$, we have that, with probability at least $1 - \rho$, the matrix $A \in \mathbb{C}^{s \times k}$ defined by $A_{i,j} := \sqrt{w_i} \cdot v_j(x_i)$ satisfying

$$\|A^* A - I\|_2 \leq \varepsilon,$$

as long as $s \geq \frac{C}{\varepsilon^2} \cdot K_{\mathsf{IS},D'} \log \frac{k}{\rho}$. Then, by Lemma B.11, it implies that for every $x \in \mathcal{F}$,

$$(1 - \varepsilon)\|x\|_T^2 \leq \|x\|_{S_D, w}^2 \leq (1 + \varepsilon)\|x\|_T^2.$$

It remains to bound the size of $S_D$; or equivalently, we need to upper-bound the condition number of the importance sampling of $D'$ (see Definition B.10):

$$K_{\mathsf{IS},D'} := \sup_t \left\{ \frac{D(t)}{D'(t)} \cdot \sup_{f \in \mathcal{F}} \left\{ \frac{|f(t)|^2}{\|f\|_D^2} \right\} \right\}$$

$$= \sup_t \{ \frac{1}{2TD(t)} \cdot \sup_{f \in \mathcal{F}} \{ \frac{|f(x)|^2}{\|f\|_D^2} \} \}$$

$$\leq \sup_t \{ \frac{1}{2TD(t)} \cdot \min\{ \frac{k}{1 - |t/T|}, k^2 \} \}$$

$$\leq \max\{ \frac{(1 - |t/T|)}{2cT} \frac{k}{1 - |t/T|}, \frac{1}{2cTk} k^2 \}$$

$$= \frac{k}{2cT}$$

$$= O(k \log k),$$

where the first step follows from the definition, the second step follows from $D(t) = \text{Uniform}([-T,T])(t) = \frac{1}{2T}$, the third step follows from Theorem C.1 and Theorem C.2, and the remaining steps follow from direct calculations. Thus, we get that

$$|S_D| \geq \Omega \left( \varepsilon^{-2} k \log(k) \log(1/\rho) \right).$$

The lemma is then proved. $\qquad \square$

### D.2 $\varepsilon$-net for sparse band-limited signals

In this section, we construct $\varepsilon$-nets for high-dimensional sparse Fourier continuous and discrete signals.

**Lemma D.3** ($\varepsilon$-net construction for continuous signals). *Given $k \in \mathbb{Z}_+$ unknown frequencies $f_1, f_2, \ldots, f_k \in [-F, F]^d$. Let $V := \{ e^{2\pi \mathbf{i} \langle f_i, t \rangle} \mid i \in [k] \}$ be a family of Fourier basis. Let $\mathcal{Q} := \{ u \in \text{span}\{V\} \mid \|u\|_T^2 = 1 \}$ be the set of all signals in $[0,T]^d$ with frequency $f_1, \ldots, f_k$, where $\|x\|_T^2 = \frac{1}{T^d} \int_{[0,T]^d} |x(t)|^2 dt$.*

*Then, there exists an $\varepsilon$-net $\mathcal{P}_d \subset \mathcal{Q}$ such that*

1. *$\forall u \in \mathcal{Q}, \exists w \in \mathcal{P}_d, \|u - w\|_T \leq \varepsilon$.*

2. *$|\mathcal{P}_d| \leq \left( 5 \frac{k}{\varepsilon} \right)^{2k}$.*

*Proof.* We first construct an $\frac{\varepsilon}{k}$-net for the unit disk in $\mathbb{C}$, i.e., $\{ z \in \mathbb{C} \mid |z| \leq 1 \}$. Let $\mathcal{P}'$ denote

$$\mathcal{P}' := \left\{ \frac{\varepsilon}{2k} j_1 + \mathbf{i} \frac{\varepsilon}{2k} j_2 \mid j_1, j_2 \in \mathbb{Z}, |j_1| \leq \frac{2k}{\varepsilon}, |j_2| \leq \frac{2k}{\varepsilon} \right\}.$$

Notice that $|\varepsilon/(2k) j_1| \leq \varepsilon/(2k) \cdot 2k/\varepsilon = 1$; and similarly, $|\varepsilon/(2k) j_2| \leq 1$. Thus, for any $a \in \mathbb{C}$, $|a| \leq 1$, there is a $b \in \mathcal{P}'$ such that

$$|a - b| \leq \varepsilon/(2k) + \varepsilon/(2k) \leq \varepsilon/k.$$

Moreover,

$$|\mathcal{P}'| \leq (2 \cdot 2k/\varepsilon + 1) \cdot (2 \cdot 2k/\varepsilon + 1) = \left( 4 \frac{k}{\varepsilon} + 1 \right)^2.$$

Hence, we conclude that,

- $\mathcal{P}'$ is an $\frac{\varepsilon}{k}$-net in the unit circle of $\mathbb{C}$.

- $\mathcal{P}'$ has size at most $(4 \frac{k}{\varepsilon} + 1)^2$.

Then, we use $\mathcal{P}'$ to construct an $\varepsilon$-net for $\mathcal{Q}$. Since the dimension of $\mathcal{Q}$ is at most $k$, we take an orthonormal basis $w_1, \cdots, w_k \in \mathcal{Q}$ such that,

$$\int_{[0,T]^d} w_i(t) \overline{w_j(t)} dt = \mathbf{1}_{i=j}.$$

And we define
$$\mathcal{P}'' := \{\sum_{i=1}^{k} \alpha_i w_i \mid \forall i \in [k], \alpha_i \in \mathcal{P}'\}.$$

First, for any $u \in \mathcal{Q}$, we have
$$u = \sum_{i=1}^{k} v_i \exp(2\pi \mathbf{i}\langle f_i, t\rangle) = \sum_{i=1}^{k} \alpha_i' w_i,$$

which implies that $|\alpha_i'| \leq 1$ for all $i \in [k]$. So, for any $a \in \mathcal{Q}$, there is a $b \in \mathcal{P}''$ such that $\|a - b\|_T \leq k \cdot \varepsilon/k = \varepsilon$. Moreover, $|\mathcal{P}''| \leq ((4\frac{k}{\varepsilon} + 1)^2)^k \leq (5\frac{k}{\varepsilon})^{2k}$. Therefore, we conclude that $\mathcal{P}''$ is an $\varepsilon$-net for $\mathcal{Q}$ and $|\mathcal{P}''| \leq \left(5\frac{k}{\varepsilon}\right)^{2k}$.

Then we define
$$\mathcal{P}_d := \{v \in \mathcal{Q} \mid \forall u \in \mathcal{P}'', v = \mathrm{argmin}_{v \in \mathcal{Q}}\{\|v - u\|_T\}\}.$$
therefore we have that, for any $a \in \mathcal{Q}$, there is a $b \in \mathcal{P}''$ such that $\|a - b\|_T \leq \varepsilon$, because there is a $c \in \mathcal{P}_d$, such that $\|c - b\|_T = \min_{d \in \mathcal{Q}}\|d - b\|_T \leq \|a - b\|_T \leq \varepsilon$. Then, $\|c - a\|_T \leq \|c - b\|_T + \|b - a\|_T \leq 2\varepsilon$. □

# E    Fast Implementation of Well-Balanced Sampling Procedure

Well-balanced sampling procedure was first defined in Chen and Price (2019a) to study the active linear regression problem. Our signal estimation algorithm will call it as a sub-procedure. In this section, we give a fast implementation of well-balanced sampling procedure based on the Randomized BSS algorithm Batson et al. (2012); Lee and Sun (2015).

First, we restate the definition of well-balanced sampling procedure in Chen and Price (2019a).

**Definition E.1** (Well-balanced sampling procedure (WBSP), Chen and Price (2019a))**.** *Given a linear family $\mathcal{F}$ and underlying distribution $D$, let $P$ be a random sampling procedure that terminates in $m$ iterations ($m$ is not necessarily fixed) and provides a coefficient $\alpha_i$ and a distribution $D_i$ to sample $x_i \sim D_i$ in every iteration $i \in [m]$.*

*We say $P$ is an $\varepsilon$-WBSP if it satisfies the following two properties:*

1. *With probability $0.9$, for weight $w_i = \alpha_i \cdot \frac{D(x_i)}{D_i(x_i)}$ of each $i \in [m]$,*
$$\sum_{i=1}^{m} w_i \cdot |h(x_i)|^2 \in \left[1 - 10\sqrt{\varepsilon}, 1 + 10\sqrt{\varepsilon}\right] \cdot \|h\|_D^2 \quad \forall h \in \mathcal{F}.$$

2. *The coefficients always have $\sum_{i=1}^{m} \alpha_i \leq \frac{5}{4}$ and $\alpha_i \cdot K_{\mathsf{IS},D_i} \leq \frac{\varepsilon}{2}$ for all $i \in [m]$.*

This definition describes a general sampling procedure that uses a few samples to represent the whole continuous signal, and the sampling procedure should satisfy two properties: one guarantees that the norm of any function in a function family is preserved, and another guarantees that the norm of noise is also preserved.

In Section E.1, we review some results in Chen and Price (2019a) and show that WBSP can be implemented via randomized spectral sparsification. In Section E.2, we design a data structure and improve the time efficiency of the WBSP. In Section E.3, we discover a tradeoff between the preprocessing cost and the query cost, which can improve the space complexity.

## E.1    Randomized BSS implies a WBSP

In this section, we review the result of Chen and Price (2019a), which shows that the Randomized BSS algorithm Batson et al. (2012); Lee and Sun (2015) implies a well-balanced sampling procedure.

**Lemma E.2** (Lemma 5.1 in Chen and Price (2019a))**.** *Let $G$ be any domain. Given any dimension $d$ linear function family $\mathcal{F}$ of function $f : G \to \mathbb{C}$,*
$$\mathcal{F} = \{f(t) = \sum_{j=1}^{d} v_j u_j(t) | v_j \in \mathbb{C}\},$$

where $u_j : G \to \mathbb{C}$. *Given any distribution $D$ over $G$, and any $\varepsilon > 0$, there exists an efficient procedure (Algorithm 2) that runs in $O(\varepsilon^{-1} d^3 |G| + \varepsilon^{-1} d^{\omega+1})$ time and outputs a set $S \subseteq G$ and weight $w$ such that*

- $|S| = O(d/\varepsilon)$, $w \in \mathbb{R}^{|S|}$,
- *the procedure is an $\varepsilon$-WBSP,*

*holds with probability $1 - \frac{1}{200}$.*

---

**Algorithm 2** A well-balanced sampling procedure based on Randomized BSS (see Chen and Price (2019a))

---

1: **procedure** RANDBSS$(d, \mathcal{F}, D, \varepsilon)$
2:     Find an orthonormal basis $v_1, \ldots, v_d$ of $\mathcal{F}$ under $D$
3:     Set $\gamma \leftarrow \sqrt{\varepsilon}/3$ and $\mathsf{mid} \leftarrow \frac{4d/\gamma}{1/(1-\gamma) - 1/(1+\gamma)}$
4:     $j \leftarrow 0, B_0 \leftarrow 0$
5:     $l_0 \leftarrow -2d/\gamma, u_0 \leftarrow 2d/\gamma$
6:     **while** $u_{j+1} - l_{j+1} < 8d/\gamma$ **do**
7:         $\Phi_j \leftarrow \text{tr}[(u_j I - B_j)^{-1}] + \text{tr}[(B_j - l_j I)^{-1}]$        ▷ The potential function at iteration $j$.
8:         Set the coefficient $\alpha_j \leftarrow \frac{\gamma}{\Phi_j} \cdot \frac{1}{\mathsf{mid}}$
9:         Set $v(x) \leftarrow \big(v_1(x), \ldots, v_d(x)\big)$
10:        **for** $x \in \text{supp}(D)$ **do**
11:            Set the distribution

$$D_j(x) \leftarrow D(x) \cdot \left( v(x)^\top (u_j I - B_j)^{-1} v(x) + v(x)^\top (B_j - l_j I)^{-1} v(x) \right) / \Phi_j$$

12:        **end for**
13:        Sample $x_j \sim D_j$ and set a scale $s_j \leftarrow \frac{\gamma}{\Phi_j} \cdot \frac{D(x_j)}{D_j(x_j)}$
14:        $B_{j+1} \leftarrow B_j + s_j \cdot v(x_j) v(x_j)^\top$
15:        $u_{j+1} \leftarrow u_j + \frac{\gamma}{\Phi_j(1-\gamma)}, \quad l_{j+1} \leftarrow l_j + \frac{\gamma}{\Phi_j(1+\gamma)}$
16:        $j \leftarrow j + 1$
17:    **end while**
18:    $m \leftarrow j$
19:    Assign the weight $w_j \leftarrow s_j/\mathsf{mid}$ for each $x_j$
20:    **return** $\{x_1, x_2, \cdots, x_m\}, w$
21: **end procedure**

---

### E.2   Fast implementation of WBSP

In this section, we give a fast implementation of Algorithm 2:

**Theorem E.3** (Fast implementation of WBSP). *Let $G$ be any domain. Given any dimension $d$ linear function family $\mathcal{F}$ of function $f : G \to \mathbb{C}$,*

$$\mathcal{F} = \{ f(t) = \sum_{j=1}^{d} v_j u_j(t) | v_j \in \mathbb{C} \},$$

*where $u_j : G \to \mathbb{C}$. Given any distribution $D$ over $G$, and any $\varepsilon > 0$, there exists an efficient procedure (Algorithm 3) that runs in $O(d^2 |G| + \varepsilon^{-1} d^3 \log |G| + \varepsilon^{-1} d^{\omega+1})$ time and outputs a set $S \subseteq G$ and weight $w \in \mathbb{R}^{|S|}$ such that the following properties hold with probability at least 0.995:*

- $|S| = O(d/\varepsilon)$,
- *the procedure is an $\varepsilon$-WBSP.*

Our algorithm is based on a data structure for solving the *online quadratic-form sampling problem* defined as follows:

---

**Algorithm 3** Our fast implementation of well-balanced sampling procedure

---

1: **procedure** RANDBSS+$(d, \mathcal{F}, D, \varepsilon)$          ▷ Theorem E.3
2:     /*Preprocessing*/
3:     Find an orthonormal basis $v_1, \ldots, v_d$ of $\mathcal{F}$ under $D$
4:     $\gamma \leftarrow \sqrt{\varepsilon}/3$ and $\mathsf{mid} \leftarrow \frac{4d/\gamma}{1/(1-\gamma) - 1/(1+\gamma)}$
5:     $j \leftarrow 0, B_0 \leftarrow 0$
6:     $l_0 \leftarrow -2d/\gamma, u_0 \leftarrow 2d/\gamma$
7:     $\delta \leftarrow 1/\mathrm{poly}(d)$
8:                                    ▷ Let $v(x) = \big(v_1(x), \ldots, v_d(x)\big) \in \mathbb{R}^d$
9:     DS.INIT$(|D|, d, \{v(x_1), \cdots, v(x_{|D|})\} \subset \mathbb{R}^d, \{D(x_1), \ldots, D(x_{|D|})\} \subset \mathbb{R})$    ▷ Algorithm 4
10:     /*Iterative step*/
11:     **while** $u_{j+1} - l_{j+1} < 8d/\gamma$ **do**
12:        $\Phi_j \leftarrow \mathrm{tr}[(u_j I - B_j)^{-1}] + \mathrm{tr}[(B_j - l_j I)^{-1}]$      ▷ The potential function at iteration $j$.
13:        $\alpha_j \leftarrow \frac{\gamma}{\Phi_j} \cdot \frac{1}{\mathsf{mid}}$
14:        $E_j \leftarrow (u_j I - B_j)^{-1} + (B_j - l_j I)^{-1}$
15:        $q \leftarrow$ DS.QUERY$(E_j/\Phi_j)$                ▷ $q \in [|D|]$, Algorithm 4
16:        $\mathsf{x}_j \leftarrow x_q$ and set a scale $s_j \leftarrow \frac{\gamma}{v(\mathsf{x}_j)^\top E_j v(\mathsf{x}_j)}$
17:        $B_{j+1} \leftarrow B_j + s_j \cdot v(\mathsf{x}_j) v(\mathsf{x}_j)^\top$
18:        $u_{j+1} \leftarrow u_j + \frac{\gamma}{\Phi_j(1-\gamma)}, \quad l_{j+1} \leftarrow l_j + \frac{\gamma}{\Phi_j(1+\gamma)}$
19:        $j \leftarrow j + 1$
20:     **end while**
21:     $m \leftarrow j$
22:     Assign the weight $w_j \leftarrow s_j/\mathsf{mid}$ for each $x_j$
23:     **return** $\{\mathsf{x}_1, \mathsf{x}_2, \cdots, \mathsf{x}_m\}, w$
24: **end procedure**

---

**Problem E.4** (Online Quadratic-Form Sampling Problem). *Given $n$ vectors $v_1, \ldots, v_n \in \mathbb{R}^d$ and $n$ coefficients $\alpha_1, \ldots, \alpha_n$, for any PSD matrix $A \in \mathbb{R}^{d \times d}$, output a sample $i \in [n]$ from the following distribution $\mathcal{D}_A$:*

$$\Pr_{\mathcal{D}_A}[i] := \frac{\alpha_i \cdot v_i^\top A v_i}{\sum_{j=1}^n \alpha_j \cdot v_j^\top A v_j} \quad \forall i \in [n]. \tag{7}$$

**Theorem E.5.** *There is a data structure (Algorithm 4) that uses $O(nd^2)$ spaces for the Online Quadratic-Form Sampling Problem with the following procedures:*

- INIT$(n, d, \{v_1, \ldots, v_n\} \subset \mathbb{R}^d, \{\alpha_1, \ldots, \alpha_n\} \subset \mathbb{R})$: *the data structure preprocesses in time $O(nd^2)$.*

- QUERY$(A \in \mathbb{R}^{d \times d})$: *Given a PSD matrix $A$, the QUERY operation samples $i \in [n]$ exactly from the probability distribution $\mathcal{D}_A$ defined in Problem E.4 in $O(d^2 \log n)$-time.*

*Proof.* The pseudo-code of the algorithm is given as Algorithm 4. The idea is to build a binary tree such that each node has an interval in $[l, \ldots, r] \subset [1, \ldots, n]$ and stores a matrix $\sum_{i=l}^r \alpha_i \cdot v_i v_i^\top$. For each internal node with interval $[l, \ldots, r]$, its left child node has interval $[l, \ldots, \lfloor (l+r)/2 \rfloor]$, and its right child node has interval $[\lfloor (l+r)/2 \rfloor + 1, \ldots, r]$.

We first prove the correctness. Suppose the output of QUERY is $i \in [n]$. We compute its probability. Let $u_0 = \mathrm{root}, u_1, \ldots, u_t$ be the path from the root of the tree to the leaf with $\mathrm{id} = i$. Then, we have

$$\Pr[u_t] = \prod_{j=1}^t \Pr[u_j | u_{j-1}] = \prod_{j=1}^t \frac{\sum_{k=l_j}^{r_j} \alpha_k \cdot v_k^\top A v_k}{\sum_{k=l_{j-1}}^{r_{j-1}} \alpha_k \cdot v_k^\top A v_k} = \frac{\alpha_i \cdot v_i^\top A v_i}{\sum_{k=1}^n \alpha_k \cdot v_k^\top A v_k},$$

where $[l_j, \ldots, r_j]$ is the range of the node $u_j$, the first step follows from the conditional probability, the second step follows from Line 34 in Algorithm 4, and the last step follows from the telescoping products. Hence, we get that

$$\Pr[\text{QUERY}(A) = i] = \Pr_{\mathcal{D}_A}[i] \quad \forall i \in [n].$$

Hence, the sampling distribution is the same as the Online Quadratic-Form Sampling Problem's distribution.

For the running time, in the preprocessing stage, we build the binary tree recursively. It is easy to see that the number of nodes in the tree is $O(n)$ and the depth is $O(\log n)$. For a leaf node, we take $O(d^2)$-time to compute the matrix $\alpha_i \cdot v_i v_i^\top \in \mathbb{R}^{d \times d}$. For an internal node, we take $O(d^2)$-time to add up the matrices of its left and right children. Thus, the total preprocessing time is $O(nd^2)$.

In the query stage, we walk along a path from the root to a leaf, which has $O(\log n)$ steps. In each step, we compute the inner product between $A$ and the current node's matrix, which takes $O(d^2)$-time. And we compute the inner product between $A$ and its left child node's matrix, which also takes $O(d^2)$-time. Then, we toss a coin and decide which subtree to move. Hence, each query takes $O(d^2 \log n)$-time.

The theorem is then proved. $\qquad\square$

**Lemma E.6** (Running time of Procedure RANDBSS+ in Algorithm 3). *Algorithm 3 runs in*

- $O(|D|d^2)$*-time for preprocessing,*
- $O(d^2 \log(|D|) + d^\omega)$*-time per iteration, and*
- $O(\varepsilon^{-1} d)$ *iterations.*

*Thus, the total running time is,*

$$O(|D|d^2 + \varepsilon^{-1} d \cdot (d^2 \log |D| + d^\omega)).$$

*Proof.* In each call of the Procedure RANDBSS+ in Algorithm 3,

- Finding orthonormal basis takes $O(|D|d^2)$.
- In the line 9, it runs $O(|D|d^2)$ times.
- The while loop repeat $O(\varepsilon^{-1} d)$ times.
  - Line 14 is computing $(u_j I - B_j) \in \mathbb{C}^{d \times d}$, $(u_j I - B_j)^{-1}$. This part takes $O(d^\omega)$ time[6].
  - Note that line 15 of Procedure RANDBSS+ in Algorithm 3 runs $O(d^2 \log |D|)$ times.

So, the time complexity of Procedure RANDBSS+ in Algorithm 3 is

$$O(|D|d^2 + \varepsilon^{-1} d \cdot (d^2 \log |D| + d^\omega)).$$

$\qquad\square$

**Lemma E.7** (Correctness of Procedure RANDBSS+ in Algorithm 3). *Given any dimension $d$ linear space $\mathcal{F}$, any distribution $D$ over the domain of $\mathcal{F}$, and any $\varepsilon > 0$, RANDBSS+$(d, \mathcal{F}, D, \varepsilon)$ is an $\varepsilon$-WBSP that terminates in $O(d/\varepsilon)$ rounds with probability $1 - 1/200$.*

*Proof.* We first claim that, for each $j \in [m]$, $\mathsf{x}_j$ has the same distribution as $D_j$, where

$$D_j(x) = D(x) \cdot (v(x)^\top E_j v(x))/\Phi_j \quad \forall x \in D$$

Notice that sampling from distribution $D_j$ can be reformulated as an Online Quadratic-Form Sampling Problem: the vectors are $\{v(x)\}_{x \in D}$, the coefficients are $\{D(x)\}_{x \in D}$, and the query matrix is $E_j' := E_j/\Phi_j$. Then, we have $D_j = \mathcal{D}_{E_j'}$ defined in Problem E.4. Hence, by Theorem E.5, we can use the data structure (Algorithm 4) to efficiently sample from $D_j$.

Therefore, the sample $\mathsf{x}_j$ in each iteration is generated from the same distribution as the original randomized BSS algorithm (Algorithm 2). Then, the WBSP guarantee and the number of iterations immediately follow from the proof of (Chen and Price, 2019a, Lemma 5.1).

The proof of the lemma is then completed. $\qquad\square$

---

[6] Note that this step seems to be very difficult to speed up via the Sherman-Morrison formula since $u_j$ changes in each iteration and the update is of high rank.

---

**Algorithm 4** Quadratic-form sampling data structure

---

```
 1: structure Node
 2:     V ∈ ℝ^{d×d}
 3:     left, right                                    ▷ Point to the left/right child in the tree
 4: end structure
 5: data structure DS
 6: members
 7:     n ∈ ℕ                                          ▷ The number of vectors
 8:     v_1, …, v_n ∈ ℝ^d                              ▷ d-dimensional vectors
 9:     α_1, …, α_n ∈ ℝ                                ▷ Coefficients
10:     root: Node                                     ▷ The root of the tree
11: end members
12: procedure BUILDTREE(l, r)                          ▷ [l, …, r] is the range of the current node
13:     p ← new Node
14:     if l = r then                                  ▷ Leaf node
15:         p.V ← α_l · v_l v_l^⊤                       ▷ It takes O(d²)-time
16:     else                                           ▷ Internal node
17:         mid ← ⌊(l + r)/2⌋
18:         p.left ← BUILDTREE(l, mid)
19:         p.right ← BUILDTREE(mid + 1, r)
20:         p.V ← (p.left).V + (p.right).V             ▷ It takes O(d²)-time
21:     end if
22:     return p
23: end procedure
24: procedure INIT(n, d, {v_i}_{i∈[n]} ⊆ ℝ^d, {α_i}_{i∈[n]} ⊆ ℝ)
25:     v_i ← v_i, α_i ← α_i for i ∈ [n]
26:     root ← BUILDTREE(1, n)
27: end procedure
28: procedure QUERY(A ∈ ℝ^{d×d})
29:     p ← root, l ← 1, r ← n
30:     s ← 0
31:     while l ≠ r do                                 ▷ There are O(log n) iterations
32:         w ← ⟨p.V, A⟩                                ▷ It takes O(d²)-time
33:         w_ℓ ← ⟨(p.left).V, A⟩
34:         Sample c from Bernoulli(w_ℓ/w)
35:         if c = 0 then
36:             p ← p.left, r ← ⌊(l + r)/2⌋
37:         else
38:             p ← p.right, l ← ⌊(l + r)/2⌋ + 1
39:         end if
40:     end while
41:     return l
42: end procedure
43: end data structure
```

---

*Proof of Theorem E.3.* The running time of the algorithm follows from Lemma E.6, and the correctness follows from Lemma E.7. □

### E.3 Trade-off between preprocessing and query

In this section, we consider the preprocessing and query trade-off in the data structure for quadratic form sampling problem. In the following theorem, we give a new data structure that takes less time in preprocessing and more time for each query than Theorem E.5, and the space complexity is also reduced from $O(nd^2)$ to $O(nd)$.

**Theorem E.8.** *There is a data structure (Algorithms 5 and 6) that uses $O(nd)$ spaces for the Online Quadratic-Form Sampling Problem with the following procedures:*

- INIT$(n, d, \{v_1, \ldots, v_n\} \subset \mathbb{R}^d, \{\alpha_1, \ldots, \alpha_n\} \subset \mathbb{R})$: *the data structure preprocesses in time* $O(nd^{\omega-1})$.

- QUERY$(A \in \mathbb{R}^{d \times d})$: *Given a PSD matrix $A$, the* QUERY *operation samples $i \in [n]$ exactly from the probability distribution $\mathcal{D}_A$ defined in Problem E.4 in $O(d^2 \log(n/d) + d^{\omega})$-time.*

*Proof.* The time and space complexities follow from Lemma E.9. And the correctness follows from Lemma E.10. □

---

**Algorithm 5** Quadratic-form sampling with preprocessing-query trade-off: Preprocessing

---

1: **structure** Node
2:    $V_1, V_2 \in \mathbb{R}^{d \times d}$
3:    left, right                                      ▷ Point to the left/right child in the tree
4: **end structure**
5: **data structure** DS+                                             ▷ Theorem E.8
6: **members**
7:    $n \in \mathbb{N}$                                          ▷ The number of vectors
8:    $m \in \mathbb{N}$                                          ▷ The number of blocks
9:    $v_1, \ldots, v_n \in \mathbb{R}^d$                           ▷ $d$-dimensional vectors
10:    root: Node                                      ▷ The root of the tree
11: **end members**
12: **procedure** BUILDTREE$(l, r)$              ▷ $[l, \ldots, r]$ is the range of the current node
13:      p $\leftarrow$ **new** Node
14:      **if** $l = r$ **then**                                    ▷ Leaf node
15:          p.$V_2 \leftarrow \begin{bmatrix} v_{(l-1)d+1} & \cdots & v_{ld} \end{bmatrix}$
16:          p.$V_1 \leftarrow$ (p.$V_2$) $\cdot$ (p.$V_2$)$^{\top}$              ▷ It takes $O(d^{\omega})$-time
17:                                   ▷ p.mat1 $= \sum_{i=(l-1)d+1}^{ld} v_i v_i^{\top}$
18:      **else**                                          ▷ Internal node
19:          $mid \leftarrow \lfloor (l+r)/2 \rfloor$
20:          p.left $\leftarrow$ BUILDTREE$(l, mid)$
21:          p.right $\leftarrow$ BUILDTREE$(mid+1, r)$
22:          p.$V_1 \leftarrow$ (p.left).$V_1$ + (p.right).$V_1$             ▷ It takes $O(d^2)$-time
23:      **end if**
24:      **return** p
25: **end procedure**
26: **procedure** INIT$(n, d, \{v_i\}_{i \in [n]} \subseteq \mathbb{R}^d, \{\alpha_i\}_{i \in [n]} \subseteq \mathbb{R})$
27:      $v_i \leftarrow v_i \cdot \sqrt{\alpha_i}$ for $i \in [n]$
28:      $m \leftarrow n/d$                            ▷ We assume that $n$ is divisible by $d$
29:      Group $\{v_i\}_{i \in [n]}$ into $m$ blocks $B_1, \ldots, B_m$      ▷ $B_i = \{v_{(i-1)d+1}, \ldots, v_{id}\}$ for $i \in [m]$
30:      root $\leftarrow$ BUILDTREE$(1, m)$
31: **end procedure**
32: **end data structure**

---

**Lemma E.9** (Time and space complexities of Algorithms 5 and 6). *The* INIT *procedure takes* $O(nd^{\omega-1})$-*time. The* QUERY *procedure takes* $O(d^2 \log(n/d) + d^{\omega})$-*time. The data structure uses* $O(nd)$-*space.*

*Proof.* We prove the space and time complexities of the data structure as follows:
**Space complexity:** Let $m = n/d$. It is easy to see that there are $O(m)$ nodes in the data structure. And each node has two $d$-by-$d$ matrices. Hence, the total space used by the data structure is $O(n/d) \cdot O(d^2) = O(nd)$.

**Time complexity:** In the preprocessing stage, the time-consuming step is the call of BUILDTREE. There are $O(m)$ internal nodes and $O(m)$ leaf nodes. Each internal node takes $O(d^2)$-time to construct the matrix $V_1$ (Line 22). For each leaf node, it takes $O(d^2)$-time to form the matrix $V_2$ (Line 15). And it takes $O(d^{\omega})$-time to compute the matrix $V_1$ (Line 16). Hence, the total running time of BUILDTREE is $O(md^{\omega}) = O(nd^{\omega-1})$.

**Algorithm 6** Quadratic-form sampling with preprocessing-query trade-off: Query

```
 1: data structure DS+                                                    ▷ Theorem E.8
 2: members
 3:    n ∈ ℕ                                                    ▷ The number of vectors
 4:    m ∈ ℕ                                                     ▷ The number of blocks
 5:    v₁, …, vₙ ∈ ℝᵈ                                            ▷ d-dimensional vectors
 6:    root: Node                                                    ▷ The root of the tree
 7: end members
 8: procedure BLOCKSAMPLING(p, l ∈ ℕ, A ∈ ℝᵈˣᵈ)        ▷ p is a leaf node with index l
 9:    U ← (p.V₂)ᵀ · A · (p.V₂)                                      ▷ It takes O(dω)-time
10:    Define a distribution 𝒟ₗ over [d] such that 𝒟ₗ(i) ∝ Uᵢ,ᵢ
11:    Sample i ∈ [d] from 𝒟ₗ                                        ▷ It takes O(d)-time
12:    return (l − 1)d + i
13: end procedure
14: procedure QUERY(A ∈ ℝᵈˣᵈ)
15:    p ← root, l ← 1, r ← m
16:    s ← 0
17:    while l ≠ r do                                      ▷ There are O(log m) iterations
18:       w ← ⟨p.V₁, A⟩                                         ▷ It takes O(d²)-time
19:       wₗ ← ⟨(p.left).V₁, A⟩
20:       Sample c from Bernoulli(wₗ/w)
21:       if c = 0 then
22:          p ← p.left, r ← ⌊(l + r)/2⌋
23:       else
24:          p ← p.right, l ← ⌊(l + r)/2⌋ + 1
25:       end if
26:    end while
27:    return BLOCKSAMPLING(p, l, A)
28: end procedure
29: end data structure
```

In the query stage, the While loop in the QUERY procedure (Line 17) is the same as in Algorithm 4. Since there are $O(m)$ nodes in the tree, it takes $O(d^2 \log m)$-time. Then, in the BLOCKSAMPLING procedure, it takes $O(d^\omega)$-time to compute the matrix $U$ (Line 9), and it takes $O(d)$-time to sample an index from the distribution $\mathcal{D}_l$ (Line 11). Hence, the total running time for each query is $O(d^2 \log m + d^\omega) = O(d^2 \log(n/d) + d^\omega)$.

The proof of the lemma is then completed. □

**Lemma E.10** (Correctness of Algorithm 6). *The distribution of the output of the QUERY(A) is $\mathcal{D}_A$ defined by Eq. (7).*

*Proof.* For simplicity, we assume that all the coefficients $\alpha_i = 1$.

Let $u_0 = \mathsf{root}, u_1, \ldots, u_t$ be the path in the While loop (Line 17) from the root of the tree to the leaf with index $l \in [m]$. By the construction of leaf node, we have

$$V_1 = V_2 V_2^\top = \begin{bmatrix} v_{(l-1)d+1} & \cdots & v_{ld} \end{bmatrix} \begin{bmatrix} v_{(l-1)d+1}^\top \\ \vdots \\ v_{ld}^\top \end{bmatrix} = \sum_{i=(l-1)d+1}^{ld} v_i v_i^\top,$$

which is the same as the $V$-matrix in Algorithm 4. Hence, similar to the proof of Theorem E.5, we have

$$\Pr[u_t] = \prod_{j=1}^{t} \Pr[u_j | u_{j-1}] = \frac{\sum_{i=(l-1)d+1}^{ld} v_i^\top A v_i}{\sum_{i=1}^{n} v_i^\top A v_i}.$$

where $\{(l-1)d + 1, \ldots, ld\}$ is the range of the node $u_t$ and $\{1, \ldots, n\}$ is the range of $u_0$.

Then, consider the BLOCKSAMPLING procedure. Let $\{v_1, \ldots, v_d\}$ be the vectors in the input block. At Line 9, we have

$$U = V_2^\top A V_2 = \begin{bmatrix} v_1^\top \\ \vdots \\ v_d^\top \end{bmatrix} A \begin{bmatrix} v_1 & \cdots & v_d \end{bmatrix}.$$

For $i \in [d]$, the $i$-th element in the diagonal of $U$ is

$$U_{i,i} = v_i^\top A v_i.$$

Hence,

$$\Pr[\text{BLOCKSAMPLING} = i] = \frac{v_i^\top A v_i}{\sum_{j=1}^d v_j^\top A v_j}.$$

Therefore, for any $k \in [n]$, if $k = (l-1)d + r$ for some $l, r \in \mathbb{N}$, then the sample probability is

$$\Pr[\text{QUERY}(A) = k] = \Pr[\text{BLOCKSAMPLING} = k \mid u_t = \text{Block } l] \cdot \Pr[u_t = \text{Block } l]$$

$$= \frac{v_k^\top A v_k}{\sum_{i=(l-1)d+1}^{ld} v_i^\top A v_i} \cdot \frac{\sum_{i=(l-1)d+1}^{ld} v_i^\top A v_i}{\sum_{i=1}^n v_i^\top A v_i}$$

$$= \frac{v_k^\top A v_k}{\sum_{i=1}^n v_i^\top A v_i}$$

$$= \mathcal{D}_A(k).$$

The lemma is then proved. $\qquad\square$

As a corollary, we get a WBSP using less space:

**Corollary E.11** (Space efficient implementation of WBSP). *By plugging-in the new data structure (Algorithms 5 and 6) to* FASTERRANDSAMPLINGBSS *(Algorithm 3), we get an algorithm taking* $O(|D|d^2 + \gamma^{-2}d \cdot (d^2 \log |D| + d^\omega))$*-time and using* $O(|D|d)$*-space.*

*Proof.* In the preprocessing stage of FASTERRANDSAMPLINGBSS, we take $O(|D|d^2)$-time for Gram-Schmidt process and $O(|D|d^{\omega-1})$-time for initializing the data structure (Algorithm 5).

The number of iterations is $\gamma^{-2}d$. In each iteration, the matrix $E_j$ can be computed in $O(d^\omega)$-time. And querying the data structure takes $O(d^2 \log(|D|/d) + d^\omega)$-time.

Hence, the total running time is

$$O\left(|D|d^2 + |D|d^{\omega-1} + \gamma^{-2}d(d^2 \log(|D|/d) + d^\omega)\right) = O\left(|D|d^2 + \gamma^{-2}d^{\omega+1} + \gamma^{-2}d^2 \log|D|\right).$$

For the space complexity, the data structure uses $O(|D|d)$-space. The algorithm uses $O(d^2)$ extra space in preprocessing and each iteration. Hence, the total space complexity is $O(|D|d)$. $\qquad\square$

# F  Sketch Distillation for Fourier Sparse Signals

In Section D, we show an oblivious approach for sketching Fourier sparse signals. However, there are two issues of using this sketching method in Signal estimation: 1. The sketch size too large. 2. The noise in the observed signal could have much larger energy on the sketching set than its average energy. To resolve these two issues, in this section, we propose a method called *sketch distillation* to post-process the sketch obtained in Section D that can reduce the sketch size to $O(k)$ and prevent the energy of noise being amplified too much. However, we need some extra information about the signal $x^*(t)$: we assume that the frequencies of the noiseless signal $x(t)$ are known. But the sketch distillation process can still be done *partially oblivious*, i.e., we do not need to access/sample the signal.

In Section F.1, we show our distillation algorithms for one-dimensional signals.

### F.1 Sketch distillation for one-dimensional signals

In this section, we show how to distill the sketch produced by Lemma D.2 from $O(k \log k)$-size to $O(k)$-size, using an $\varepsilon$-well-balanced sampling procedure developed in Section E.

**Lemma F.1** (Fast distillation for one-dimensional signal). *Given* $f_1, f_2, \cdots, f_k \in \mathbb{R}$. *Let* $x^*(t) = \sum_{j=1}^{k} v_j \exp(2\pi \mathbf{i} f_j t)$. *Let* $\eta = \min_{i \neq j} |f_j - f_i|$. *For any accuracy parameter* $\varepsilon \in (0, 0.1)$, *there is an algorithm* FASTDISTILL1D *(Algorithm 7) that runs in* $O(\varepsilon^{-2} k^{\omega+1})$*-time and outputs a set* $S \subset [-T, T]$ *of size* $s = O(k/\varepsilon^2)$ *and a weight vector* $w \in \mathbb{R}_{\geq 0}^s$ *such that,*

$$(1 - \varepsilon)\|x^*(t)\|_T \leq \|x^*(t)\|_{S,w} \leq (1 + \varepsilon)\|x^*(t)\|_T$$

*holds with probability* 0.99.

*Furthermore, for any noise signal* $g(t)$, *the following holds with high probability:*

$$\|g\|_{S,w}^2 \lesssim \|g\|_T^2,$$

*where* $\|x\|_T^2 := \frac{1}{2T} \int_{-T}^{T} |x(t)|^2 \mathrm{d}t$.

*Proof.* For the convenient, in the proof, we use time duration $[-T, T]$. Let $D(t)$ be defined as follows:

$$D(t) = \begin{cases} c/(1 - |t/T|), & \text{for } |t| \leq T(1 - 1/k) \\ c \cdot k, & \text{for } |t| \in [T(1 - 1/k), T] \end{cases}$$

where $c = O(T^{-1} \log^{-1}(k))$ a fixed value such that $\int_{-T}^{T} D(t)\mathrm{d}t = 1$.

First, we randomly pick up a set $S_0 = \{t_1, \cdots, t_{s_0}\}$ of $s_0 = O(\varepsilon_0^{-2} k \log(k) \log(1/\rho_0))$ i.i.d. samples from $D(t)$, and let $w_i' := 2/(T s_0 D(t_i))$ for $i \in [s_0]$ be the weight vector, where $\varepsilon_0, \rho_0$ are parameters to be chosen later.

By Lemma D.2, we know that $(S_0, w')$ gives a good weighted sketch of the signal that can preserve the norm with high probability. More specifically, with probability $1 - \rho_0$,

$$(1 - \varepsilon_0)\|x^*(t)\|_T^2 \leq \|x^*(t)\|_{S_0, w'}^2 \leq (1 + \varepsilon_0)\|x^*(t)\|_T^2. \tag{8}$$

Then, we will select $s = O(k/\varepsilon_1^2)$ elements from $S_0$ and output the corresponding weights $w_1, w_2, \cdots, w_s$ by applying RANDBSS+ with the following parameter: replacing $d$ by $k$, $\varepsilon$ by $\varepsilon_1^2$, and $D$ by $D(t_i) = w_i'/\sum_{j \in [s_0]} w_j'$ for $i \in [s_0]$.

By Theorem E.3 and the property of WBSP (Definition E.1), we obtain that with probability 0.995,

$$(1 - \varepsilon_1)\|x^*(t)\|_{S_0, w'}^2 \leq \|x^*(t)\|_{S, w}^2 \leq (1 + \varepsilon_1)\|x^*(t)\|_{S_0, w'}^2.$$

Combining with Eq. (8), we conclude that

$$\begin{aligned} \|x^*\|_{S,w}^2 &\in [1 - \varepsilon_1, 1 + \varepsilon_1] \cdot \|x^*\|_{S_0, w'}^2 \\ &\in [(1 - \varepsilon_0)(1 - \varepsilon_1), (1 + \varepsilon_0)(1 + \varepsilon_1)] \cdot \|x^*\|_T^2 \\ &\in [1 - \varepsilon, 1 + \varepsilon] \cdot \|x^*\|_T^2, \end{aligned}$$

where the second step follows from Eq. (8) and the last stpe follows by taking $\varepsilon_0 = \varepsilon_1 = \varepsilon/4$.

The overall success probability follows by taking union bound over the two steps and taking $\rho_0 = 0.001$. The running time of Algorithm 7 follows from Claim F.2. And the furthermore part follows from Claim F.3.

The proof of the lemma is then completed. $\square$

**Claim F.2** (Running time of Procedure FASTDISTILL1D in Algorithm 7). *Procedure* FASTDIS-TILL1D *in Algorithm 7 runs in*

$$O(\varepsilon^{-2} k^{\omega+1})$$

*time.*

---

**Algorithm 7** Fast distillation for one-dimensional signal

---

1: **procedure** WEIGHTEDSKETCH$(k, \varepsilon, T, \mathcal{B})$         ▷ Lemma D.2
2:      $c \leftarrow O(T^{-1} \log^{-1}(k))$
3:      $D(t)$ is defined as follows:

$$D(t) \leftarrow \begin{cases} c/((1 - |t/T|) \log k), & \text{if } |t| \leq T(1 - 1/k), \\ c \cdot k, & \text{if } |t| \in [T(1 - 1/k), T]. \end{cases}$$

4:      $S_0 \leftarrow O(\varepsilon^{-2} k \log(k))$ i.i.d. samples from $D$
5:      **for** $t \in S_0$ **do**
6:          $w_t \leftarrow \frac{2}{T \cdot |S_0| \cdot D(t)}$
7:      **end for**
8:      Set a new distribution $D'(t) \leftarrow w_t / \sum_{t' \in S_0} w_{t'}$ for all $t \in S_0$
9:      **return** $D'$
10: **end procedure**
11: **procedure** FASTDISTILL1D$(k, \varepsilon, F = \{f_1, \ldots, f_k\}, T)$         ▷ Lemma F.1
12:      Distribution $D' \leftarrow$ WEIGHTEDSKETCH$(k, \varepsilon, T, \mathcal{B})$
13:      Set the function family $\mathcal{F}$ as follows:

$$\mathcal{F} := \Big\{ f(t) = \sum_{j=1}^{k} v_j \exp(2\pi \mathbf{i} f_j t) \ \Big| \ v_j \in \mathbb{C} \Big\}.$$

14:      $s, \{t_1, t_2, \cdots, t_s\}, w \leftarrow$ RANDBSS+$(k, \mathcal{F}, D', (\varepsilon/4)^2)$       ▷ $s = O(k/\varepsilon^2)$, Algorithm 3
15:      **return** $\{t_1, t_2, \cdots, t_s\}$ and $w$
16: **end procedure**

---

*Proof.* First, it is easy to see that Procedure WEIGHTEDSKETCH takes $O(\varepsilon^{-2} k \log(k))$-time.

By Theorem E.3 with $|D| = O(\varepsilon^{-2} k \log(k))$, $d = k$, we have that the running time of Procedure RANDBSS+ is

$$O\left( k^2 \cdot \varepsilon^{-2} k \log(k) + \varepsilon^{-2} k^3 \log\left( \varepsilon^{-2} k \log(k) \right) + \varepsilon^{-2} k^{\omega+1} \right)$$
$$= O\left( \varepsilon^{-2} k^{\omega+1} \right).$$

Hence, the total running time of Algorithm 7 is $O\left( \varepsilon^{-2} k^{\omega+1} \right)$.

$\square$

**Claim F.3** (Preserve the energy of noise). *Let $(S, w)$ be the outputs of Algorithm 7. Then, we have that*

$$\|g(t)\|_{S,w}^2 \lesssim \|g(t)\|_T^2,$$

*holds with probability 0.99.*

*Proof.* For the convenient, in the proof, we use time duration $[-T, T]$. Algorithm 7 has two stages of sampling.

In the first stage, Procedure WEIGHTEDSKETCH samples a set $S_0 = \{t'_1, \ldots, t'_{s_0}\}$ of i.i.d. samples from the distribution $D$, and a weight vector $w'$. Then, we have

$$\mathbb{E}\left[\|g(t)\|_{S_0, w'}^2\right] = \mathbb{E}\left[ \sum_{i=1}^{s_0} w'_i |g(t'_i)|^2 \right]$$
$$= \sum_{i=1}^{s_0} \mathbb{E}_{t'_i \sim D}[w'_i |g(t'_i)|^2]$$
$$= \sum_{i=1}^{s_0} \mathbb{E}_{t'_i \sim D}\left[ \frac{2}{T s_0 D(t'_i)} |g(t'_i)|^2 \right]$$

$$= \sum_{i=1}^{s_0} \mathop{\mathbb{E}}_{t_i' \sim \mathrm{Uniform}([-T,T])} [s_0^{-1} |g(t_i')|^2]$$

$$= \mathop{\mathbb{E}}_{t \sim \mathrm{Uniform}([-T,T])} [|g(t)|^2]$$

$$= \|g(t)\|_T^2$$

where the first step follows from the definition of the norm, the third step follows from the definition of $w_i$, the forth step follows from $\mathbb{E}_{t \sim D_0(t)}[\frac{D_1(t)}{D_0(t)} f(t)] = \mathbb{E}_{t \sim D_1(t)} f(t)$.

In the second stage, let $P$ denote the Procedure RANDBSS+. With high probability, $P$ is a $\varepsilon$-WBSP (Definition E.1). By the Definition E.1, each sample $t_i \sim D_i(t)$ and $w_i = \alpha_i \cdot \frac{D'(t_i)}{D_i(t_i)}$ in every iteration $i \in [s]$, where $\sum_{i=1}^s \alpha_i \le 5/4$ and $D'(t) = \frac{w_t'}{\sum_{t' \in S_0} w_{t'}'}$. As a result,

$$\mathop{\mathbb{E}}_{P}[\|g(t)\|_{S,w}^2] = \mathop{\mathbb{E}}_{P}\Big[ \sum_{i=1}^s w_i |g(t_i)|^2 \Big]$$

$$= \sum_{i=1}^s \mathop{\mathbb{E}}_{t_i \sim D_i(t_i)} [w_i |g(t_i)|^2]$$

$$= \sum_{i=1}^s \mathop{\mathbb{E}}_{t_i \sim D_i(t_i)} \Big[ \alpha_i \cdot \frac{D'(t_i)}{D_i(t_i)} |g(t_i)|^2 \Big]$$

$$= \sum_{i=1}^s \mathop{\mathbb{E}}_{t_i \sim D'(t_i)} [\alpha_i |g(t_i)|^2]$$

$$\le \sup_{P} \{ \sum_{i=1}^s \alpha_i \} \mathop{\mathbb{E}}_{t \sim D'(t)} [|g(t)|^2]$$

$$= \sup_{P} \{ \sum_{i=1}^s \alpha_i \} \|g(t)\|_{S_0,w'}^2 \cdot ( \sum_{t' \in S_0} w_{t'}' )^{-1}$$

$$\lesssim \rho^{-1} \cdot \|g(t)\|_{S_0,w'}^2.$$

where the first step follows from the definition of the norm, the third step follows from $w_i = \alpha_i \cdot \frac{D'(t_i)}{D_i(t_i)}$, the forth step follows from $\mathbb{E}_{t \sim D_0(t)} \frac{D_1(t)}{D_0(t)} f(t) = \mathbb{E}_{t \sim D_1(t)} f(t)$, the sixth step follows from $D'(t) = \frac{w_t'}{\sum_{t' \in S_0} w_{t'}'}$ and the definition of the norm, the last step follows from $\sum_{i=1}^s \alpha_i \le 5/4$ and $(\sum_{t' \in S_0} w_{t'}')^{-1} = O(\rho^{-1})$ with probability at least $1 - \rho/2$.

Hence, combining the two stages together, we have

$$\mathbb{E}\big[ \mathop{\mathbb{E}}_{P}[\|g(t)\|_{S,w}^2] \big] \lesssim \rho^{-1} \cdot \mathbb{E}\big[ \|g(t)\|_{S_0,w'}^2 \big] = \rho^{-1} \cdot \|g\|_T^2.$$

And by Markov inequality and union bound, we have

$$\Pr\big[ \|g(t)\|_{S,w}^2 \lesssim \rho^{-2} \|g(t)\|_T^2 \big] \le 1 - \rho.$$

$\square$

### F.1.1 Sharper bound for the energy of orthogonal part of noise

In this section, we give a sharper analysis for the energy of $g^\perp$ on the sketch, which is the orthogonal projection of $g$ to the space $\mathcal{F}$. More specifically, we can decompose an arbitrary function $g$ into $g^\parallel + g^\perp$, where $g^\parallel \in \mathcal{F}$ and $\int_{[0,T]} \overline{h(t)} g^\perp(t) \mathrm{d}t = 0$ for all $h \in \mathcal{F}$. The motivation of considering $g^\perp$ is that $g^\parallel$ is also a Fourier sparse signal and its energy will not be amplified in the Signal Estimation problem. And the nontrivial part is to avoid the blowup of the energy of $g^\perp$, which is shown in the following lemma:

**Lemma F.4** (Preserving the orthogonal energy). *Let $\mathcal{F}$ be an $m$-dimensional linear function family with an orthonormal basis $\{v_1, \dots, v_m\}$ with respect to a distribution $D$. Let $P$ be the $\varepsilon$-WBSP that*

generate a sample set $S = \{t_1, \ldots, t_s\}$ and coefficients $\alpha \in \mathbb{R}^s_{>0}$, where each $t_i$ is sampled from distribution $D_i$ for $i \in [s]$. Define the weight vector $w \in \mathbb{R}^s$ be such that $w_i := \alpha_i \frac{D(t_i)}{D_i(t_i)}$ for $i \in [s]$.

For any noise function $g(t)$ that is orthogonal to $\mathcal{F}$ with respect to $D$, the following property holds with probability 0.99:

$$\sum_{i=1}^{m} |\langle g, v_i \rangle_{S,w}|^2 \lesssim \varepsilon \|g\|_D^2,$$

where $\langle g, v \rangle_{S,w} := \sum_{j=1}^{s} w_j \overline{v(t_j)} g(t_j)$.

**Remark F.5.** *We note that this lemma works for both continuous and discrete signals.*

**Remark F.6.** $|\langle g, v_i \rangle_{S,w}|^2$ *corresponds to the energy of $g$ on the sketch points in $S$. On the other hand, if we consider the energy on the whole time domain, we have $\langle g, v_i \rangle = 0$ for all $i \in [m]$. The above lemma indicates that this part of energy could be amplified by at most $O(\varepsilon)$, as long as the sketch comes from a WBSP.*

*Proof.* We can upper-bound the expectation of $\sum_{i=1}^{m} |\langle g, v_i \rangle_{S,w}|^2$ as follows:

$$\mathbb{E}\Big[\sum_{i=1}^{m} |\langle g, v_i \rangle_{S,w}|^2\Big] = \mathop{\mathbb{E}}_{D_1,\ldots,D_s}\Big[\|w\|_1^2 \sum_{i=1}^{m} |\mathop{\mathbb{E}}_{t \sim D'}[\overline{v_j(t)}g(t)]|^2\Big]$$

$$= \mathop{\mathbb{E}}_{D_1,\ldots,D_s}\Big[\sum_{i=1}^{m} |\sum_{j=1}^{s} w_j \overline{v_i(t_j)}g(t_j)|^2\Big]$$

$$= \sum_{i=1}^{m} \mathop{\mathbb{E}}_{D_1,\ldots,D_s}\Big[|\sum_{j=1}^{s} w_j \overline{v_i(t_j)}g(t_j)|^2\Big]$$

$$= \sum_{i=1}^{m} \mathop{\mathbb{E}}_{D_1,\ldots,D_s}\Big[\sum_{j=1}^{s} w_j^2 |v_i(t_j)|^2 |g(t_j)|^2\Big]$$

$$= \sum_{j=1}^{s} \mathop{\mathbb{E}}_{D_j}\Big[\sum_{i=1}^{m} w_j |v_i(t_j)|^2 \cdot w_j |g(t_j)|^2\Big]$$

$$\leq \sum_{j=1}^{s} \sup_{t \in D_j} \Big\{w_j \sum_{i=1}^{m} |v_i(t)|^2\Big\} \cdot \mathop{\mathbb{E}}_{D_j}[w_j |g(t_j)|^2],$$

where the first step follows from Fact F.7, the second step follows from the definition of $D'$, the third follows from the linearity of expectation, the forth step follows from Fact F.8, the last step follows by pulling out the maximum value of $w_j \sum_{i=1}^{k} |v_i(t)|^2$ from the expectation.

Next, we consider the first term:

$$\sup_{t \in D_j} \Big\{w_j \sum_{i=1}^{m} |v_i(t)|^2\Big\} = \sup_{t \in D_j} \Big\{\alpha_j \frac{D(t)}{D_j(t)} \sum_{i=1}^{m} |v_i(t)|^2\Big\}$$

$$= \alpha_j \sup_{t \in D_j} \Big\{\frac{D(t)}{D_j(t)} \sup_{h \in \mathcal{F}} \Big\{\frac{|h(t)|^2}{\|h\|_D^2}\Big\}\Big\}$$

$$= \alpha_j K_{\mathsf{IS},D_j}.$$

where the first step follows from the definition of $w_j$, the second step follows from Fact F.9 that $\sup_{h \in \mathcal{F}}\{\frac{|h(t_j)|^2}{\|h\|_D^2}\} = \sum_{i=1}^{k} |v_i(t_j)|^2$, the last step follows from the definition of $K_{\mathsf{IS},D_j}$ (Eq. (5)).

Then, we bound the last term:

$$\mathop{\mathbb{E}}_{D_j}[w_j |g(t_j)|^2] = \mathop{\mathbb{E}}_{t_j \sim D_j}\Big[\alpha_j \frac{D(t_j)}{D_j(t_j)}|g(t_j)|^2\Big] = \alpha_j \mathop{\mathbb{E}}_{t_j \sim D}[|g(t_j)|^2] = \alpha_j \|g\|_D^2.$$

Combining the two terms together, we have

$$\mathbb{E}\Big[\sum_{i=1}^{m}|\langle g, v_i\rangle_{S,w}|^2\Big] \le \sum_{j=1}^{s}(\alpha_j K_{\mathsf{IS},D_j} \cdot \alpha_j \|g\|_D^2)$$

$$\le \Big(\sum_{j=1}^{s}\alpha_j\Big) \cdot \max_{j\in[s]}\{\alpha_j K_{\mathsf{IS},D_j}\} \cdot \|g\|_D^2$$

$$\le \varepsilon\|g\|_D^2.$$

where the last step follows from $P$ being a $\varepsilon$-WBSP (Definition E.1), which implies that $\sum_{j=1}^{s}\alpha_j = \frac{5}{4}$ and $\alpha_j K_{\mathsf{IS},D_j} \le \varepsilon/2$ for all $j \in [s]$.

Finally, by Markov's inequality, we have that

$$\sum_{i=1}^{m}|\langle g, v_i\rangle_{S,w}|^2 \lesssim \varepsilon\|g\|_D^2$$

holds with probability 0.99. $\qquad\square$

**Fact F.7.**

$$\sum_{i=1}^{m}|\langle g, v_i\rangle_{S,w}|^2 = \|w\|_1^2 \cdot \sum_{i=1}^{m}\Big|\mathop{\mathbb{E}}_{t\sim D'}[\overline{v_i(t)}g(t)]\Big|^2,$$

*where $D'$ is a distribution defined by $D'(t_i) := \frac{w_i}{\|w\|_1}$ for $i \in [s]$.*

*Proof.* We have:

$$\sum_{i=1}^{m}|\langle g, v_i\rangle_{S,w}|^2 = \sum_{i=1}^{m}\Big|\sum_{j=1}^{s}w_j\overline{v_i(t_j)}g(t_j)\Big|^2$$

$$= \sum_{i=1}^{m}\Big|\sum_{j=1}^{s}\frac{w_j\overline{v_i(t_j)}g(t_j)}{\sum_{j'=1}^{s}w_{j'}}\Big|^2 \cdot \Big(\sum_{j'=1}^{s}w_{j'}\Big)^2$$

$$= \Big(\sum_{j'=1}^{s}w_{j'}\Big)^2 \cdot \sum_{i=1}^{m}\Big|\mathop{\mathbb{E}}_{t\sim D'}[\overline{v_i(t)}g(t)]\Big|^2.$$

$\qquad\square$

**Fact F.8.** *For any $i \in [m]$, we have*

$$\mathop{\mathbb{E}}_{D_1,\ldots,D_s}\Big[|\sum_{j=1}^{s}w_j\overline{v_i(t_j)}g(t_j)|^2\Big] = \mathop{\mathbb{E}}_{D_1,\ldots,D_s}\Big[\sum_{j=1}^{m}w_j^2|v_i(t_j)|^2|g(t_j)|^2\Big].$$

*Proof.* We first show that for any $i \in [m]$ and $j \in [s]$,

$$\mathop{\mathbb{E}}_{t_j\sim D_j}[w_j\overline{v_i(t_j)}g(t_j)] = \mathop{\mathbb{E}}_{t_j\sim D_j}[\alpha_j\frac{D(t_j)}{D_j(t_j)}\overline{v_i(t_j)}g(t_j)]$$

$$= \alpha_j\mathop{\mathbb{E}}_{t_j\sim D}[\overline{v_i(t_j)}g(t_j)]$$

$$= 0. \tag{9}$$

where the first step follows from the definition of $w_i$, the third step follows from $g(t)$ is orthonormal with $v_i(t)$ for any $i \in [k]$.

Then, we can expand LHS as follows:

$$\mathop{\mathbb{E}}_{D_1,\ldots,D_s}\Big[|\sum_{j=1}^{s}w_j\overline{v_i(t_j)}g(t_j)|^2\Big]$$

$$
\begin{aligned}
&= \underset{D_1,\ldots,D_s}{\mathbb{E}}\Big[\Big(\sum_{j=1}^{s} w_j \overline{v_i(t_j)}g(t_j)\Big)^{*}\Big(\sum_{j=1}^{s} w_j \overline{v_i(t_j)}g(t_j)\Big)\Big] \\
&= \underset{D_1,\ldots,D_s}{\mathbb{E}}\Big[\sum_{j,j'=1}^{s} w_j w_{j'} v_i(t_j)\overline{g(t_j)v_i(t_{j'})}g(t_{j'})\Big] \\
&= \sum_{j,j'=1}^{s} \underset{D_1,\ldots,D_s}{\mathbb{E}}[w_j w_{j'} v_i(t_j)\overline{g(t_j)v_i(t_{j'})}g(t_{j'})] \\
&= \sum_{j=1}^{s} \mathbb{E}[w_j^2 |v_i(t_j)|^2 |g(t_j)|^2] + \sum_{1\le j<j'\le s} 2\Re \underset{D_1,\ldots,D_j}{\mathbb{E}}[w_j w_{j'} v_i(t_j)\overline{g(t_j)v_i(t_{j'})}g(t_{j'})] \\
&= \mathrm{RHS} + \sum_{1\le j<j'\le s} 2\Re \underset{D_1,\ldots,D_j}{\mathbb{E}}\Big[w_j v_i(t_j)\overline{g(t_j)} \underset{D_{j+1},\ldots,D_{j'}}{\mathbb{E}}[w_{j'}\overline{v_i(t_{j'})}g(t_{j'})]\Big] \\
&= \mathrm{RHS} + \sum_{1\le j<j'\le s} 2\Re \underset{D_1,\ldots,D_j}{\mathbb{E}}[w_j v_i(t_j)\overline{g(t_j)}\cdot 0] \\
&= \mathrm{RHS},
\end{aligned}
$$

where the third step follows from the linearity of expectation, the fifth step follows from $t_j$ only depends on $t_1,\ldots,t_{j-1}$, and the sixth step follows from Eq. (9). $\qquad\square$

**Fact F.9.** *Let $\{v_1,\ldots,v_k\}$ be an orthonormal basis of $\mathcal{F}$ with respect to the distribution $D$. Then, we have*

$$
\sup_{h\in\mathcal{F}}\Big\{\frac{|h(t)|^2}{\|h\|_D^2}\Big\} = \sum_{i=1}^{k} |v_i(t)|^2
$$

*Proof.* Then,

$$
\begin{aligned}
\sup_{h\in\mathcal{F}}\Big\{\frac{|h(t)|^2}{\|h\|_D^2}\Big\} &= \sup_{a\in\mathbb{C}^k}\Big\{\frac{|\sum_{i=1}^{k} a_i v_i(t)|^2}{\|a\|_2^2}\Big\} \\
&= \sup_{a\in\mathbb{C}^k:\|a\|_2=1}\Big|\sum_{i=1}^{k} a_i v_i(t)\Big|^2 \\
&= \sum_{i=1}^{k} |v_i(t)|^2,
\end{aligned}
$$

where the first step follows from each $h\in\mathcal{F}$ can be expanded as $h = \sum_{i=1}^{k} a_i v_i$ and $\|h(t)\|_D^2 = \|a\|_2^2$ (Fact B.9), the second step follows from the Cauchy-Schwartz inequality and taking $a = \frac{v(t)}{\|v(t)\|_2}$. $\qquad\square$

## G  One-dimensional Signal Estimation

In this section, we apply the tools developed in previous sections to show two efficient reductions from Frequency Estimation to Signal Estimation for one-dimensional semi-continuous Fourier signals. The first reduction in Section G.1 is optimal in sample complexity, which takes linear number of samples from the signal but only achieves constant accuracy. The section reduction in Section G.2 takes nearly-linear number of samples but can achieve very high-accuracy (i.e., $(1+\varepsilon)$-estimation error).

### G.1  Sample-optimal reduction

The main theorem of this section is Theorem G.1. The optimal sample complexity is achieved via the sketch distillation in Lemma F.1.

**Theorem G.1** (Sample-optimal algorithm for one-dimensional Signal Estimation)**.** *For $\eta \in \mathbb{R}$, let $\Lambda(\mathcal{B}) \subset \mathbb{R}$ denote the lattice $\Lambda(\mathcal{B}) = \{c\eta \mid c \in \mathbb{Z}\}$. Suppose that $f_1, f_2, \cdots, f_k \in \Lambda(\mathcal{B})$.*

Let $x^*(t) = \sum_{j=1}^k v_j \exp(2\pi\mathbf{i}f_j t)$, and let $g(t)$ denote the noise. Given observations of the form $x(t) = x^*(t) + g(t)$, $t \in [0, T]$. Let $\eta = \min_{i \neq j} |f_j - f_i|$.

Given $D, \eta \in \mathbb{R}_+$. Suppose that there is an algorithm FREQEST that

- *takes $\mathcal{S}_{\text{freq}}$ samples,*

- *runs in $\mathcal{T}_{\text{freq}}$-time, and*

- *outputs a set $\mathcal{L}$ of frequencies such that with probability $0.99$, the following condition holds:*

$$\forall i \in [k], \ \exists f_i' \in \mathcal{L} \text{ s.t. } |f_i - f_i'| \leq \frac{D}{T}.$$

*Then, there is an algorithm (Algorithm 8) such that*

- *takes $O(\widetilde{k} + \mathcal{S}_{\text{freq}})$ samples*

- *runs $O(\widetilde{k}^{\omega+1} + \mathcal{T}_{\text{freq}})$ time,*

- *outputs $y(t) = \sum_{j=1}^{\widetilde{k}} v_j' \cdot \exp(2\pi\mathbf{i}f_j't)$ with $\widetilde{k} = O(|\mathcal{L}|(1 + D/(T\eta)))$ such that with probability at least $0.9$, we have*

$$\|y(t) - x(t)\|_T^2 \lesssim \|g(t)\|_T^2.$$

---

**Algorithm 8** Signal estimation algorithm for one-dimensional signals (sample optimal version)

1: **procedure** SIGNALESTIMATIONFAST$(x, k, F, T, \mathcal{B})$          ▷ Theorem G.1
2:      $\varepsilon \leftarrow 0.01$
3:      $L \leftarrow$ FREQEST$(x, k, D, F, T, \mathcal{B})$
4:      $\{f_1', f_2', \cdots, f_{\widetilde{k}}'\} \leftarrow \{f \in \Lambda(\mathcal{B}) \mid \exists f' \in L, \ |f' - f| < D/T\}$
5:      $s, \{t_1, t_2, \cdots, t_s\}, w \leftarrow$ FASTDISTILL1D$(\widetilde{k}, \sqrt{\varepsilon}, \{f_i'\}_{i \in [\widetilde{k}]}, T, \mathcal{B})$    ▷ $\widetilde{k}, w \in \mathbb{R}^{\widetilde{k}}$,
       Algorithm 7
6:      $A_{i,j} \leftarrow \exp(2\pi\mathbf{i}f_j't_i), \ A \in \mathbb{C}^{s \times \widetilde{k}}$
7:      $b \leftarrow (x(t_1), x(t_2), \cdots, x(t_s))^\top$
8:      Solving the following weighted linear regression          ▷ Fact A.4

$$v' \leftarrow \arg\min_{v' \in \mathbb{C}^{\widetilde{k}}} \|\sqrt{w} \circ (Av' - b)\|_2.$$

9:      **return** $y(t) = \sum_{j=1}^{\widetilde{k}} v_j' \cdot \exp(2\pi\mathbf{i}f_j't)$.
10: **end procedure**

---

*Proof.* First, we recover the frequencies by utilizing the algorithm FREQEST. Let $L$ be the set of frequencies output by the algorithm FREQEST$(x, k, D, T, F, \mathcal{B})$.

We define $\widetilde{L}$ as follows:

$$\widetilde{L} := \left\{ \widetilde{f} \in \Lambda(\mathcal{B}) \mid \exists f' \in L, \ |f' - \widetilde{f}| < D/T \right\}.$$

We use $\widetilde{k}$ to denote the size of set $\widetilde{L}$. And we use $\widetilde{f}_1, \widetilde{f}_2, \cdots, \widetilde{f}_{\widetilde{k}}$ to denote the frequencies in the set $\widetilde{L}$. It is easy to see that

$$\widetilde{k} \leq |\mathcal{L}|(1 + D/(T\eta)).$$

Next, we focus on recovering magnitude $v' \in \mathbb{C}^{\widetilde{k}}$. First we run Procedure FASTDISTILL1D in Algorithm 7 and obtain a set $S = \{t_1, t_2, \cdots, t_s\} \subset [0, T]$ of size $s = O(\widetilde{k})$ and a weight vector

$w \in \mathbb{R}_{>0}^s$. Then, we sample the signal at $t_1, \ldots, t_s$ and let $x(t_1), \ldots, x(t_s)$ be the samples. Consider the following weighted linear regression problem:

$$\min_{v' \in \mathbb{C}^{\widetilde{k}}} \ \left\| \sqrt{w} \circ (Av' - b) \right\|_2, \tag{10}$$

where $\sqrt{w} := (\sqrt{w_1}, \ldots, \sqrt{w_s})$, and the coefficients matrix $A \in \mathbb{C}^{s \times \widetilde{k}}$ and the target vector $b \in \mathbb{C}^s$ are defined as follows:

$$A := \begin{bmatrix} \exp(2\pi \mathbf{i} \widetilde{f}_1 t_1) & \exp(2\pi \mathbf{i} \widetilde{f}_2 t_1) & \cdots & \exp(2\pi \mathbf{i} \widetilde{f}_{\widetilde{k}} t_1) \\ \exp(2\pi \mathbf{i} \widetilde{f}_1 t_2) & \exp(2\pi \mathbf{i} \widetilde{f}_2 t_2) & \cdots & \exp(2\pi \mathbf{i} \widetilde{f}_{\widetilde{k}} t_2) \\ \vdots & \vdots & \ddots & \vdots \\ \exp(2\pi \mathbf{i} \widetilde{f}_1 t_s) & \exp(2\pi \mathbf{i} \widetilde{f}_2 t_s) & \cdots & \exp(2\pi \mathbf{i} \widetilde{f}_{\widetilde{k}} t_s) \end{bmatrix} \text{ and } b := \begin{bmatrix} x(t_1) \\ x(t_2) \\ \vdots \\ x(t_s) \end{bmatrix}$$

Then, we output a signal

$$y(t) = \sum_{j=1}^{\widetilde{k}} v_j' \cdot \exp(2\pi \mathbf{i} \widetilde{f}_j t),$$

where $v'$ is an optimal solution of Eq. (10).

The running time follows from Lemma G.2. And the estimation error guarantee $\|y(t) - x(t)\|_T \lesssim \|g(t)\|_T$ follows from Lemma G.3.

The theorem is then proved. $\qquad\square$

**Lemma G.2** (Running time of Algorithm 8). *Algorithm 8 takes $O(\widetilde{k}^{\omega+1})$-time, giving the output of Procedure* FREQEST.

*Proof.* At Line 5, we run Procedure FASTDISTILL1D, which takes $O(\widetilde{k}^{\omega+1})$-time by Lemma F.1.

At Line 8, we solve the weighted linear regression, which takes

$$O(s\widetilde{k}^{\omega-1}) = O(\widetilde{k}^{\omega})$$

time by Fact A.4.

Thus, the total running time is $O(\widetilde{k}^{\omega+1})$. $\qquad\square$

**Lemma G.3** (Estimation error of Algorithm 8). *Let $y(t)$ be the output signal of Algorithm 8. With high probability, we have*

$$\|y(t) - x(t)\|_T \lesssim \|g(t)\|_T.$$

*Proof.* We have

$$
\begin{aligned}
\|y(t) - x(t)\|_T &\leq \|y(t) - x^*(t)\|_T + \|g(t)\|_T \\
&\leq (1+\varepsilon)\|y(t) - x^*(t)\|_{S,w} + \|g(t)\|_T \\
&\leq (1+\varepsilon)\|y(t) - x(t)\|_{S,w} + (1+\varepsilon)\|g(t)\|_{S,w} + \|g(t)\|_T \\
&\leq (1+\varepsilon)\|x^*(t) - x(t)\|_{S,w} + (1+\varepsilon)\|g(t)\|_{S,w} + \|g(t)\|_T \\
&\lesssim \|x^*(t) - x(t)\|_{S,w} + \|g(t)\|_T \\
&\lesssim \|x^*(t) - x(t)\|_T + \|g(t)\|_T \\
&\lesssim \|g(t)\|_T, \tag{11}
\end{aligned}
$$

where the first step follows from triangle inequality, the second step follows from Lemma F.1 with 0.99 probability, the third step follows from triangle inequality, the forth step follows from $y(t)$ is the optimal solution of the linear system, the fifth step follows from Claim F.3, the sixth step follows from Lemma F.1, and the last step follows from the definition of $g(t)$. $\qquad\square$

## G.2 High-accuracy reduction

In this section, we prove Theorem G.4, which achieves $(1 + \varepsilon)$-estimation error by a sharper bound on the energy of noise in Lemma F.4.

**Theorem G.4** (High-accuracy algorithm for one-dimensional Signal Estimation). *For $\eta \in \mathbb{R}$, let $\Lambda(\mathcal{B}) \subset \mathbb{R}$ denote the lattice $\Lambda(\mathcal{B}) = \{c\eta \mid c \in \mathbb{Z}\}$. Suppose that $f_1, f_2, \cdots, f_k \in \Lambda(\mathcal{B})$. Let $x^*(t) = \sum_{j=1}^{k} v_j \exp(2\pi \mathbf{i} f_j t)$, and let $g(t)$ denote the noise. Given observations of the form $x(t) = x^*(t) + g(t)$, $t \in [0, T]$. Let $\eta = \min_{i \neq j} |f_j - f_i|$.*

*Given $D, \eta \in \mathbb{R}_+$. Suppose that there is an algorithm* FREQEST *that*

- *takes $\mathcal{S}_{\mathsf{freq}}$ samples,*

- *runs in $\mathcal{T}_{\mathsf{freq}}$-time, and*

- *outputs a set $\mathcal{L}$ of frequencies such that, for each $f_i$, there exists an $f_i' \in \mathcal{L}$ with $|f_i - f_i'| \leq D/T$, holds with probability 0.99.*

*Then, there is an algorithm (Algorithm 9) such that*

- *takes $O(\varepsilon^{-1} \widetilde{k} \log(\widetilde{k}) + \mathcal{S})$ samples,*

- *runs $O(\varepsilon^{-1} \widetilde{k}^\omega \log(\widetilde{k}) + \mathcal{T})$ time,*

- *outputs $y(t) = \sum_{j=1}^{\widetilde{k}} v_j' \cdot \exp(2\pi \mathbf{i} f_j' t)$ with $\widetilde{k} = O(|\mathcal{L}|(1 + D/(T\eta)))$ such that with probability at least 0.9, we have*

$$\|y(t) - x^*(t)\|_T^2 \leq (1 + \varepsilon)\|g(t)\|_T^2.$$

**Remark G.5.** *For simplicity, we state the constant failure probability. It is straightforward to get failure probability $\rho$ by blowing up a $\log(1/\rho)$ factor in both samples and running time.*

*Proof.* Let $L$ be the set of frequencies output by the Frequency Estimation algorithm FREQEST. We have the guarantee that with probability 0.99, for each true frequency $f_i$, there exists an $f_i' \in \mathcal{L}$ with $|f_i - f_i'| \leq D/T$. Conditioning on this event, we define a set $\widetilde{L}$ as follows:

$$\widetilde{L} := \{f \in \Lambda(\mathcal{B}) \mid \exists f' \in L, |f' - f| < D/T\}.$$

Since we assume that $\{f_1, \ldots, f_k\} \subset \Lambda(\mathcal{B})$, we have $\{f_1, \ldots, f_k\} \subset \widetilde{L}$. We use $\widetilde{k}$ to denote the size of set $\widetilde{L}$, and we denote the frequencies in $\widetilde{L}$ by $\widetilde{f}_1, \widetilde{f}_2, \cdots, \widetilde{f}_{\widetilde{k}}$.

Next, we need to recover magnitude $v' \in \mathbb{C}^{\widetilde{k}}$.

We first run Procedure WEIGHTEDSKETCH in Algorithm 7 and obtain a set $S = \{t_1, t_2, \cdots, t_s\} \subset [0, T]$ of size $s = O(\varepsilon^{-2} \widetilde{k} \log(\widetilde{k}))$ and a weight vector $w \in \mathbb{R}_{>0}^s$. Then, we sample the signal at $t_1, \ldots, t_s$ and let $x(t_1), \ldots, x(t_s)$ be the samples. Consider the following weighted linear regression problem:

$$\min_{v' \in \mathbb{C}^{\widetilde{k}}} \left\| \sqrt{w} \circ (Av' - b) \right\|_2, \tag{12}$$

where $\sqrt{w} := (\sqrt{w_1}, \ldots, \sqrt{w_s})$, and the coefficients matrix $A \in \mathbb{C}^{s \times \widetilde{k}}$ and the target vector $b \in \mathbb{C}^s$ are defined as follows:

$$A := \begin{bmatrix} \exp(2\pi \mathbf{i} \widetilde{f}_1 t_1) & \exp(2\pi \mathbf{i} \widetilde{f}_2 t_1) & \cdots & \exp(2\pi \mathbf{i} \widetilde{f}_{\widetilde{k}} t_1) \\ \exp(2\pi \mathbf{i} \widetilde{f}_1 t_2) & \exp(2\pi \mathbf{i} \widetilde{f}_2 t_2) & \cdots & \exp(2\pi \mathbf{i} \widetilde{f}_{\widetilde{k}} t_2) \\ \vdots & \vdots & \ddots & \vdots \\ \exp(2\pi \mathbf{i} \widetilde{f}_1 t_s) & \exp(2\pi \mathbf{i} \widetilde{f}_2 t_s) & \cdots & \exp(2\pi \mathbf{i} \widetilde{f}_{\widetilde{k}} t_s) \end{bmatrix} \text{ and } b := \begin{bmatrix} x(t_1) \\ x(t_2) \\ \vdots \\ x(t_s) \end{bmatrix}$$

Note that if $v'$ corresponds to the true coefficients $v$, then we have $\|\sqrt{w} \circ (Av' - b)\|_2 = \|\sqrt{w} \circ g(S)\|_2 = \|g\|_{S,w}$. Let $v'$ be the exact solution of the weighted linear regression in Eq. (12), i.e.,

$$v' := \arg \min_{v' \in \mathbb{C}^{\widetilde{k}}} \left\| \sqrt{w} \circ (Av' - b) \right\|.$$

And we define the output signal to be:

$$y(t) := \sum_{j=1}^{\widetilde{k}} v_j' \cdot \exp(2\pi \mathbf{i} f_j' t).$$

The estimation error guarantee $\|y(t) - x^*(t)\|_T \leq (1 + \varepsilon)\|g(t)\|_T$ follows from Lemma G.7. The running time follows from Lemma G.6.

The theorem is then proved. $\qquad\square$

---

**Algorithm 9** Signal estimation algorithm for one-dimensional signals (high-accuracy version)

---

1: **procedure** SIGNALESTIMATIONACC$(x, \varepsilon, k, F, T, \mathcal{B})$  $\qquad\qquad\qquad\qquad$ ▷ Theorem G.4
2: $\quad L \leftarrow$ FREQEST$(x, k, D, F, T, \mathcal{B})$
3: $\quad \{f_1', f_2', \cdots, f_{\widetilde{k}}'\} \leftarrow \{f \in \Lambda(\mathcal{B}) \mid \exists f' \in L,\ |f' - f| < D/T\}$
4: $\quad s, \{t_1, t_2, \cdots, t_s\}, w \leftarrow$ WEIGHTEDSKETCH$(\widetilde{k}, \sqrt{\varepsilon}, T, \mathcal{B})$ $\qquad$ ▷ $\widetilde{k}, w \in \mathbb{R}^{\widetilde{k}}$, Algorithm 7
5: $\quad A_{i,j} \leftarrow \exp(2\pi \mathbf{i} f_j' t_i),\ A \in \mathbb{C}^{s \times \widetilde{k}}$
6: $\quad b \leftarrow (x(t_1), x(t_2), \cdots, x(t_s))^\top$
7: $\quad$ Solving the following weighted linear regression $\qquad\qquad\qquad\qquad\qquad$ ▷ Fact A.4

$$v' \leftarrow \arg\min_{v' \in \mathbb{C}^{\widetilde{k}}} \|\sqrt{w} \circ (Av' - b)\|_2.$$

8: $\quad$ **return** $y(t) = \sum_{j=1}^{\widetilde{k}} v_j' \cdot \exp(2\pi \mathbf{i} f_j' t)$.
9: **end procedure**

---

**Lemma G.6** (Running time of Algorithm 9). *Algorithm 9 takes $O(\varepsilon^{-1}\widetilde{k}^\omega \log(\widetilde{k}))$-time, giving the output of Procedure* FREQEST.

*Proof.* At Line 7, the regression solver takes

$$O(s\widetilde{k}^{\omega-1}) = O(\varepsilon^{-1}\widetilde{k}\log(\widetilde{k}) \cdot \widetilde{k}^{\omega-1}) = O(\varepsilon^{-1}\widetilde{k}^\omega \log(\widetilde{k}))$$

time. The remaining part of Algorithm 9 takes at most $O(s)$-time. $\qquad\square$

**Lemma G.7** (Estimation error of Algorithm 9). *Let $y(t)$ be the output signal of Algorithm 9. With high probability, we have*

$$\|y(t) - x^*(t)\|_T \leq (1 + \varepsilon)\|g(t)\|_T.$$

*Proof.* Let $\mathcal{F}$ be the family of signals with frequencies in $\widetilde{L}$:

$$\mathcal{F} = \left\{ h(t) = \sum_{j=1}^{\widetilde{k}} v_j \cdot e^{2\pi \mathbf{i} \widetilde{f}_j t} \mid \forall v_j \in \mathbb{C}, j \in [\widetilde{k}] \right\}.$$

Suppose the dimension of $\mathcal{F}$ is $m \leq k$. Let $\{u_1, u_2, \cdots, u_m\}$ be an orthonormal basis of $\mathcal{F}$, i.e.,

$$\frac{1}{T} \int_{[0,T]} u_i(t)\overline{u_j(t)} \mathrm{d}t = \mathbf{1}_{i=j}, \quad \forall i, j \in [m],$$

On the other hand, since $u_i \in \mathcal{F}$, we can also expand these basis vectors in the Fourier basis. Let $V \in \mathbb{C}^{m \times \widetilde{k}}$ be an linear transformation[7] such that

$$u_i = \sum_{j=1}^{\widetilde{k}} V_{i,j} \cdot \exp(2\pi \mathbf{i} \widetilde{f}_j t) \quad \forall i \in [m].$$

---

[7]When $m < \widetilde{k}$, $V$ is not unique, and we take any one of such linear transformation.

Then, we have

$$\begin{bmatrix} \exp(2\pi\mathbf{i}\widetilde{f}_1 t) \\ \vdots \\ \exp(2\pi\mathbf{i}\widetilde{f}_{\widetilde{k}} t) \end{bmatrix} = V^+ \cdot \begin{bmatrix} u_1 \\ \vdots \\ u_m \end{bmatrix},$$

where $V^+ \in \mathbb{C}^{\widetilde{k} \times m}$ is the pseudoinverse of $V$; or equivalently, the $i$-th row of $V^+$ contains the coefficients of expanding $\exp(2\pi\mathbf{i}\widetilde{f}_i t)$ under $\{u_1, \ldots, u_m\}$. Define a linear operator $\alpha : \mathcal{F} \to \mathbb{C}^m$ such that for any $h(t) = \sum_{j=1}^{\widetilde{k}} v_j \exp(2\pi\mathbf{i}f_j t)$,

$$\alpha(h) := V^+ \cdot v,$$

which gives the coefficients of $h$ under the basis $\{u_1, \cdots, u_{\widetilde{k}}\}$.

Define an $s$-by-$m$ matrix $B$ as follows:

$$B := A \cdot V^\top = \begin{bmatrix} u_1(t_1) & u_2(t_1) & \cdots & u_m(t_1) \\ u_1(t_2) & u_2(t_2) & \cdots & u_m(t_2) \\ \vdots & \vdots & \ddots & \vdots \\ u_1(t_s) & u_2(t_s) & \cdots & u_m(t_s) \end{bmatrix}.$$

$B = AV$. It is easy to see that $\mathrm{Im}(B) = \mathrm{Im}(A)$. Thus, solving Eq. (12) is equivalent to solving:

$$\min_{z \in \mathbb{C}^m} \|\sqrt{w} \circ (Bz - b)\|_2. \tag{13}$$

Since $y(t)$ is an solution of Eq. (12), we also know that $\alpha(y)$ is an solution of Eq. (13).

For convenience, we define some notations. Let $\sqrt{W} := \mathrm{diag}(\sqrt{w})$ and define

$$B_w := \sqrt{W} \cdot B,$$
$$X_w := \sqrt{W} \cdot \begin{bmatrix} x(t_1) & x(t_2) & \cdots & x(t_s) \end{bmatrix}^\top$$
$$X_w^* := \sqrt{W} \cdot \begin{bmatrix} x^*(t_1) & x^*(t_2) & \cdots & x^*(t_s) \end{bmatrix}^\top$$

By Fact A.4, we know that the solution of the weighted linear regression Eq. (13) has the following closed form:

$$\alpha(y) = (B^* W B)^{-1} B^* W b = (B_w^* B_w)^{-1} B_w^* X_w. \tag{14}$$

Then, consider the noise in the signal. Since $g$ is an arbitrary noise, let $g^\|$ be the projection of $g(x)$ to $\mathcal{F}$ and $g^\perp = g - g^\|$ be the orthogonal part to $\mathcal{F}$ such that

$$g^\|(t) \in \mathcal{F}, \text{ and } \int_{[0,T]} g^\|(t)\overline{g^\perp(t)}\mathrm{d}t = 0.$$

Similarly, we also define

$$g_w := \sqrt{W} \cdot \begin{bmatrix} g(t_1) & g(t_2) & \cdots & g(t_s) \end{bmatrix}^\top$$
$$g_w^\| := \sqrt{W} \cdot \begin{bmatrix} g^\|(t_1) & g^\|(t_2) & \cdots, g^\|(t_s) \end{bmatrix}^\top,$$
$$g_w^\perp := \sqrt{W} \cdot \begin{bmatrix} g^\perp(t_1) & g^\perp(t_2) & \cdots, g^\perp(t_s) \end{bmatrix}^\top.$$

By Claim G.8, the error can be decomposed into two terms:

$$\|y(t) - x^*(t)\|_T \leq \left\|(B_w^* B_w)^{-1} B_w^* \cdot g_w^\perp\right\|_2 + \left\|(B_w^* B_w)^{-1} B_w^* \cdot g_w^\|\right\|_2.$$

By Claim G.10, we have

$$\left\|(B_w^* B_w)^{-1} B_w^* \cdot g_w^\perp\right\|_2^2 \lesssim \varepsilon \left\|g^\perp(t)\right\|_T^2.$$

And by Claim G.13, we have

$$\left\|(B_w^* B_w)^{-1} B_w^* \cdot g_w^\|\right\|_2^2 = \left\|g^\|\right\|_T^2.$$

Combining them together (and re-scaling $\varepsilon$ be an constant factor), we have that

$$\|y(t) - x^*(t)\|_T \leq \|g^\|\|_T + \sqrt{\varepsilon}\|g^\perp\|_T.$$

Since $\|g^\|\|_T^2 + \|g^\perp\|_T^2 = \|g\|_T^2$, by Cauchy–Schwarz inequality, we have that

$$(\|g^\|\|_T + \sqrt{\varepsilon}\|g^\perp\|_T)^2 \leq (\|g^\|\|_T^2 + \|g^\perp\|_T^2) \cdot (1 + \varepsilon) = (1 + \varepsilon) \cdot \|g\|_T^2.$$

That is,

$$\|y(t) - x^*(t)\|_T^2 \leq (1 + \varepsilon)\|g(t)\|_T^2.$$

$\square$

**Claim G.8** (Error decomposition).

$$\|y(t) - x^*(t)\|_T \leq \left\|(B_w^* B_w)^{-1} B_w^* \cdot g_w^\perp\right\|_2 + \left\|(B_w^* B_w)^{-1} B_w^* \cdot g_w^\|\right\|_2.$$

*Proof.* Since $y, x^* \in \mathcal{F}$ and $\{u_1, \ldots, u_{\widetilde{k}}\}$ is an orthonormal basis, we have $\|y - x^*\|_T = \|\alpha(y) - \alpha(x^*)\|_2$. Furthermore, by Eq. (14), we have $\alpha(y) = (B_w^* B_w)^{-1} B_w^* \cdot X_w$. And by Fact G.9, since $x^* \in \mathcal{F}$, we have $\alpha(x^*) = (B_w^* B_w)^{-1} B_w^* \cdot X_w^*$.

Thus, we have

$$\begin{aligned}
\|\alpha(y) - \alpha(x^*)\|_2 &= \|(B_w^* B_w)^{-1} B_w^* \cdot (X_w - X_w^*)\|_2 \\
&= \|(B_w^* B_w)^{-1} B_w^* \cdot g_w\|_2 \\
&= \|(B_w^* B_w)^{-1} B_w^* \cdot (g_w^\perp + g_w^\|)\|_2 \\
&\leq \|(B_w^* B_w)^{-1} B_w^* \cdot g_w^\perp\|_2 + \|(B_w^* B_w)^{-1} B_w^* \cdot g_w^\|\|_2
\end{aligned}$$

where the second step follows from the definition of $g_w$, the forth step follows from $g_w = g^\| + g^\perp$, and the last step follows from triangle inequality.

Hence, we get that $\|y(t) - x^*(t)\|_T \leq \|(B_w^* B_w)^{-1} B_w^* \cdot g_w^\perp\|_2 + \|(B_w^* B_w)^{-1} B_w^* \cdot g_w^\|\|_2.$ $\square$

**Fact G.9.** *For any $h \in \mathcal{F}$,*

$$\alpha(h) = (B_w^* B_w)^{-1} B_w^* \cdot h_w,$$

*where $h_w = \sqrt{W} \begin{bmatrix} h(t_1) & \cdots & h(t_s) \end{bmatrix}^\top$.*

*Proof.* Suppose $h(t) = \sum_{j=1}^{\widetilde{k}} v_j \exp(2\pi \mathbf{i} \widetilde{f}_j t)$. We have

$$\begin{aligned}
B_w \alpha(h) = \sqrt{W} B \cdot \alpha(h) \\
= \sqrt{W} B \cdot (V^+ v) \\
= h_w,
\end{aligned}$$

where the second step follows from $V^+$ is a change of coordinates.

Hence, by the Moore-Penrose inverse, we have

$$\alpha(h) = B_w^\dagger h_w = (B_w^* B_w)^{-1} B_w^* h_w.$$

$\square$

**Claim G.10** (Bound the first term). *The following holds with high probability:*

$$\left\|(B_w^* B_w)^{-1} B_w^* \cdot g_w^\perp\right\|_2^2 \lesssim \varepsilon \left\|g^\perp(t)\right\|_T^2.$$

*Proof.* By Lemma D.2, with high probability, we have

$$(1 - \varepsilon)\|x\|_T \leq \|x\|_{S,w} \leq (1 + \varepsilon)\|x\|_T,$$

where $(S, w)$ is the output of Procedure WEIGHTEDSKETCH. Conditioned on this event, by Lemma B.11,

$$\lambda(B_w^* B_w) \in [1 - \varepsilon, 1 + \varepsilon],$$

since $B_w$ is the same as the matrix $A$ in the lemma.

Hence,

$$\begin{aligned}
\|(B_w^* B_w)^{-1} B_w^* \cdot g_w^\perp\|_2^2 &\leq \lambda_{\max}((B_w^* B_w)^{-1})^2 \cdot \|B_w^* \cdot g_w^\perp\|_2^2 \\
&\leq (1 - \varepsilon)^{-2} \|B_w^* \cdot g_w^\perp\|_2^2 \\
&\lesssim \varepsilon \|g^\perp(t)\|_T^2
\end{aligned}$$

where the second step follows from $\lambda_{\max}((B_w^* B_w)^{-1}) \leq (1 - \varepsilon)^{-1}$, and the third step follows from Lemma F.4 and Corollary G.12. $\qquad\square$

**Lemma G.11** (Lemma 6.2 of Chen and Price (2019a)). *There exists a universal constant $C_1$ such that given any distribution $D'$ with the same support of $D$ and any $\varepsilon > 0$, the random sampling procedure with $m = C_1(K_{D'} \log d + \varepsilon^{-1} K_{D'})$ i.i.d. random samples from $D'$ and coefficients $\alpha_1 = \cdots = \alpha_m = 1/m$ is an $\varepsilon$-well-balanced sampling procedure.*

**Corollary G.12.** *Procedure* WEIGHTEDSKETCH *in Algorithm 7 is a $\varepsilon$-WBSP (Definition E.1).*

**Claim G.13** (Bound the second term).

$$\left\|(B_w^* B_w)^{-1} B_w^* \cdot g_w^\|\right\|_2^2 = \left\|g^\|\right\|_T^2.$$

*Proof.*

$$\|(B_w^* B_w)^{-1} B_w^* \cdot g_w^\|\|_2^2 = \|\alpha(g^\|)\|_2^2 = \|g^\|\|_T^2,$$

where the first step follows from Fact G.9 and $g^\| \in \mathcal{F}$, the second step follows from the definition of $\alpha$. $\qquad\square$

# H  High-Accuracy Fourier Interpolation Algorithm

In this section, we propose an algorithm for one-dimensional continuous Fourier interpolation problem, which significantly improves the accuracy of the algorithm in Chen et al. (2016).

This section is organized as follows. In Sections H.1 and H.2, we provide some technical tools for Fourier-sparse signals, low-degree polynomials and filter functions. In Section H.3, we design a high sensitivity frequency estimation method using these tools. In Section H.4, we combine the frequency estimation with our Fourier set query framework, and give a $(9+\varepsilon)$-approximate Fourier interpolation algorithm. Then, in Section H.5, we build a sharper error control, and in Section H.6, we analysis the HASHTOBINS procedure. Based on these result, in Section H.8, we develop the ultra-high sensitivity frequency estimation method. In Section H.10, we show the a $(3 + \sqrt{2} + \varepsilon)$-approximate Fourier interpolation algorithm.

## H.1  Technical tools I: Fourier-polynomial equivalence

In this section, we show that low-degree polynomials and Fourier-sparse signals can be transformed to each other with arbitrarily small errors.

The following lemma upper-bounds the error of using low-degree polynomial to approximate Fourier-sparse signal.

**Lemma H.1** (Fourier signal to polynomial, Chen et al. (2016))**.** *For any $\Delta > 0$ and any $\delta > 0$, let $x^*(t) = \sum_{j\in[k]} v_j e^{2\pi \mathbf{i} f_j t}$ where $|f_j| \leq \Delta$ for each $j \in [k]$. There exists a polynomial $P(t)$ of degree at most*

$$d = O(T\Delta + k^3 \log k + k \log 1/\delta)$$

*such that*

$$\|P - x^*\|_T^2 \leq \delta \|x^*\|_T^2.$$

As a corollary, we can expand a Fourier-sparse signal under the *mixed Fourier-monomial basis* (i.e., $\{e^{2\pi \mathbf{i} f_i t} \cdot t^j\}_{i\in[k], j\in[d]}$).

**Corollary H.2** (Mixed Fourier-polynomial approximation)**.** *For any $\Delta > 0$, $\delta > 0$, $n_j \in \mathbb{Z}_{\geq 0}, j \in [k], \sum_{j\in[k]} n_j = k$. Let*

$$x^*(t) = \sum_{j\in[k]} e^{2\pi \mathbf{i} f_j t} \sum_{i=1}^{n_j} v_{j,i} e^{2\pi \mathbf{i} f'_{j,i} t},$$

*where $|f'_{j,i}| \leq \Delta$ for each $j \in [k], i \in [n_j]$. There exist $k$ polynomials $P_j(t)$ for $j \in [k]$ of degree at most*

$$d = O(T\Delta + k^3 \log k + k \log 1/\delta)$$

*such that*

$$\Big\| \sum_{j\in[k]} e^{2\pi \mathbf{i} f_j t} P_j(t) - x^*(t) \Big\|_T^2 \leq \delta \|x^*(t)\|_T^2.$$

The following lemma bounds the error of approximating a low-degree polynomial using Fourier-sparse signal.

**Lemma H.3** (Polynomial to Fourier signal, Chen et al. (2016))**.** *For any degree-$d$ polynomial $Q(t) = \sum\limits_{j=0}^{d} c_j t^j$, any $T > 0$ and any $\varepsilon > 0$, there always exist $\gamma > 0$ and*

$$x^*(t) = \sum_{j=0}^{d} \alpha_j e^{2\pi \mathbf{i}(\gamma j)t}$$

*with some coefficients $\alpha_0, \cdots, \alpha_d$ such that*

$$\forall t \in [0, T], |x^*(t) - Q(t)| \leq \varepsilon.$$

## H.2 Technical tools II: filter functions

In this section, we introduce the filter functions $H$ and $G$ designed by Chen et al. (2016), and we generalize their constructions to achieve higher sensitivity.

We first construct the $H$-filter, which uses rect and sinc functions.

**Fact H.4** (rect function Fourier transform). *For $s > 0$, let $\mathrm{rect}_s(t) := \mathbf{1}_{|t| \leq s/2}$. Then, we have*

$$\widehat{\mathrm{rect}_s}(f) = \mathrm{sinc}(sf) = \frac{\sin(sf)}{\pi sf}.$$

**Definition H.5.** *Given $s_1, s_2 > 0$ and an even number $\ell \in \mathbb{N}_+$, we define the filter function $H_1(t)$ and its Fourier transform $\widehat{H}_1(f)$ as follows:*

$$
\begin{aligned}
H_1(t) &= s_0 \cdot (\mathrm{sinc}^\ell(s_1 t)) \star \mathrm{rect}_{s_2}(t) \\
\widehat{H}_1(f) &= s_0 \cdot (\mathrm{rect}_{s_1} \star \cdots \star \mathrm{rect}_{s_1})(f) \cdot \mathrm{sinc}(fs_2)
\end{aligned}
$$

*where $s_0 = C_0 s_1 \sqrt{\ell}$ is a normalization parameter such that $H_1(0) = 1$, and $\star$ means convolution.*

**Definition H.6** ($H$-filter's construction, Chen et al. (2016)). *Given any $0 < s_1, s_3 < 1$, $0 < \delta < 1$, we define $H_{s_1, s_3, \delta}(t)$ from the filter function $H_1(t)$ (Definition H.5) as follows:*

- *let $\ell := \Theta(k \log(k/\delta))$, $s_2 := 1 - \frac{2}{s_1}$, and*

- *shrink $H_1$ by a factor $s_3$ in time domain, i.e.,*

$$
\begin{aligned}
H_{s_1, s_3, \delta}(t) &:= H_1(t/s_3) & (15) \\
\widehat{H_{s_1, s_3, \delta}}(f) &= s_3 \widehat{H}_1(s_3 f) & (16)
\end{aligned}
$$

*We call the "filtered cluster" around a frequency $f_0$ to be the support of $(\delta_{f_0} \star \widehat{H_{s_1, s_3, \delta}})(f)$ in the frequency domain and use*

$$\Delta_h = |\mathrm{supp}(\widehat{H_{s_1, s_3, \delta}})| = \frac{s_1 \cdot \ell}{s_3} \tag{17}$$

*to denote the width of the cluster.*

**Lemma H.7** (High sensitivity $H$-filter's properties). *Given $\varepsilon \in (0, 0.1)$, $s_1, s_3 \in (0, 1)$ with $\min(\frac{1}{1-s_3}, s_1) \geq \widetilde{O}(k^4)/\varepsilon$, and $\delta \in (0, 1)$. Let the filter function $H := H_{s_1, s_3, \delta}(t)$ defined in Definition H.6. Then, $H$ satisfies the following properties:*

$$
\begin{aligned}
\text{Property I}: \quad & H(t) \in [1 - \delta, 1], \text{ when } |t| \leq (\frac{1}{2} - \frac{2}{s_1})s_3. \\
\text{Property II}: \quad & H(t) \in [0, 1], \text{ when } (\frac{1}{2} - \frac{2}{s_1})s_3 \leq |t| \leq \frac{1}{2}s_3. \\
\text{Property III}: \quad & H(t) \leq s_0 \cdot (s_1(\frac{|t|}{s_3} - \frac{1}{2}) + 2)^{-\ell}, \text{ when } |t| > \frac{1}{2}s_3. \\
\text{Property IV}: \quad & \mathrm{supp}(\widehat{H}) \subseteq [-\frac{s_1 \ell}{2s_3}, \frac{s_1 \ell}{2s_3}].
\end{aligned}
$$

*For any exact $k$-Fourier-sparse signal $x^*(t)$, we shift the interval from $[0, T]$ to $[-1/2, 1/2]$ and consider $x^*(t)$ for $t \in [-1/2, 1/2]$ to be our observation, which is also $x^*(t) \cdot \mathrm{rect}_1(t)$.*

$$
\begin{aligned}
\text{Property V}: \quad & \int_{-\infty}^{+\infty} \left| x^*(t) \cdot H(t) \cdot (1 - \mathrm{rect}_1(t)) \right|^2 \mathrm{d}t < \delta \int_{-\infty}^{+\infty} |x^*(t) \cdot \mathrm{rect}_1(t)|^2 \mathrm{d}t. \\
\text{Property VI}: \quad & \int_{-\infty}^{+\infty} |x^*(t) \cdot H(t) \cdot \mathrm{rect}_1(t)|^2 \mathrm{d}t \in [1 - \varepsilon, 1] \cdot \int_{-\infty}^{+\infty} |x^*(t) \cdot \mathrm{rect}_1(t)|^2 \mathrm{d}t.
\end{aligned}
$$

**Remark H.8.** *By Property I, and II, and III, we have that $H(t) \leq 1$ for $t \in [0, T]$.*

*Proof.* The proof of Property I - V easily follows from Chen et al. (2016). We prove Property VI in below.

First, because of for any $t$, $|H_1(t)| \leq 1$, thus we prove the upper bound for LHS,

$$\int_{-\infty}^{+\infty} |x^*(t) \cdot H(t) \cdot \text{rect}_1(t)|^2 dt \leq \int_{-\infty}^{+\infty} |x^*(t) \cdot 1 \cdot \text{rect}_1(t)|^2 dt.$$

Second, as mentioned early, we need to prove the general case when $s_3 = 1 - 1/\text{poly}(k)$. Define interval $S = [-s_3(\frac{1}{2} - \frac{1}{s_1}), s_3(\frac{1}{2} - \frac{1}{s_1})]$, by definition, $S \subset [-1/2, 1/2]$. Then define $\overline{S} = [-1/2, 1/2] \setminus S$, which is $[-1/2, -s_3(\frac{1}{2} - \frac{1}{s_1})) \cup (s_3(\frac{1}{2} - \frac{1}{s_1}), 1/2]$. By Property I, we have

$$\int_S |x^*(t) \cdot H(t)|^2 dt \geq (1 - \delta)^2 \int_S |x^*(t)|^2 dt \tag{18}$$

Then we can show

$$\int_{\overline{S}} |x^*(t)|^2 dt$$

$$\leq |\overline{S}| \cdot \max_{t \in [-1/2, 1/2]} |x^*(t)|^2$$

$$\leq (1 - s_3(1 - \frac{2}{s_1})) \cdot O(k^2) \int_{-\frac{1}{2}}^{\frac{1}{2}} |x^*(t)|^2 dt$$

$$\leq \varepsilon \int_{-\frac{1}{2}}^{\frac{1}{2}} |x^*(t)|^2 dt \tag{19}$$

where the first step follows from $\overline{S} \subset [-1/2, 1/2]$, the second step follows from Theorem C.1, the third step follows from $(1 - s_3(1 - \frac{2}{s_1})) \cdot O(k^2) \leq \varepsilon$.

Combining Equations (18) and (19) gives a lower bound for LHS,

$$\int_{-\infty}^{+\infty} |x^*(t) \cdot H(t) \cdot \text{rect}_1(t)|^2 dt$$

$$\geq \int_S |x^*(t) H(t)|^2 dt$$

$$\geq (1 - 2\delta) \int_S |x^*(t)|^2 dt$$

$$= (1 - 2\delta) \int_{S \cup \overline{S}} |x^*(t)|^2 dt - (1 - 2\delta) \int_{\overline{S}} |x^*(t)|^2 dt$$

$$\geq (1 - 2\delta) \int_{S \cup \overline{S}} |x^*(t)|^2 dt - (1 - 2\delta)\varepsilon \int_{S \cup \overline{S}} |x^*(t)|^2 dt$$

$$= (1 - 2\delta - \varepsilon) \int_{-\frac{1}{2}}^{\frac{1}{2}} |x^*(t)|^2 dt$$

$$\geq (1 - 2\varepsilon) \int_{-\infty}^{+\infty} |x^*(t) \cdot \text{rect}_1(t)|^2 dt,$$

where the first step follows from $S \subset [-1/2, 1/2]$, the second step follows from Eq. (18), the third step follows from $S \cap \overline{S} = \emptyset$, the forth step follows from Eq. (19), the fifth step follows from $S \cup \overline{S} = [-1/2, 1/2]$, the last step follows from $\varepsilon \gg \delta$.

$\square$

As remarked in Chen et al. (2016), to match $(H(t), \widehat{H}(f))$ on $[-1/2, 1/2]$ with signal $x(t)$ on $[0, T]$, we will scale the time domain from $[-1/2, 1/2]$ to $[-T/2, T/2]$ and shift it to $[0, T]$. Then, in frequency domain, the Property IV in Lemma H.7 becomes

$$\text{supp}(\widehat{H}(f)) \subseteq [-\frac{\Delta_h}{2}, \frac{\Delta_h}{2}], \text{ where } \Delta_h = \frac{s_1 \ell}{s_3 T}. \tag{20}$$

We also need another filter function, $G$, whose construction and properties are given below.

**Definition H.9** ($G$-filter's construction, Chen et al. (2016)). *Given $B > 1$, $\delta > 0$, $\alpha > 0$. Let $l := \Theta(\log(k/\delta))$. Define $G_{B,\delta,\alpha}(t)$ and its Fourier transform $\widehat{G_{B,\delta,\alpha}}(f)$ as follows:*

$$G_{B,\delta,\alpha}(t) := b_0 \cdot (\mathrm{rect}_{\frac{B}{(\alpha\pi)}}(t))^{\star l} \cdot \mathrm{sinc}(t\tfrac{\pi}{2B}),$$

$$\widehat{G_{B,\delta,\alpha}}(f) := b_0 \cdot (\mathrm{sinc}(\tfrac{B}{\alpha\pi}f))^{\cdot l} * \mathrm{rect}_{\frac{\pi}{2B}}(f),$$

*where $b_0 = \Theta(B\sqrt{l}/\alpha)$ is the normalization factor such that $\widehat{G}(0) = 1$.*

**Lemma H.10** ($G$-filter's properties, Chen et al. (2016)). *Given $B > 1$, $\delta > 0$, $\alpha > 0$, let $G := G_{B,\delta,\alpha}(t)$ be defined in Definition H.9. Then, $G$ satisfies the following properties:*

$$\text{Property I}: \quad \widehat{G}(f) \in [1 - \delta/k, 1], \text{ if } |f| \leq (1 - \alpha)\frac{2\pi}{2B}.$$

$$\text{Property II}: \quad \widehat{G}(f) \in [0, 1], \text{ if } (1 - \alpha)\frac{2\pi}{2B} \leq |f| \leq \frac{2\pi}{2B}.$$

$$\text{Property III}: \quad \widehat{G}(f) \in [-\delta/k, \delta/k], \text{ if } |f| > \frac{2\pi}{2B}.$$

$$\text{Property IV}: \quad \mathrm{supp}(G(t)) \subset [\frac{l}{2} \cdot \frac{-B}{\pi\alpha}, \frac{l}{2} \cdot \frac{B}{\pi\alpha}].$$

$$\text{Property V}: \quad \max_t |G(t)| \lesssim \mathrm{poly}(B, l).$$

## H.3   High sensitivity frequency estimation

In this section, we show a high sensitivity frequency estimation. Compared with the result in Chen et al. (2016), we relax the condition of the frequencies that can be recovered by the algorithm.

**Definition H.11** (Definition 2.4 in Chen et al. (2016)). *Given $x^*(t) = \sum_{j=1}^{k} v_j e^{2\pi \mathbf{i} f_j t}$, any $\mathcal{N} > 0$, and a filter function $H$ with bounded support in frequency domain. Let $L_j$ denote the interval of $\mathrm{supp}(\widehat{e^{2\pi \mathbf{i} f_j t} \cdot H})$ for each $j \in [k]$. Define an equivalence relation $\sim$ on the frequencies $f_i$ as follows:*

$$f_i \sim f_j \;\; \textit{iff} \;\; L_i \cap L_j \neq \emptyset \;\; \forall i, j \in [k].$$

*Let $S_1, \ldots, S_n$ be the equivalence classes under this relation for some $n \leq k$.*

*Define $C_i := \bigcup_{f \in S_i} L_i$ for each $i \in [n]$. We say $C_i$ is an $\mathcal{N}$-heavy cluster iff*

$$\int_{C_i} |\widehat{H \cdot x^*}(f)|^2 \mathrm{d}f \geq T \cdot \mathcal{N}^2/k.$$

The following claim gives a tight error bound for approximating the true signal $x^*(t)$ by the signal $x_{S^*}(t)$ whose frequencies are in heavy-clusters. It improves the Claim 2.5 in Chen et al. (2016).

**Claim H.12** (Approximation by heavy-clusters). *Given $x^*(t) = \sum_{j=1}^{k} v_j e^{2\pi \mathbf{i} f_j t}$ and any $\mathcal{N} > 0$, let $C_1, \cdots, C_l$ be the $\mathcal{N}$-heavy clusters from Definition H.11. For*

$$S^* = \left\{ j \in [k] \middle| f_j \in C_1 \cup \cdots C_l \right\},$$

*we have $x_{S^*}(t) = \sum_{j \in S^*} v_j e^{2\pi \mathbf{i} f_j t}$ approximating $x^*$ within distance*

$$\|x_{S^*} - x^*\|_T^2 \leq (1 - l/k)(1 + \varepsilon)\mathcal{N}^2.$$

*Proof.* Let $H$ be the filter function defined as in Definition H.6.

Let

$$x_{\overline{S^*}}(t) := \sum_{j \in [k] \setminus S^*} v_j e^{2\pi \mathbf{i} f_j t}.$$

Notice that $\|x^* - x_{S^*}\|_T^2 = \|x_{\overline{S^*}}\|_T^2$.

By Property VI in Lemma H.7 with setting $\varepsilon = \varepsilon_0 := \varepsilon/2$, we have

$$
\begin{aligned}
(1 - \varepsilon_0) \cdot T \|x_{\overline{S^*}}\|_T^2 &= (1 - \varepsilon_0) \int_0^T |x_{\overline{S^*}}(t)|^2 \mathrm{d}t \\
&= (1 - \varepsilon_0) \int_0^T |x_{\overline{S^*}}(t) \cdot \mathrm{rect}_T(t)|^2 \mathrm{d}t \\
&\le \int_{-\infty}^{+\infty} |x_{\overline{S^*}}(t) \cdot H(t) \cdot \mathrm{rect}_T(t)|^2 \mathrm{d}t, \\
&\le \int_{-\infty}^{+\infty} |x_{\overline{S^*}}(t) \cdot H(t)|^2 \mathrm{d}t,
\end{aligned}
$$

where the first step follows from the definition of the norm, the second step follows from the definition of $\mathrm{rect}_T(t) = 1, \forall t \in [0, T]$, the third step follows from Lemma H.7, the forth step follows from $\mathrm{rect}_T(t) \le 1$.

From Definition H.11, we have

$$
\begin{aligned}
\int_{-\infty}^{+\infty} |x_{\overline{S^*}}(t) \cdot H(t)|^2 \mathrm{d}t &= \int_{-\infty}^{+\infty} |\widehat{x_{\overline{S^*}} \cdot H}(f)|^2 \mathrm{d}f \\
&= \int_{[-\infty, +\infty] \setminus C_1 \cup \cdots \cup C_l} |\widehat{x^* \cdot H}(f)|^2 \mathrm{d}f \\
&\le (k - l) \cdot T \mathcal{N}^2 / k.
\end{aligned}
$$

where the first step follows from Parseval's theorem, the second step follows from Definition H.11, Property IV of Lemma H.7, the definition of $S^*$, thus, $\mathrm{supp}(\widehat{x_{S^*} \cdot H}(f)) = C_1 \cup \cdots \cup C_l$, $\mathrm{supp}(\widehat{x_{S^*} \cdot H}(f)) \cap \mathrm{supp}(\widehat{x_{\overline{S^*}} \cdot H}(f))) = \emptyset$, the last step follows from Definition H.11.

Overall, we have $(1 - \varepsilon_0) \|x_{\overline{S^*}}\|_T^2 \le \mathcal{N}^2$. Thus, $\|x_{S^*}(t) - x^*(t)\|_T^2 \le (1 - l/k)((1 + \varepsilon)\mathcal{N}^2$ by the basic algebra fact: $\frac{1}{1 - \varepsilon/2} \le 1 + \varepsilon$ for any $\varepsilon \in [0, 1]$. $\qquad \square$

Due to the noisy observations, not all frequencies in heavy-clusters are recoverable. Thus, we define the recoverable frequency as follows:

**Definition H.13** (Recoverable frequency). *Let $C$ be an $\mathcal{N}_1$-heavy cluster. We say $C$ is $\mathcal{N}_2$-recoverable if it satisfies:*

$$\int_C |\widehat{H \cdot x}(f)|^2 \ge T \mathcal{N}_2^2 / k.$$

*A frequency $f$ is $(\mathcal{N}_1, \mathcal{N}_2)$-recoverable if $f$ is in an $\mathcal{N}_1$-heavy, $\mathcal{N}_2$-recoverable cluster $C$.*

The following lemma shows that most heavy clusters are also recoverable.

**Lemma H.14** (Heavy-clusters are almost recoverable). *Let $x^*(t) = \sum_{j=1}^k v_j e^{2\pi \mathbf{i} f_j t}$ and $x(t) = x^*(t) + g(t)$ be our observable signal. Let $\mathcal{N}^2 := \|g\|_T^2 + \delta \|x^*\|_T^2$. Let $C_1, \cdots, C_l$ are the $2\mathcal{N}$-heavy clusters from Definition H.11. Let $S^*$ denotes the set of frequencies $f^* \in \{f_j\}_{j \in [k]}$ such that, $f^* \in C_i$ for some $i \in [l]$. Let $S \subset S^*$ be the set of $(2\mathcal{N}, \mathcal{N})$-recoverable frequencies.*

*Then we have that,*

$$\|x_S - x^*\|_T \le (3 - l/k + \varepsilon)\mathcal{N}.$$

*Proof.* If a cluster $C_i$ is $2\mathcal{N}$-heavy but not $\mathcal{N}$-recoverable, then it holds that:

$$\int_{C_i} |\widehat{H \cdot x^*}(f)|^2 \mathrm{d}f \geq 4T\mathcal{N}^2/k \geq 4 \int_{C_i} |\widehat{H \cdot x}(f)|^2 \mathrm{d}f \tag{21}$$

where the first steps follows from $C_i \subset \bigcup_{f_j \in S^*} C_j$, the second step follows from $C_i \not\subset \bigcup_{f_j \in S} C_j$.

So,

$$\begin{aligned}
\int_{C_i} |\widehat{H \cdot g}(f)|^2 \mathrm{d}f &= \int_{C_i} |\widehat{H \cdot (x - x^*)}(f)|^2 \mathrm{d}f \\
&\geq \left( \sqrt{\int_{C_i} |\widehat{H \cdot x^*}(f)|^2 \mathrm{d}f} - \sqrt{\int_{C_i} |\widehat{H \cdot x}(f)|^2 \mathrm{d}f} \right)^2 \\
&\geq \frac{1}{4} \int_{C_i} |\widehat{H \cdot x^*}(f)|^2 \mathrm{d}f
\end{aligned} \tag{22}$$

where the first step follows from $g(t) = x(t) - x^*(t)$, and the second step follows from triangle inequality, the last step follows from Eq. (21).

Let $C' := \bigcup_{f_j \in S^* \setminus S} C_j$, i.e., the union of heavy but not recoverable clusters. Then, we have

$$\|\widehat{H \cdot g}\|_2^2 \geq \sum_{C_i \in C'} \int_{C_i} |\widehat{H \cdot g(f)}|^2 \mathrm{d}f \geq \frac{1}{4} \sum_{C_i \in C'} \int_{C_i} |\widehat{H \cdot x^*}(f)|^2 \mathrm{d}f \tag{23}$$

where the first step follows from the definition of the norm and $C_i \cap C_j = \emptyset, \forall i \neq j$, the second step follows from Eq. (22).

Then we have that

$$\begin{aligned}
T\|x_{S^* \setminus S}\|_T^2 &\leq \frac{T}{1 - \varepsilon/2} \|x_{S^* \setminus S} \cdot H\|_T^2 \\
&\leq (1 + \varepsilon) \sum_{C_i \in C'} \int_{C_i} |\widehat{H \cdot x^*}(f)|^2 \mathrm{d}f \\
&\leq 4(1 + \varepsilon) \|\widehat{H \cdot g}\|_2^2 \\
&= 4(1 + \varepsilon) T \|H \cdot g\|_T^2 \\
&\leq 4(1 + \varepsilon) T \|g\|_T^2 \\
&\leq 4(1 + \varepsilon) T \mathcal{N}^2.
\end{aligned}$$

where the first step follows from Property VI of $H$ in Lemma H.7 (taking $\varepsilon$ there to be $\varepsilon/2$), the second step follows from $\varepsilon \in [0, 1]$ and the definition of $C_i$, the third step follows from Eq. (23), the forth step follows from $g(t) = 0, \forall t \notin [0, T]$, the fifth step follows from Remark H.8, the last step follows from the definition of $\mathcal{N}^2$. Thus, we get that:

$$\|x_{S^* \setminus S}\|_T \leq (2 - l/k + \varepsilon)\mathcal{N}, \tag{24}$$

which follows from $\sqrt{1 + \varepsilon} \leq 1 + \varepsilon/2$.

Finally, we can conclude that

$$\begin{aligned}
\|x_S - x^*\|_T &\leq \|x_S - x_{S^*}\|_T + \|x_{S^*} - x^*\|_T \\
&= \|x_{S^* \setminus S}\|_T + \|x_{S^*} - x^*\|_T \\
&\leq \|x_{S^* \setminus S}\|_T + (1 + \varepsilon)\mathcal{N} \\
&\leq (3 - l/k + 2\varepsilon)\mathcal{N},
\end{aligned}$$

where the first step follows from triangle inequality, the second step follows from the definition of $x_{S^* \setminus S}$, the third step follows from Claim H.12, the last step follows from Eq. (24). The lemma follows by re-scaling $\varepsilon$ to $\varepsilon/2$. $\qquad\square$

## H.4 $(9 + \varepsilon)$-approximate Fourier interpolation algorithm

The goal of this section is to prove Theorem H.20, which gives a Fourier interpolation algorithm with approximation error $(9 + \varepsilon)$. It improves the constant (more than 1000) error algorithm in Chen et al. (2016).

**Claim H.15** (Mixed Fourier-polynomial energy bound, Chen et al. (2016)). *For any*

$$u(t) \in \mathrm{span}\left\{e^{2\pi \mathbf{i} f_i t} \cdot t^j \;\middle|\; j \in \{0, \cdots, d\}, i \in [k]\right\},$$

*we have that*

$$\max_{t \in [0, T]} |u(t)|^2 \lesssim (kd)^4 \log^3(kd) \cdot \|u\|_T^2$$

**Claim H.16** (Condition number of Mixed Fourier-polynomial). *Let $\mathcal{F}$ is a linear function family as follows:*

$$\mathcal{F} := \mathrm{span}\left\{e^{2\pi \mathbf{i} f_i t} \cdot t^j \;\middle|\; j \in \{0, \cdots, d\}, i \in [k]\right\},$$

*Then the condition number of $\mathrm{Uniform}[0, T]$ with respect to $\mathcal{F}$ is as follows:*

$$K_{\mathrm{Uniform}[0,T]} := \sup_{t \in [0,T]} \sup_{f \in \mathcal{F}} \frac{|f(t)|^2}{\|f\|_T^2} = O((kd)^4 \log^3(kd))$$

The following definition extends the well-balanced sampling procedure (Definition E.1) to high probability.

**Definition H.17** $((\varepsilon, \rho)$-well-balanced sampling procedure). *Given a linear family $\mathcal{F}$ and underlying distribution $D$, let $P$ be a random sampling procedure that terminates in $m$ iterations ($m$ is not necessarily fixed) and provides a coefficient $\alpha_i$ and a distribution $D_i$ to sample $x_i \sim D_i$ in every iteration $i \in [m]$.*

*We say $P$ is an $\varepsilon$-WBSP if it satisfies the following two properties:*

   *1. With probability $1 - \rho$, for weight $w_i = \alpha_i \cdot \frac{D(x_i)}{D_i(x_i)}$ of each $i \in [m]$,*

$$\sum_{i=1}^m w_i \cdot |h(x_i)|^2 \in \left[1 - 10\sqrt{\varepsilon}, 1 + 10\sqrt{\varepsilon}\right] \cdot \|h\|_D^2 \quad \forall h \in \mathcal{F}.$$

   *2. The coefficients always have $\sum_{i=1}^m \alpha_i \leq \frac{5}{4}$ and $\alpha_i \cdot K_{\mathsf{IS}, D_i} \leq \frac{\varepsilon}{2}$ for all $i \in [m]$.*

The following lemma shows an $(\varepsilon, \rho)$-WBSP for mixed Fourier-polynomial family.

**Lemma H.18** (WBSP for mixed Fourier-polynomial family). *Given any distribution $D'$ with the same support of $D$ and any $\varepsilon > 0$, the random sampling procedure with $m = O(\varepsilon^{-1} K_{\mathsf{IS}, D'} \log(d/\rho))$ i.i.d. random samples from $D'$ and coefficients $\alpha_1 = \cdots = \alpha_m = 1/m$ is an $(\varepsilon, \rho)$-WBSP.*

*Proof.* By Lemma B.12 with setting $\varepsilon = \sqrt{\varepsilon}$, we have that, as long as $m \geq O(\frac{1}{\varepsilon} \cdot K_{\mathsf{IS}, D'} \log \frac{d}{\rho})$, then with probability $1 - \rho$,

$$\|A^* A - I\|_2 \leq \sqrt{\varepsilon}$$

By Lemma B.11, we have that, for every $h \in \mathcal{F}$,

$$\sum_{j=1}^s w_j \cdot |h(x_j)|^2 \in [1 \pm \varepsilon] \cdot \|h\|_D^2,$$

where $S$ is the $m$ i.i.d. random samples from $D'$, $w_i = \alpha_i D(x_i)/D'(x_i)$.

Moreover, $\sum_{i=1}^m \alpha_i = 1 \leq 5/4$ and

$$\alpha_i \cdot K_{\mathsf{IS}, D'} = \frac{K_{\mathsf{IS}, D'}}{m} \leq \frac{\varepsilon}{\log(d/\rho)} \leq \varepsilon,$$

where the first step follows from the definition of $\alpha_i$, the second step follows from the definition of $m$, the third step follows from $\log(d/\rho) > 1$. $\qquad\square$

Now, we can solve the Signal Estimation problem for mixed Fourier-polynomial signals.

**Lemma H.19** (Mixed Fourier-polynomial signal estimation). *Given $d$-degree polynomials $P_j(t), j \in [k]$ and frequencies $f_j, j \in [k]$. Let $x_S(t) = \sum_{j=1}^k P_j(t) \exp(2\pi \mathbf{i} f_j t)$, and let $g(t)$ denote the noise. Given observations of the form $x(t) := x_S(t) + g'(t)$ for arbitrary noise $g'$ in time duration $t \in [0, T]$.*

*Then, there is an algorithm such that*

- *takes $O(\varepsilon^{-1} \mathrm{poly}(kd) \log(1/\rho))$ samples from $x(t)$,*

- *runs $O(\varepsilon^{-1} \mathrm{poly}(kd) \log(1/\rho))$ time,*

- *outputs $y(t) = \sum_{j=1}^k P_j'(t) \exp(2\pi \mathbf{i} f_j t)$ with $d$-degree polynomial $P_j'(t)$, such that with probability at least $1 - \rho$, we have*

$$\|y - x_S\|_T^2 \le (1 + \varepsilon)\|g'\|_T^2.$$

*Proof sketch.* The proof is almost the same as Theorem G.4 where we follow the four-step Fourier set-query framework. Claim H.15 gives the energy bound for the family of mixed Fourier-polynomial signals, which implies that uniformly sampling $m = \widetilde{O}(\varepsilon^{-1}|L|^4 d^4)$ points in $[0, T]$ forms an oblivious sketch for $x^*$. Moreover, by Lemma H.18, we know that it is also an $(\varepsilon, \rho)$-WBSP, which gives the error guarantee. Then, we can obtain a mixed Fourier-polynomial signal $y(t)$ by solving a weighted linear regression. □

Now, we are ready to prove the main result of this section, a $(9 + \varepsilon)$-approximate Fourier interpolation algorithm.

**Theorem H.20** (Fourier interpolation with $(9 + \varepsilon)$-approximation error). *Let $x(t) = x^*(t) + g(t)$, where $x^*$ is $k$-Fourier-sparse signal with frequencies in $[-F, F]$. Given samples of $x$ over $[0, T]$ we can output $y(t)$ such that with probability at least $1 - 2^{-\Omega(k)}$,*

$$\|y - x^*\|_T \le (9 + \varepsilon)\|g\|_T + \delta\|x^*\|_T.$$

*Our algorithm uses $\mathrm{poly}(k, \varepsilon^{-1}, \log(1/\delta)) \log(FT)$ samples and $\mathrm{poly}(k, \varepsilon^{-1}, \log(1/\delta)) \cdot \log^2(FT)$ time. The output $y$ is $\mathrm{poly}(k, \log(1/\delta))\varepsilon^{-1.5}$-Fourier-sparse signal.*

*Proof.* Let $\mathcal{N}^2 := \|g(t)\|_T^2 + \delta\|x^*(t)\|_T^2$ be the heavy cluster parameter.

First, by Lemma H.14, there is a set of frequencies $S \subset [k]$ and $x_S(t) = \sum_{j \in S} v_j e^{2\pi \mathbf{i} f_j t}$ such that

$$\|x_S - x^*\|_T \le (3 + O(\varepsilon))\mathcal{N}. \tag{25}$$

Furthermore, each $f_j$ with $j \in S$ belongs to an $\mathcal{N}$-heavy cluster $C_j$ with respect to the filter function $H$ defined in Definition H.6.

By Definition H.11 of heavy cluster, it holds that

$$\int_{C_j} |\widehat{H \cdot x^*}(f)|^2 \mathrm{d}f \ge T\mathcal{N}^2/k.$$

By Definition H.11, we also have $|C_j| \le k \cdot \Delta_h$, where $\Delta_h$ is the bandwidth of $\widehat{H}$.

Let $\Delta \in \mathbb{R}_+$, and $\Delta > k \cdot \Delta_h$, which implies that $C_j \subseteq [f_j - \Delta, f_j + \Delta]$. Thus, we have

$$\int_{f_j - \Delta}^{f_j + \Delta} |\widehat{H \cdot x^*}(f)|^2 \mathrm{d}f \ge T\mathcal{N}^2/k.$$

Now it is enough to recover only $x_S$, instead of $x^*$.

By applying Theorem H.36, there is an algorithm that outputs a set of frequencies $L \subset \mathbb{R}$ such that, $|L| = O(k)$, and with probability at least $1 - 2^{-\Omega(k)}$, for any $f_j$ with $j \in S_f$, there is a $\widetilde{f} \in L$ such that,

$$|f_j - \widetilde{f}| \lesssim \Delta\sqrt{\Delta T}.$$

We define a map $p : \mathbb{R} \to L$ as follows:

$$p(f) := \arg\min_{\widetilde{f} \in L} |f - \widetilde{f}| \quad \forall f \in \mathbb{R}.$$

Then, $x_S(t)$ can be expressed as

$$
\begin{aligned}
x_{S_f}(t) &= \sum_{j \in S_f} v_j e^{2\pi \mathbf{i} f_j t} \\
&= \sum_{j \in S_f} v_j e^{2\pi \mathbf{i} \cdot p(f_j) t} \cdot e^{2\pi \mathbf{i} \cdot (f_j - p(f_j)) t} \\
&= \sum_{\widetilde{f} \in L} e^{2\pi \mathbf{i} \widetilde{f} t} \cdot \sum_{j \in S_f : \, p(f_j) = \widetilde{f}} v_j e^{2\pi \mathbf{i} (f_j - \widetilde{f}) t},
\end{aligned}
$$

where the first step follows from the definition of $x_S$, the last step follows from interchanging the summations.

For each $\widetilde{f}_i \in L$, by Corollary H.2 with $x^* = x_{S_f}, \Delta = \Delta\sqrt{\Delta T}$, we have that there exist degree $d = O(T\Delta\sqrt{\Delta T} + k^3 \log k + k \log 1/\delta)$ polynomials $P_i(t)$ corresponding to $\widetilde{f}_i \in L$ such that,

$$\|x_{S_f}(t) - \sum_{\widetilde{f}_i \in L} e^{2\pi \mathbf{i} \widetilde{f}_i t} P_i(t)\|_T \leq \delta \|x_{S_f}(t)\|_T \tag{26}$$

Define the following function family:

$$\mathcal{F} := \operatorname{span}\Big\{ e^{2\pi \mathbf{i} \widetilde{f} t} \cdot t^j \mid \forall \widetilde{f} \in L, j \in \{0, 1, \ldots, d\} \Big\}.$$

Note that $\sum_{\widetilde{f}_i \in L} e^{2\pi \mathbf{i} \widetilde{f}_i t} P_i(t) \in \mathcal{F}$.

By Claim H.16, for function family $\mathcal{F}$, $K_{\mathrm{Uniform}[0,\mathrm{T}]} = O((|L|d)^4 \log^3(|L|d))$.

By Lemma H.18, we have that, choosing a set $W$ of $O(\varepsilon^{-1} K_{\mathrm{Uniform}[0,\mathrm{T}]} \log(|L|d/\rho))$ i.i.d. samples uniformly at random over duration $[0, T]$ is a $(\varepsilon, \rho)$-WBSP.

By Lemma H.19, there is an algorithm that runs in $O(\varepsilon^{-1} |W| (|L|d)^{\omega - 1} \log(1/\rho))$-time using samples in $W$, and outputs $y'(t) \in \mathcal{F}$ such that, with probability $1 - \rho$,

$$\|y'(t) - \sum_{\widetilde{f}_i \in L} e^{2\pi \mathbf{i} \widetilde{f}_i t} P_i(t)\|_T \leq (1 + \varepsilon) \|x(t) - \sum_{\widetilde{f}_i \in L} e^{2\pi \mathbf{i} \widetilde{f}_i t} P_i(t)\|_T \tag{27}$$

Then by Lemma H.3, we have that there is a $O(kd)$-Fourier-sparse signal $y(t)$, such that

$$\|y(t) - y'(t)\|_T \leq \delta' \tag{28}$$

where $\delta' > 0$ is any positive real number, thus, $y$ can be arbitrarily close to $y'$.

Moreover, the sparsity of $y(t)$ is $kd = kO(T\Delta\sqrt{\Delta T} + k^3 \log k + k \log 1/\delta) = \varepsilon^{-1.5} \operatorname{poly}(k, \log(1/\delta))$.

Therefore, the total approximation error can be upper bounded as follows:

$$
\begin{aligned}
&\|y - x^*\|_T \\
&\leq \|y - y'\|_T + \Big\| y' - \sum_{\widetilde{f}_i \in L} e^{2\pi \mathbf{i} \widetilde{f}_i t} P_i(t) \Big\|_T + \Big\| \sum_{\widetilde{f}_i \in L} e^{2\pi \mathbf{i} \widetilde{f}_i t} P_i(t) - x^* \Big\|_T && \text{(Triangle inequality)} \\
&\leq (1 + o(1)) \Big\| y - \sum_{\widetilde{f}_i \in L} e^{2\pi \mathbf{i} \widetilde{f}_i t} P_i(t) \Big\|_T + \Big\| \sum_{\widetilde{f}_i \in L} e^{2\pi \mathbf{i} \widetilde{f}_i t} P_i(t) - x^* \Big\|_T && \text{(Eq. (28))} \\
&\leq (1 + \varepsilon) \Big\| x - \sum_{\widetilde{f}_i \in L} e^{2\pi \mathbf{i} \widetilde{f}_i t} P_i(t) \Big\|_T + \Big\| \sum_{\widetilde{f}_i \in L} e^{2\pi \mathbf{i} \widetilde{f}_i t} P_i(t) - x^* \Big\|_T && \text{(Eq. (27))}
\end{aligned}
$$

$$\leq (1+2\varepsilon)\|g\|_T + (2+\varepsilon)\Big\|\sum_{\widetilde{f}_i \in L} e^{2\pi \mathbf{i}\widetilde{f}_i t} P_i(t) - x^*\Big\|_T \qquad \text{(Triangle inequality)}$$

$$\leq (1+2\varepsilon)\|g\|_T + (2+\varepsilon)\Big\|\sum_{\widetilde{f}_i \in L} e^{2\pi \mathbf{i}\widetilde{f}_i t} P_i(t) - x_{S_f}\Big\|_T + (2+\varepsilon)\|x_{S_f} - x^*\|_T$$
$$\text{(Triangle inequality)}$$

$$\leq (1+2\varepsilon)\|g\|_T + (2+\varepsilon)\delta\|x_{S_f}\|_T + (2+\varepsilon)\|x_{S_f} - x^*\|_T \qquad \text{(Eq. (26))}$$

$$\leq (1+2\varepsilon)\|g\|_T + O(\delta)\|x^*\|_T + (2+\varepsilon)(1+\delta)\|x_{S_f} - x^*\|_T \qquad \text{(Triangle inequality)}$$

$$\leq (1+2\varepsilon)\|g\|_T + O(\delta)\|x^*\|_T + (2+\varepsilon)(1+\delta)(\|x_{S_f} - x_S\|_T + \|x_S - x^*\|_T)$$
$$\text{(Triangle inequality)}$$

$$\leq (1+2\varepsilon)\|g\|_T + O(\delta)\|x^*\|_T + (2+\varepsilon+O(\delta))(4+O(\varepsilon))\mathcal{N} \qquad \text{(Eq. (25) and Lemma H.41)}$$

$$= (1+2\varepsilon)\|g\|_T + O(\delta)\|x^*\|_T + (8+O(\varepsilon+\delta))\mathcal{N},$$

Since we take

$$\mathcal{N} = \sqrt{\|g\|_T^2 + \delta\|x^*\|_T^2} \leq \|g\|_T + \sqrt{\delta}\|x^*\|_T,$$

we have

$$\|y - x^*\|_T \leq (9+O(\varepsilon))\|g\|_T + O(\sqrt{\delta})\|x^*\|_T.$$

By re-scaling $\varepsilon$ and $\delta$, we prove the theorem.

$\qquad \square$

## H.5   Sharper error control by signal-noise cancellation effect

In this section, we significantly improve the error analysis in Section H.3. Our key observation is the *signal-noise cancellation effect*: if there is a frequency $f^*$ in a $\mathcal{N}_1$-heavy cluster but not $(\mathcal{N}_1, \mathcal{N}_2)$-recoverable for some $\mathcal{N}_2 < \mathcal{N}_1$, then it indicates that the contribution of $f^*$ to the signal $x^*$'s energy are cancelled out by the noise $g$.

In the following lemma, we improving Lemma H.14 by considering $g$'s effect in the gap between heavy-cluster signal and recoverable signal.

**Lemma H.21** (Sharper error bound for recoverable signal, an improved version of Lemma H.14)**.**
*Let* $x^*(t) = \sum_{j=1}^k v_j e^{2\pi \mathbf{i}f_j t}$ *and* $x(t) = x^*(t) + g(t)$ *be our observable signal. Let* $\mathcal{N}_1^2 := \|g(t)\|_T^2 + \delta\|x^*(t)\|_T^2$. *Let* $C_1, \cdots, C_l$ *are the* $\mathcal{N}_1$-*heavy clusters from Definition H.11. Let* $S^*$ *denotes the set of frequencies* $f^* \in \{f_j\}_{j \in [k]}$ *such that,* $f^* \in C_i$ *for some* $i \in [l]$. *Let* $S \subset S^*$ *be the set of* $(\mathcal{N}_1, \sqrt{\varepsilon_2}\mathcal{N}_1)$-*recoverable frequencies (Definition H.13).*

*Then we have that,*

$$\|H \cdot x_{S^*} - H \cdot x_S\|_T^2 + \|H \cdot x - H \cdot x_S\|_T^2 \leq (1+O(\sqrt{\varepsilon_2}))\|x - x_{S^*}\|_T^2.$$

*Proof.* Let $g'(t) := g(t) + x^*(t) - x_{S^*}(t) = x(t) - x_{S^*}(t)$.

In order for cluster $C_i$ to be missed, we must have that

$$\int_{C_i} |\widehat{H \cdot x_{S^*}}(f)|^2 \mathrm{d}f \geq T\mathcal{N}_1^2/k \geq \frac{1}{\varepsilon_2}\int_{C_i} |\widehat{H \cdot x}(f)|^2 \mathrm{d}f \qquad (29)$$

where the first steps follows from $C_i \subset \cup_{f_j \in S^*} C_j$, the second step follows from $C_i \not\subset \cup_{f_j \in S} C_j$.

Thus,

$$\int_{C_i} |\widehat{H \cdot g'}(f)|^2 \mathrm{d}f = \int_{C_i} |\widehat{H \cdot (x - x_{S^*})}(f)|^2 \mathrm{d}f$$

$$\geq \left(\sqrt{\int_{C_i} |\widehat{H \cdot x_{S^*}}(f)|^2 \mathrm{d}f} - \sqrt{\int_{C_i} |\widehat{H \cdot x}(f)|^2 \mathrm{d}f}\right)^2$$

$$\geq (\frac{1}{\sqrt{\varepsilon_2}} - 1)^2 \int_{C_i} |\widehat{H \cdot x}(f)|^2 \mathrm{d}f$$

$$\geq \frac{1}{2\varepsilon_2} \int_{C_i} |\widehat{H \cdot x}(f)|^2 \mathrm{d}f, \tag{30}$$

where the first step follows from the definition of $g'$, the second step follows from triangle inequality, the third step follows from Eq. (29), the last step follows from $\varepsilon_2 \leq 0.1$.

**Bound** $\|H \cdot x - H \cdot x_S\|_T$. Let $I' = \cup_{f_j \in S^* \setminus S} C_j$, then we have that,

$$T\|H \cdot x - H \cdot x_S\|_T^2 \leq \int_{-\infty}^{\infty} |H \cdot x(t) - H \cdot x_S(t)|^2 \mathrm{d}t$$

$$= \int_{-\infty}^{\infty} |(\widehat{H \cdot x} - \widehat{H \cdot x_S})(f)|^2 \mathrm{d}f$$

$$= \int_{I'} |(\widehat{H \cdot x} - \widehat{H \cdot x_S})(f)|^2 \mathrm{d}f + \int_{\overline{I'}} |(\widehat{H \cdot x} - \widehat{H \cdot x_S})(f)|^2 \mathrm{d}f \tag{31}$$

where the first step follows from the definition of the norm, the second step follows from Parseval's theorem, the third step follows from $I' \cup \overline{I'} = [-\infty, \infty]$.

**Bound** $\|H \cdot x_{S^*} - H \cdot x_S\|_T$ We can upper-bound it as follows:

$$T\|H \cdot x_{S^*} - H \cdot x_S\|_T^2 \leq \int_{-\infty}^{\infty} |H \cdot x_{S^*}(t) - H \cdot x_S(t)|^2 \mathrm{d}t$$

$$= \int_{-\infty}^{\infty} |(\widehat{H \cdot x_{S^*}} - \widehat{H \cdot x_S})(f)|^2 \mathrm{d}f$$

$$= \int_{I'} |(\widehat{H \cdot x_{S^*}} - \widehat{H \cdot x_S})(f)|^2 \mathrm{d}f + \int_{\overline{I'}} |(\widehat{H \cdot x_{S^*}} - \widehat{H \cdot x_S})(f)|^2 \mathrm{d}f$$

$$= \int_{I'} |(\widehat{H \cdot x_{S^*}} - \widehat{H \cdot x_S})(f)|^2 \mathrm{d}f \tag{32}$$

where the first step follows from the definition of the norm, the second step follows from Parseval's theorem, the third step follows from $I' \cup \overline{I'} = [-\infty, \infty]$, the last step follows from $(\cup_{f_j \in S^*/S} C_j) \cap \overline{I'} = \emptyset$.

**Putting it all together.** By Eqs. (31) and (32), we get that

$$T\|H \cdot x_{S^*} - H \cdot x_S\|_T^2 + T\|H \cdot x - H \cdot x_S\|_T^2$$

$$\leq \int_{I'} |(\widehat{H \cdot x_{S^*}} - \widehat{H \cdot x_S})(f)|^2 \mathrm{d}f + \int_{I'} |(\widehat{H \cdot x} - \widehat{H \cdot x_S})(f)|^2 \mathrm{d}f + \int_{\overline{I'}} |(\widehat{H \cdot x} - \widehat{H \cdot x_S})(f)|^2 \mathrm{d}f.$$

For the first integral, we have

$$\sqrt{\int_{I'} |(\widehat{H \cdot x_{S^*}} - \widehat{H \cdot x_S})(f)|^2 \mathrm{d}f} = \sqrt{\int_{I'} |\widehat{H \cdot x_{S^*}}(f)|^2 \mathrm{d}f}$$

$$\leq \sqrt{\int_{I'} |\widehat{H \cdot x}(f)|^2 \mathrm{d}f} + \sqrt{\int_{I'} |\widehat{H \cdot g'}(f)|^2 \mathrm{d}f}$$

$$\leq \sqrt{2\varepsilon_2 \int_{I'} |\widehat{H \cdot g'}(f)|^2 \mathrm{d}f} + \sqrt{\int_{I'} |\widehat{H \cdot g'}(f)|^2 \mathrm{d}f}$$

$$\leq (1 + \sqrt{2\varepsilon_2}) \sqrt{\int_{I'} |\widehat{H \cdot g'}(f)|^2 \mathrm{d}f}, \tag{33}$$

where the first step follows from $(\cup_{f_j \in S} C_j) \cap I' = \emptyset$, the second step follows from triangle inequality, the third step follows from Eq. (30), the last step is straightforward.

For the second integral, we have

$$\int_{I'} |(\widehat{H \cdot x} - \widehat{H \cdot x_S})(f)|^2 \mathrm{d}f = \int_{I'} |\widehat{H \cdot x}(f)|^2 \mathrm{d}f$$

$$\leq 2\varepsilon_2 \int_{I'} |\widehat{H \cdot g'}(f)|^2 \mathrm{d}f, \tag{34}$$

where the first step follows from $(\cup_{f_j \in S} C_j) \cap I' = \emptyset$, the second step follows from Eq. (30).

For the third integral, together with the $\int_{I'} |\widehat{H \cdot g'}(f)|^2 \mathrm{d}f$ term in the first integral's upper bound (Eq. (33)), we have

$$
\begin{aligned}
&\int_{\overline{I'}} |(\widehat{H \cdot x} - \widehat{H \cdot x_S})(f)|^2 \mathrm{d}f + \int_{I'} |\widehat{H \cdot g'}(f)|^2 \mathrm{d}f \\
&= \int_{\overline{I'}} |H \cdot (x_{S^*} \widehat{+ g'} - x_S)(f)|^2 \mathrm{d}f + \int_{I'} |\widehat{H \cdot g'}(f)|^2 \mathrm{d}f \\
&= \int_{\overline{I'}} |\widehat{H \cdot g'}(f)|^2 \mathrm{d}f + \int_{I'} |\widehat{H \cdot g'}(f)|^2 \mathrm{d}f \\
&= \int_{-\infty}^{\infty} |\widehat{H \cdot g'}(f)|^2 \mathrm{d}f \\
&= \int_{-\infty}^{\infty} |H \cdot g'(t)|^2 \mathrm{d}t \\
&= T\|H \cdot g'(t)\|_T^2 \\
&\leq T\|g'\|_T^2, \tag{35}
\end{aligned}
$$

where the first step follows from the definition of $g'$, the second step follows from $(\cup_{f_j \in S^*} C_j) \cap \overline{I'} = (\cup_{f_j \in S} C_j)$, the third step follows from $I' \cup \overline{I'} = [-\infty, \infty]$, the forth step follows from Parseval's theorem, the fifth step follows from $g'(t) = 0, \forall t \notin [0, T]$, the last step follows from $H(t) \leq 1$ by Remark H.8.

Furthermore, we have that

$$\int_{I'} |\widehat{H \cdot g'}(f)|^2 \mathrm{d}f \leq \int_{-\infty}^{\infty} |\widehat{H \cdot g'}(f)|^2 \mathrm{d}f \leq T\|g'(t)\|_T^2. \tag{36}$$

Therefore, we conclude that

$$
\begin{aligned}
&T\|H \cdot x_{S^*} - H \cdot x_S\|_T^2 + T\|H \cdot x - H \cdot x_S\|_T^2 \\
&\leq T\|H \cdot x_{S^*} - H \cdot x_S\|_T^2 + \int_{I'} |(\widehat{H \cdot x} - \widehat{H \cdot x_S})(f)|^2 \mathrm{d}f + \int_{\overline{I'}} |(\widehat{H \cdot x} - \widehat{H \cdot x_S})(f)|^2 \mathrm{d}f \\
&\leq \int_{I'} |(\widehat{H \cdot x_{S^*}} - \widehat{H \cdot x_S})(f)|^2 \mathrm{d}f + \int_{I'} |(\widehat{H \cdot x} - \widehat{H \cdot x_S})(f)|^2 \mathrm{d}f + \int_{\overline{I'}} |(\widehat{H \cdot x} - \widehat{H \cdot x_S})(f)|^2 \mathrm{d}f \\
&\leq (1 + \sqrt{\varepsilon_2})^2 \int_{I'} |\widehat{H \cdot g'}(f)|^2 \mathrm{d}f + \int_{I'} |(\widehat{H \cdot x} - \widehat{H \cdot x_S})(f)|^2 \mathrm{d}f + \int_{\overline{I'}} |(\widehat{H \cdot x} - \widehat{H \cdot x_S})(f)|^2 \mathrm{d}f \\
&\leq (1 + \sqrt{\varepsilon_2})^2 \int_{I'} |\widehat{H \cdot g'}(f)|^2 \mathrm{d}f + 2\varepsilon_2 \int_{I'} |\widehat{H \cdot g'}(f)|^2 \mathrm{d}f + \int_{\overline{I'}} |(\widehat{H \cdot x} - \widehat{H \cdot x_S})(f)|^2 \mathrm{d}f \\
&= O(\sqrt{\varepsilon_2}) \int_{I'} |\widehat{H \cdot g'}(f)|^2 \mathrm{d}f + \int_{I'} |\widehat{H \cdot g'}(f)|^2 \mathrm{d}f + \int_{\overline{I'}} |(\widehat{H \cdot x} - \widehat{H \cdot x_S})(f)|^2 \mathrm{d}f \\
&\leq O(\sqrt{\varepsilon_2})T\|g'\|_T^2 + \int_{I'} |\widehat{H \cdot g'}(f)|^2 \mathrm{d}f + \int_{\overline{I'}} |(\widehat{H \cdot x} - \widehat{H \cdot x_S})(f)|^2 \mathrm{d}f \\
&\leq O(\sqrt{\varepsilon_2})T\|g'\|_T^2 + T\|g'\|_T^2 \\
&= (1 + O(\sqrt{\varepsilon_2}))T\|g'\|_T^2
\end{aligned}
$$

where the first step follows from Eq. (31), the second step follows from Eq. (32), the third step follows from Eq. (33), the forth step follows from Eq. (34), the fifth step follows from $(1 + \sqrt{2\varepsilon_2})^2 \leq 1 + O(\sqrt{\varepsilon_2})$, the sixth step follows from Eq. (36), the seventh step follows from Eq. (35), the last step is straightforward.

The lemma is then proved. $\qquad\square$

As a consequence, we can easily bound $\|x_{S^*} - x_S\|_T$ as follows.

**Corollary H.22.** *Let $S^*$ and $S$ be defined as in Lemma H.21. Then, we have that,*

$$\|x_{S^*} - x_S\|_T^2 \leq (1 + O(\sqrt{\varepsilon_2}))\|x - x_{S^*}\|_T^2$$

*Proof.* We have that,

$$\|x_{S^*} - x_S\|_T^2 \leq (1 + 2\varepsilon)\|H \cdot x_{S^*} - H \cdot x_S\|_T^2 \leq (1 + 2\varepsilon)(1 + O(\sqrt{\varepsilon_2}))\|x - x_{S^*}\|_T^2$$

where the first step follows from Lemma H.7 Property VI, the second step follows from Lemma H.21 and $\varepsilon = \varepsilon_2$. $\qquad\square$

In Lemma H.21, we introduce an extra term $\|H \cdot x - H \cdot x_S\|_T$. The following lemma shows that this term appears in the approximation error $\|x - x_S\|_T$, which can be used to upper-bound the Signal Estimation's error.

**Lemma H.23** (Decomposing the approximation error of recoverable signal)**.** *Let $x^*(t) = \sum_{j=1}^{k} v_j e^{2\pi \mathbf{i} f_j t}$ and $x(t) = x^*(t) + g(t)$ be our observable signal. Let $\mathcal{N}_1^2 := \|g(t)\|_T^2 + \delta\|x^*(t)\|_T^2$. Let $C_1, \cdots, C_l$ are the $\mathcal{N}_1$-heavy clusters from Definition H.11. Let $S^*$ denotes the set of frequencies $f^* \in \{f_j\}_{j \in [k]}$ such that, $f^* \in C_i$ for some $i \in [l]$, and*

$$\int_{C_i} |\widehat{x^* \cdot H}(f)|^2 \mathrm{d}f \geq T\mathcal{N}_1^2/k,$$

*Let $S$ denotes the set of frequencies $f^* \in S^*$ such that, $f^* \in C_j$ for some $j \in [l]$, and*

$$\int_{C_j} |\widehat{x \cdot H}(f)|^2 \mathrm{d}f \geq \varepsilon_2 T\mathcal{N}_1^2/k,$$

*Then we have that,*

$$\|x - x_S\|_T \leq \|H(x - x_S)\|_T + \|g\|_T + O(\varepsilon)\|x^* - x_S\|_T.$$

*Proof.* We first decompose $\|x - x_S\|_T$ into the part that passes through the filter $H$ and the part that does not pass through $H$:

$$\begin{aligned}
\|x - x_S\|_T^2 &\leq \|H(x - x_S)\|_T^2 + \|(1 - H)(x - x_S)\|_T^2 \\
&\leq \|H(x - x_S)\|_T^2 + \|(1 - H)(x - x^*)\|_T^2 + \|(1 - H)(x^* - x_S)\|_T^2 \\
&\leq \|H(x - x_S)\|_T^2 + \|(1 - H)g\|_T^2 + \|(1 - H)(x^* - x_S)\|_T^2,
\end{aligned}$$

where the first step follows from triangle inequality, the second step follows from triangle inequality, the last step follows from the definition of $g$.

For the second term, we have that

$$\|(1 - H)g\|_T^2 \leq \|g\|_T^2,$$

by Remark H.8.

For the third term, we have that,

$$\|(1 - H)(x^* - x_S)\|_T^2 = \|x^* - x_S\|_T^2 - \|H(x^* - x_S)\|_T^2 \leq \varepsilon\|x^* - x_S\|_T^2,$$

where the first step follows from $1 - H > 0$, the second step follows from $x^* - x_S$ is $k$-Fourier-sparse, thus combine Property VI of Lemma H.7, we have that $\|H(x^* - x_S)\|_T^2 \geq (1 - \varepsilon)\|x^* - x_S\|_T^2$.

Combining them together, we prove the lemma. $\qquad\square$

## H.6 Technical tools III: HASHTOBINS

In this section, we provide some definitions and technical lemmas for the HASHTOBINS procedure, which will be very helpful for frequency estimation.

HASHTOBINS partitions the frequency coordinates into $B = O(k)$ bins and collects rotated magnitudes in each bins. Ideally, each bins only contains a single ground-truth frequency, which allows us to recover its magnitude.

More specifically, HASHTOBINS first randomly hashes the frequency coordinates into the interval $[0, 1]$. After equally dividing $[0, 1]$ into $O(k)$ small bins, each coordinate lays in a different bin. This step can be implemented by multiplying the signal in the frequency domain with a *period pulse function* $G_{\sigma,b}^{(j)}$. Then, even if the signal does not have frequency gap, the HASHTOBINS procedure can still partition it into several one-cluster signals with high probability.

**Definition H.24** (Hash function, Chen et al. (2016)). *Let $\pi_{\sigma,b}(f) = \sigma(f+b) \pmod 1$ and $h_{\sigma,b}(f) =$ round$(\pi_{\sigma,b}(f) \cdot B)$ be the hash function that maps frequency $f \in [-F, F]$ into bins $\{0, \cdots, B-1\}$.*

**Claim H.25** (Collision probability, Chen et al. (2016)). *For any $\Delta_0 > 0$, let $\sigma$ be a sample uniformly at random from $[\frac{1}{4B\Delta_0}, \frac{1}{2B\Delta_0}]$. Then, we have:*

*I. If $4\Delta_0 \leq |f^+ - f^-| < 2(B-1)\Delta_0$, then $\Pr[h_{\sigma,b}(f^+) = h_{\sigma,b}(f^-)] = 0$.*

*II. If $2(B-1)\Delta_0 \leq |f^+ - f^-|$, then $\Pr[h_{\sigma,b}(f^+) = h_{\sigma,b}(f^-)] \lesssim \frac{1}{B}$.*

**Definition H.26** (Filter for bins). *Given $B > 1$, $\delta > 0$, $\alpha > 0$, let $G(t) := G_{B,\delta,\alpha}(2\pi t)$ where $G_{B,\delta,\alpha}$ is defined in Definition H.9. For any $\sigma > 0, b \in \mathbb{R}$ and $j \in [B]$. define*

$$G_{\sigma,b}^{(j)}(t) := \frac{1}{\sigma} G(t/\sigma) e^{2\pi \mathbf{i} t (j/B - \sigma b)/\sigma},$$

*and its Fourier transformation:*

$$\widehat{G}_{\sigma,b}^{(j)}(f) = \sum_{i \in \mathbb{Z}} \widehat{G}(i + \frac{j}{B} - \sigma f - \sigma b).$$

**Definition H.27** (($\varepsilon_0, \Delta_0$)-one-cluster signal, Chen et al. (2016)). *We say that a signal $z(t)$ is an $(\varepsilon_0, \Delta_0)$-one-cluster signal around $f_0$ iff $z(t)$ and $\widehat{z}(f)$ satisfy the following two properties:*

$$\text{Property I} \quad : \quad \int_{f_0 - \Delta_0}^{f_0 + \Delta_0} |\widehat{z}(f)|^2 \mathrm{d}f \geq (1 - \varepsilon_0) \int_{-\infty}^{+\infty} |\widehat{z}(f)|^2 \mathrm{d}f$$

$$\text{Property II} \quad : \quad \int_0^T |z(t)|^2 \mathrm{d}t \geq (1 - \varepsilon_0) \int_{-\infty}^{+\infty} |z(t)|^2 \mathrm{d}t.$$

**Definition H.28** (Well-isolation, Chen et al. (2016)). *We say that a frequency $f^*$ is well-isolated under the hashing $(\sigma, b)$ if, for $j = h_{\sigma,b}(f^*)$ and $\overline{I_{f^*}} = (-\infty, \infty) \setminus (f^* - \Delta_0, f^* + \Delta_0)$,*

$$\int_{\overline{I_{f^*}}} |(\widehat{H \cdot x} \cdot \widehat{G}_{\sigma,b}^{(j)})(f)|^2 \mathrm{d}f \lesssim \varepsilon_0 \cdot T \mathcal{N}_2^2 / k,$$

*where $\mathcal{N}_2^2 := \varepsilon_1 \varepsilon_2 (\|g(t)\|_T^2 + \delta \|x^*(t)\|_T^2)$.*

**Lemma H.29** (Well-isolation implies one-cluster signal, a variation of Lemma 7.20 in Chen et al. (2016)). *Let $f^*$ satisfy*

$$\int_{f^* - \Delta}^{f^* + \Delta} |\widehat{x^* \cdot H}(f)|^2 \mathrm{d}f \geq T \mathcal{N}_2^2 / k,$$

*where $\mathcal{N}_2^2 := \varepsilon_1 \varepsilon_2 (\|g(t)\|_T^2 + \delta \|x^*(t)\|_T^2)$. Let $\widehat{z} = \widehat{x^* \cdot H} \cdot \widehat{G}_{\sigma,b}^{(j)}$ where $j = h_{\sigma,b}(f^*)$. If $f^*$ is well-isolated, then $z$ and $\widehat{z}$ satisfying Property II of one-cluster signal (Definition H.27), i.e.,*

$$\int_0^T |z(t)|^2 \mathrm{d}t \geq (1 - \varepsilon_0) \int_{-\infty}^{+\infty} |z(t)|^2 \mathrm{d}t,$$

**Lemma H.30** (Well-isolation by randomized hashing, Chen et al. (2016)). *Given $B = \Theta(k/(\varepsilon_0 \varepsilon_1 \varepsilon_2))$ and $\sigma \in [\frac{1}{4B\Delta_0}, \frac{1}{2B\Delta_0}]$ chosen uniformly at random. Let $f^*$ be any frequency. Then $f^*$ is well-isolated by a hashing $(\sigma, b)$ with probability at least $0.9$.*

*Proof.* Let $S' = \{f_i\}_{i \in [k]} \cap \overline{I_{f^*}}$. By Claim H.25, with probability at least $(1 - 1/B)^k \geq 1 - k/B \geq 1 - \varepsilon_0 \varepsilon_1 \varepsilon_2 \geq 0.99$, for all the frequencies $f \in S'$, we have that $h_{\sigma,b}(f^*) \neq h_{\sigma,b}(f)$.

Hence,

$$
\begin{aligned}
\int_{\overline{I_{f^*}}} |\widehat{x^* \cdot H} \cdot \widehat{G}_{\sigma,b}^{(j)}(f)|^2 \mathrm{d}f &\lesssim \frac{\delta^2}{k^2} \int_{\overline{I_{f^*}}} |\widehat{x^* \cdot H}(f)|^2 \mathrm{d}f \\
&\leq \frac{\delta^2}{k^2} \int_{-\infty}^{\infty} |\widehat{x^* \cdot H}(f)|^2 \mathrm{d}f \\
&= \frac{\delta^2}{k^2} \int_{-\infty}^{\infty} |x^* \cdot H(t)|^2 \mathrm{d}t \\
&= \frac{\delta^2}{k^2} \int_{[-\infty,\infty] \setminus [0,T]} |x^* \cdot H(t)|^2 \mathrm{d}t + \frac{\delta^2}{k^2} \int_{[0,T]} |x^* \cdot H(t)|^2 \mathrm{d}t \\
&\leq \frac{\delta^2}{k^2} \int_{[-\infty,\infty] \setminus [0,T]} |x^* \cdot H(t)|^2 \mathrm{d}t + \frac{\delta^2}{k^2} T \|x^*\|_T^2 \\
&\leq \frac{\delta^2(1+\delta)}{k^2} T \|x^*\|_T^2
\end{aligned}
\tag{37}
$$

where the first step follows by the Property III in the Lemma H.10 that $|\widehat{G}(f)| \leq \delta/k$, which implies that $|\widehat{G}_{\sigma,b}^{(j)}(f)| \leq O(\delta/k)$ for $f \in S'$, the second step follows from $\overline{I_{f^*}} \subset [-\infty, \infty]$, the third step follows from Parseval's theorem, the forth step is straight forward, the fifth step follows from the property VI of Lemma H.7, the sixth step follows from V of Lemma H.7.

Moreover, let $I'$ denote the set of frequencies that hash into the same bin as $f^*$, then we have that,

$$
\begin{aligned}
\int_{\overline{I_{f^*}}} |\widehat{g \cdot H} \cdot \widehat{G}_{\sigma,b}^{(j)}(f)|^2 \mathrm{d}f &\leq \int_{I'} |\widehat{g \cdot H} \cdot \widehat{G}_{\sigma,b}^{(j)}(f)|^2 \mathrm{d}f + \int_{\overline{I'}} |\widehat{g \cdot H} \cdot \widehat{G}_{\sigma,b}^{(j)}(f)|^2 \mathrm{d}f \\
&\lesssim \int_{I'} |\widehat{g \cdot H}(f)|^2 \mathrm{d}f + \int_{\overline{I'}} |\widehat{g \cdot H} \cdot \widehat{G}_{\sigma,b}^{(j)}(f)|^2 \mathrm{d}f \\
&\lesssim \int_{I'} |\widehat{g \cdot H}(f)|^2 \mathrm{d}f + \frac{\delta^2}{k^2} \int_{\overline{I'}} |\widehat{g \cdot H}(f)|^2 \mathrm{d}f \\
&\leq \int_{I'} |\widehat{g \cdot H}(f)|^2 \mathrm{d}f + \frac{\delta^2 T}{k^2} \|g\|_T^2
\end{aligned}
\tag{38}
$$

where the first step follows from $I' \cup \overline{I'} = [-\infty, \infty]$, the second step follows from for any $f \in \mathbb{R}$, $\widehat{G}_{\sigma,b}^{(j)}(f) \lesssim 1$, the third step follows from for any $f \in \overline{I'}$, $\widehat{G}_{\sigma,b}^{(j)}(f) \lesssim \delta/k$, the last step follows from

$$
\int_{\overline{I'}} |\widehat{g \cdot H}(f)|^2 \mathrm{d}f \leq \int_{-\infty}^{\infty} |\widehat{g \cdot H}(f)|^2 \mathrm{d}f = \int_{-\infty}^{\infty} |g \cdot H(t)|^2 \mathrm{d}t = T \|g \cdot H\|_T^2 \leq T \|g\|_T^2.
$$

where the first step follows from $\overline{I'} \in [-\infty, \infty]$, the second step follows from Parseval's theorem, the third step follows from $g(t) = 0, \forall t \notin [0, T]$, the last step follows from Remark H.8.

Next, we consider

$$
\begin{aligned}
\mathop{\mathbb{E}}_{\sigma,b} \left[ \int_{I'} |\widehat{g \cdot H}(f)|^2 \mathrm{d}f \right] &\approx \frac{1}{B} \int_{-\infty}^{\infty} |\widehat{g \cdot H}(f)|^2 \mathrm{d}f \\
&\lesssim \frac{\varepsilon_0 \varepsilon_1 \varepsilon_2}{k} T \|g\|_T^2
\end{aligned}
$$

where the first step follows from $\sigma, b$ are chosen randomly, the second step follows from $\int_{-\infty}^{\infty} |\widehat{g \cdot H}(f)|^2 \mathrm{d}f \leq T \|g\|_T^2$.

Thus, by Markov inequality, with probability at least $0.99$,

$$
\int_{I'} |\widehat{g \cdot H}(f)|^2 \mathrm{d}f \lesssim \frac{\varepsilon_0 \varepsilon_1 \varepsilon_2}{k} T \|g\|_T^2.
\tag{39}
$$

Finally, we can conclude that

$$
\begin{aligned}
\int_{I_{f^*}} |(\widehat{H \cdot x} \cdot \widehat{G}_{\sigma,b}^{(j)})(f)|^2 \mathrm{d}f &= \int_{I_{f^*}} |(H \cdot \widehat{(x^* + g)} \cdot \widehat{G}_{\sigma,b}^{(j)})(f)|^2 \mathrm{d}f \\
&\leq 2\int_{I_{f^*}} |\widehat{x^* \cdot H} \cdot \widehat{G}_{\sigma,b}^{(j)}(f)|^2 \mathrm{d}f + 2\int_{I_{f^*}} |\widehat{g \cdot H} \cdot \widehat{G}_{\sigma,b}^{(j)}(f)|^2 \mathrm{d}f \\
&\lesssim \frac{\delta^2(1+\delta)}{k^2} T\|x^*\|_T^2 + 2\int_{I_{f^*}} |\widehat{g \cdot H} \cdot \widehat{G}_{\sigma,b}^{(j)}(f)|^2 \mathrm{d}f \\
&\lesssim \frac{\delta^2(1+\delta)}{k^2} T\|x^*\|_T^2 + \frac{\delta^2 T}{k^2} \|g\|_T^2 + \int_{I'} |\widehat{g \cdot H}(f)|^2 \mathrm{d}f \\
&\lesssim \frac{\delta^2(1+\delta)}{k^2} T\|x^*\|_T^2 + \frac{\delta^2 T}{k^2} \|g\|_T^2 + \frac{\varepsilon_0\varepsilon_1\varepsilon_2}{k} T\|g\|_T^2 \\
&= \frac{\delta(1+\delta)}{\varepsilon_0\varepsilon_1\varepsilon_2 k} \varepsilon_0\varepsilon_1\varepsilon_2 T\delta\|x^*\|_T^2/k + (\frac{\delta^2}{\varepsilon_0\varepsilon_1\varepsilon_2 k} + 1)\varepsilon_0\varepsilon_1\varepsilon_2 T\|g\|_T^2/k \\
&\leq \varepsilon_0\varepsilon_1\varepsilon_2 T\delta\|x^*\|_T^2/k + 2\varepsilon_0\varepsilon_1\varepsilon_2 T\|g\|_T^2/k \\
&\lesssim \varepsilon_0 \cdot T\mathcal{N}_2^2/k,
\end{aligned}
$$

where the first step follows from the definition of $g$, the second step follows from $(a+b)^2 \leq 2a^2 + 2b^2$, the third step follows from Eq. (37), the forth step follows from Eq. (38), the fifth step follows from Eq. (39), the sixth step is straightforward, the seventh step follows from $\frac{\delta(1+\delta)}{\varepsilon_0\varepsilon_1\varepsilon_2 k} \leq 1$ and $(\frac{\delta^2}{\varepsilon_0\varepsilon_1\varepsilon_2 k} + 1) \leq 2$, the last step follows from the definition of $\mathcal{N}_2^2$.

$\square$

**Lemma H.31** ((Chen et al., 2016, Lemma 7.21))**.** *Given any noise $g(t) : [0,T] \to \mathbb{C}$ and $g(t) = 0, \forall t \notin [0,T]$. We have, $\forall j \in [B]$,*

$$
\mathbb{E}_{\sigma,b}\left[\int_{-\infty}^{+\infty} |g(t)H(t) * G_{\sigma,b}^{(j)}(t)|^2 \mathrm{d}t\right] \lesssim \frac{1}{B}\int_{-\infty}^{+\infty} |g(t)H(t)|^2 \mathrm{d}t
$$

### H.7 High signal-to-noise ratio (SNR) band approximation

In the this section, we will give the upper bound of $\|x_{S_f}(t) - x_S(t)\|_T$.

**Definition H.32** (High SNR and Recoverable Set)**.** *For $j \in [B]$, let $z_j^*(t) := (x^* \cdot H) \cdot G_{\sigma,b}^{(j)}$, we define the set as follows*

$$
S_{g_1} := \left\{ j \in [B] \mid \|g_j(t)\|_T^2 \leq (1 - c\varepsilon) \cdot \|z_j^*(t)\|_T^2 \right\}
$$

*where $c$ is constant. And we also give the definition of recoverable set which is the same with $s$ above*

$$
S_{g_2} := \left\{ j \in [B] \mid \exists f_0, h_{\sigma,b}(f_0) = j \text{ and } \int_{f^*-\Delta}^{f^*+\Delta} |\widehat{x \cdot H}(f)|^2 \mathrm{d}f \geq T\mathcal{N}_2^2/k \right\}
$$

*where $\mathcal{N}_2^2 := \varepsilon_1\varepsilon_2(\|g(t)\|_T^2 + \delta\|x^*(t)\|_T^2)$.*

*And then we define a High SNR and recoverable set as follows*

$$
S_g := S_{g_1} \cap S_{g_2}
$$

*Let $S_f := \{j \in [k] \mid h_{\sigma,b}(f_g) \in S_g\} \cap S$. We have $x_{S_f}(t) := \sum_{j \in S_f} v_j e^{2\pi i f_j t}$*

**Remark H.33.** *In the left part of the paper, we focus on the frequency in set $S_f$ which is a subset of the recoverable frequency set $S$.*

The following lemma shows that for any recoverable frequency (i.e., those satisfy Eq. (40)), HASH-TOBINS will output a one-cluster signal around it with high probability. Now we will consider a $f^*$ satisfy the assumption introduced in Definition H.32.

**Lemma H.34** (HASHTOBINS for recoverable and HSR frequency). *Let $f^* \in [-F, F]$ satisfy:*

$$\int_{f^*-\Delta}^{f^*+\Delta} |\widehat{x \cdot H}(f)|^2 \mathrm{d}f \geq T\mathcal{N}_2^2/k, \tag{40}$$

*where $\mathcal{N}_2^2 := \varepsilon_1\varepsilon_2(\|g(t)\|_T^2 + \delta\|x^*(t)\|_T^2)$.*

*For a random hashing $(\sigma, b)$, let $j = h_{\sigma,b}(f^*)$ be the bucket that $f^*$ maps to under the hash such that $z = (x \cdot H) * G_{\sigma,b}^{(j)}$ and $\widehat{z} = \widehat{x \cdot H} \cdot \widehat{G}_{\sigma,b}^{(j)}$. Given that $S_f$ and $c$ is defined in Definition H.32, $j \in S_f$. With probability at least $0.9$, $z(t)$ is an $(\varepsilon_0, \Delta_0)$-one-cluster (See Definition H.27) signal around $f^*$.*

*Proof.* The proof consists of two parts. In part 1, we prove that $z(t)$ satisfies Property I of the one-cluster signal around $f^*$ (Definition H.27). In part 2, we prove that $z(t)$ satisfies Property II of Definition H.27.

**Part 1.** Let region $I_{f^*} = (f^* - \Delta, f^* + \Delta)$ with complement $\overline{I_{f^*}} = (-\infty, \infty) \setminus I_{f^*}$.

Next, with probability at least $0.99$, we have that

$$\int_{I_{f^*}} |\widehat{z}(f)|^2 \mathrm{d}f \geq (1 - \delta/k) \int_{I_{f^*}} |\widehat{x \cdot H}(f)|^2 \mathrm{d}f \gtrsim T\mathcal{N}_2^2/k$$

where the probability follows from $\Delta_0 > 1000\Delta$, the first step follows from Property I of $G$ in Lemma H.10, the second step follows from Eq. (40).

On the other hand, $f^*$ is well-isolated with probability $0.9$, thus by the definition of well-isolated, we have that

$$\int_{\overline{I_{f^*}}} |\widehat{z}(f)|^2 \mathrm{d}f \lesssim \varepsilon_0 T\mathcal{N}_2^2/k.$$

Hence, $\widehat{z}$ satisfies the Property I (in Definition H.27) of one-mountain recovery.

**Part 2.** By Lemma H.29, we know that $(x^* \cdot H) * G_{\sigma,b}^{(j)}$ always satisfies Property II (in Definition H.27):

$$\int_0^T |x^*(t)H(t) * G_{\sigma,b}^{(j)}(t)|^2 \mathrm{d}t \geq (1 - \varepsilon_0) \int_{-\infty}^{+\infty} |x^*(t)H(t) * G_{\sigma,b}^{(j)}(t)|^2 \mathrm{d}t$$

As a result, by $[-\infty, \infty] = [-\infty, 0] \cup [0, T] \cup [T, \infty]$,

$$\varepsilon_0 \int_{-\infty}^{+\infty} |x^*(t)H(t) * G_{\sigma,b}^{(j)}(t)|^2 \mathrm{d}t \geq \int_{-\infty}^0 |x^*(t)H(t) * G_{\sigma,b}^{(j)}(t)|^2 \mathrm{d}t + \int_T^\infty |x^*(t)H(t) * G_{\sigma,b}^{(j)}(t)|^2 \mathrm{d}t \tag{41}$$

Then, we claim that

$$\begin{aligned}
\int_{-\infty}^\infty |x(t) \cdot H(t) * G_{\sigma,b}^{(j)}(t)|^2 \mathrm{d}t &= \int_{-\infty}^\infty |\widehat{x \cdot H}(f) \cdot \widehat{G}_{\sigma,b}^{(j)}(f)|^2 \mathrm{d}f \\
&\geq \int_{f^*-\Delta}^{f^*+\Delta} |\widehat{x \cdot H}(f) \cdot \widehat{G}_{\sigma,b}^{(j)}(f)|^2 \mathrm{d}f \\
&\gtrsim \int_{f^*-\Delta}^{f^*+\Delta} |\widehat{x \cdot H}(f)|^2 \mathrm{d}f \\
&\geq T\mathcal{N}_2^2/k, \tag{42}
\end{aligned}$$

where the first step follows from Parseval's theorem, the second step follows from $[f^* - \Delta, f^* + \Delta] \subset [-\infty, \infty]$, the third step holds with probability at least $0.99$ and follows from $\Delta_0 > 1000\Delta$ and Property I of Lemma H.10, the last step follows from the definition of $f^*$.

By Definition H.32, we have that

$$\int_{-\infty}^{+\infty} |g(t) \cdot H(t) * G_{\sigma,b}^{(j)}(t)|^2 \mathrm{d}t = \int_0^T |g(t) \cdot H(t) * G_{\sigma,b}^{(j)}(t)|^2 \mathrm{d}t \tag{43}$$

$$\leq c\varepsilon \int_0^T |z_j^*(t)|^2 \mathrm{d}t$$

$$\leq c\varepsilon \int_{-\infty}^{+\infty} |z_j^*(t)|^2 \mathrm{d}t$$

$$\leq \int_{-\infty}^{+\infty} c\varepsilon |x(t) \cdot H(t) * G_{\sigma,b}^{(j)}(t)|^2 \mathrm{d}t$$

where the first step from $g(t) = 0, \forall t \notin [0, T]$, the second step follows from Definition H.32, the third step follows from simple algebra, the last step is due to Definition of $z_j^*(t)$.

Then, we claim that

$$\sqrt{\int_{-\infty}^{\infty} |x^* \cdot H * G_{\sigma,b}^{(j)}|^2 \mathrm{d}t} \leq \sqrt{\int_{-\infty}^{\infty} |(x^* + g) \cdot H * G_{\sigma,b}^{(j)}|^2 \mathrm{d}t} + \sqrt{\int_{-\infty}^{\infty} |g \cdot H * G_{\sigma,b}^{(j)}|^2 \mathrm{d}t}$$

$$\lesssim \sqrt{\int_{-\infty}^{\infty} |(x^* + g) \cdot H * G_{\sigma,b}^{(j)}|^2 \mathrm{d}t} \tag{44}$$

where the first step follows from triangle inequality, the second step follows from Eq. (43).

Next, we consider

$$\sqrt{\int_T^{\infty} |(x^* + g) \cdot H * G_{\sigma,b}^{(j)}|^2 \mathrm{d}t} \leq \sqrt{\int_T^{\infty} |x^* \cdot H * G_{\sigma,b}^{(j)}|^2 \mathrm{d}t} + \sqrt{\int_T^{\infty} |g \cdot H * G_{\sigma,b}^{(j)}|^2 \mathrm{d}t}$$

$$\leq \sqrt{\varepsilon_0 \int_{-\infty}^{\infty} |x^* \cdot H * G_{\sigma,b}^{(j)}|^2 \mathrm{d}t} + \sqrt{\int_T^{\infty} |g \cdot H * G_{\sigma,b}^{(j)}|^2 \mathrm{d}t}$$

$$\leq \sqrt{\varepsilon_0 \int_{-\infty}^{\infty} |x^* \cdot H * G_{\sigma,b}^{(j)}|^2 \mathrm{d}t} + \sqrt{\varepsilon_0 \int_{-\infty}^{\infty} |x \cdot H * G_{\sigma,b}^{(j)}|^2 \mathrm{d}t}$$

$$\lesssim \sqrt{\varepsilon_0 \int_{-\infty}^{\infty} |x \cdot H * G_{\sigma,b}^{(j)}|^2 \mathrm{d}t}, \tag{45}$$

where the first step follows from triangle inequality, the second step follows from Eq. (41), the third step follows from Eq. (43), the forth step follows from Eq. (44).

Similarly,

$$\sqrt{\int_{-\infty}^0 |(x^* + g) \cdot H * G_{\sigma,b}^{(j)}|^2 \mathrm{d}t} \lesssim \sqrt{\varepsilon_0 \int_{-\infty}^{\infty} |x \cdot H * G_{\sigma,b}^{(j)}|^2 \mathrm{d}t} \tag{46}$$

Combine equations above, we have that,

$$\sqrt{\int_{-\infty}^0 |(x^* + g) \cdot H * G_{\sigma,b}^{(j)}|^2 \mathrm{d}t + \int_T^{\infty} |(x^* + g) \cdot H * G_{\sigma,b}^{(j)}|^2 \mathrm{d}t}$$

$$\leq \sqrt{\int_{-\infty}^0 |(x^* + g) \cdot H * G_{\sigma,b}^{(j)}|^2 \mathrm{d}t} + \sqrt{\int_T^{\infty} |(x^* + g) \cdot H * G_{\sigma,b}^{(j)}|^2 \mathrm{d}t}$$

$$\lesssim \sqrt{\varepsilon_0 \int_{-\infty}^{\infty} |x \cdot H * G_{\sigma,b}^{(j)}|^2 \mathrm{d}t}$$

where the first step follows from $\sqrt{a + b} \leq \sqrt{a} + \sqrt{b}$, the second step follows from Eq. (45) and Eq. (46).

Hence, we have that $z = (x^* + g) \cdot H * G_{\sigma,b}^{(j)}$ satisfies Property II (in Definition H.27) with probability 0.95.

$\square$

## H.8 Ultra-high sensitivity frequency estimation

In this section, we improve the high sensitivity frequency estimation in Section H.3 with even higher sensitivity, using the results in previous sections. More specifically, we show how to estimate the frequencies of the signal $x_S$ whose frequencies are only $\varepsilon^2 \mathcal{N}$-heavy, while in section H.3 the recoverable signal's frequencies are $\mathcal{N}$-heavy.

**Lemma H.35** (Frequency estimation for one-cluster signal, Lemma 7.3 in Chen et al. (2016)). *For a sufficiently small constant $\varepsilon_0 > 0$, any $f_0 \in [-F, F]$, and $\Delta_0 > 0$, given an $(\varepsilon_0, \Delta_0)$- one-cluster signal $z(t)$ around $f_0$, Procedure* FREQUENCYRECOVERY1CLUSTER, *returns* $\widetilde{f}_0$ *with* $|\widetilde{f}_0 - f_0| \lesssim \Delta_0 \cdot \sqrt{\Delta_0 T}$ *with probability at least* $1 - 2^{-\Omega(k)}$.

The following theorem shows the algorithm for ultra-high sensitivity frequency estimation.

**Theorem H.36** (Ultra-high sensitivity frequency estimation algorithm with low success probability). *Let $x^*(t) = \sum_{j=1}^{k} v_j e^{2\pi \mathbf{i} f_j t}$ and $x(t) = x^*(t) + g(t)$ be our observable signal where $\|g(t)\|_T^2 \leq c\|x^*(t)\|_T^2$ for a sufficiently small constant $c$. Then Procedure* FREQUENCYRECOVERYKCLUSTER *returns a set $L$ of $O(k/(\varepsilon_0 \varepsilon_1 \varepsilon_2))$ frequencies that cover all $\mathcal{N}_2$-heavy clusters and have high SNR (See Definition H.32) of $x^*$, which uses* $\mathrm{poly}(k, \varepsilon^{-1}, \varepsilon_0^{-1}, \varepsilon_1^{-1}, \varepsilon_2^{-1}, \log(1/\delta)) \log(FT)$ *samples and* $\mathrm{poly}(k, \varepsilon^{-1}, \varepsilon_0^{-1}, \varepsilon_1^{-1}, \varepsilon_2^{-1}, \log(1/\delta)) \log^2(FT)$ *time.*

*In particular, for $\Delta_0 = \varepsilon^{-1} \mathrm{poly}(k, \log(1/\delta))/T$ and $\mathcal{N}_2^2 := \varepsilon_1 \varepsilon_2 (\|g(t)\|_T^2 + \delta\|x^*(t)\|_T^2)$, with probability $0.9$, for any $f^*$ with*

$$\int_{f^*-\Delta}^{f^*+\Delta} |\widehat{x \cdot H}(f)|^2 \mathrm{d}f \geq T\mathcal{N}_2^2/k, \tag{47}$$

*there exists an $\widetilde{f} \in L$ satisfying*

$$|f^* - \widetilde{f}| \lesssim \Delta_0 \sqrt{\Delta_0 T}.$$

*Proof.* By Lemma H.34 and Lemma H.35, we prove the theorem. $\qquad\square$

**Theorem H.37** (Ultra-high sensitivity frequency estimation algorithm with high success probability). *Let $x^*(t) = \sum_{j=1}^{k} v_j e^{2\pi \mathbf{i} f_j t}$ and $x(t) = x^*(t) + g(t)$ be our observable signal where $\|g(t)\|_T^2 \leq c\|x^*(t)\|_T^2$ for a sufficiently small constant $c$. Then Procedure* FREQUENCYRECOVERYKCLUSTER *returns a set $L$ of $O(k/(\varepsilon_0 \varepsilon_1 \varepsilon_2))$ frequencies that covers all $\mathcal{N}_2$- heavy clusters of $x^*$, which uses* $\mathrm{poly}(k, \varepsilon^{-1}, \varepsilon_0^{-1}, \varepsilon_1^{-1}, \varepsilon_2^{-1}, \log(1/\delta)) \log(FT)$ *samples and* $\mathrm{poly}(k, \varepsilon^{-1}, \varepsilon_0^{-1}, \varepsilon_1^{-1}, \varepsilon_2^{-1}, \log(1/\delta)) \log^2(FT)$ *time.*

*In particular, for $\Delta_0 = \varepsilon^{-1} \mathrm{poly}(k, \log(1/\delta))/T$ and $\mathcal{N}_2^2 := \varepsilon_1 \varepsilon_2 (\|g(t)\|_T^2 + \delta\|x^*(t)\|_T^2)$, with probability $1 - 2^{-\Omega(k)}$, for any $f^*$ with*

$$\int_{f^*-\Delta}^{f^*+\Delta} |\widehat{x \cdot H}(f)|^2 \mathrm{d}f \geq T\mathcal{N}_2^2/k, \tag{48}$$

*there exists an $\widetilde{f} \in L$ satisfying*

$$|f^* - \widetilde{f}| \lesssim \Delta_0 \sqrt{\Delta_0 T}.$$

The following lemma shows the approximation error guarantee for the recoverable signal $x_S$ of the ultra-high sensitivity frequency estimation algorithm (Theorem H.37).

**Lemma H.38** (Recoverable signal's approximation error guarantee). *Let $x^*(t) = \sum_{j=1}^{k} v_j e^{2\pi \mathbf{i} f_j t}$ and $x(t) = x^*(t) + g(t)$ be our observable signal. Let $\mathcal{N}_1^2 := \varepsilon_1 (\|g(t)\|_T^2 + \delta\|x^*(t)\|_T^2)$. Let $C_1, \cdots, C_l$ are the $\mathcal{N}_1$-heavy clusters from Definition H.11. Let $S^*$ denotes the set of frequencies $f^* \in \{f_j\}_{j\in[k]}$ such that, $f^* \in C_i$ for some $i \in [l]$, and*

$$\int_{C_i} |\widehat{x^* \cdot H}(f)|^2 \mathrm{d}f \geq T\mathcal{N}_1^2/k,$$

*Let $S$ denotes the set of frequencies $f^* \in S^*$ such that, $f^* \in C_j$ for some $j \in [l]$, and*

$$\int_{C_j} |\widehat{x \cdot H}(f)|^2 \mathrm{d}f \geq \varepsilon_2 T \mathcal{N}_1^2 / k,$$

*Then, we have that,*

$$\|x - x_S\|_T + \|x_S - x^*\|_T \leq (1 + \sqrt{2} + O(\sqrt{\varepsilon})) \|g\|_T + O(\sqrt{\delta}) \|x^*\|_T.$$

*Proof.* Following from the fact that $\sqrt{1 + \varepsilon} = 1 + O(\varepsilon)$ for $\varepsilon < 1$, we have

$$\mathcal{N}_1 = \sqrt{\varepsilon_1(\|g\|_T^2 + \delta\|x^*\|_T^2)} \leq \sqrt{\varepsilon_1}\|g\|_T + \sqrt{\delta\varepsilon_1}\|x^*\|_T$$

We have that

$$
\begin{aligned}
\|x^* - x_S\|_T &\leq \|x_{S^*} - x_S\|_T + \|x^* - x_{S^*}\|_T \\
&\leq (1 + O(\sqrt{\varepsilon_2}))\|x - x_{S^*}\|_T + \|x^* - x_{S^*}\|_T \\
&\leq (1 + O(\sqrt{\varepsilon_2}))\|x - x^*\|_T + (2 + O(\sqrt{\varepsilon_2}))\|x^* - x_{S^*}\|_T \\
&\leq (1 + O(\sqrt{\varepsilon_2}))\|g\|_T + (2 + O(\sqrt{\varepsilon_2} + \varepsilon))\mathcal{N}_1 \quad (49)
\end{aligned}
$$

where the first step follows from triangle inequality, the second step follows from Corollary H.22, the third step follows from triangle inequality, the forth step follows from Claim H.12.

Thus, we have that

$$
\begin{aligned}
\|x - x_{S^*}\|_T &\leq \|x - x^*\|_T + \|x^* - x_{S^*}\|_T \\
&\leq \|g\|_T + \|x^* - x_{S^*}\|_T \\
&\leq \|g\|_T + (1 + \varepsilon)\mathcal{N}_1 \quad (50)
\end{aligned}
$$

where the first step follows from triangle inequality, the second step follows from the definition of $g$, the third step follows from Claim H.12.

Therefore,

$$
\begin{aligned}
&\|x - x_S\|_T + \|x_S - x^*\|_T \\
&\leq (\|H(x - x_S)\|_T + \|g\|_T + O(\varepsilon)\|x^* - x_S\|_T) + \|x_S - x^*\|_T \\
&\leq (\|H(x - x_S)\|_T + \|g\|_T + O(\varepsilon)\|x^* - x_S\|_T) + \|x_S - x_{S^*}\|_T + \|x_{S^*} - x^*\|_T \\
&\leq (\|H(x - x_S)\|_T + \|g\|_T + O(\varepsilon)\|x^* - x_S\|_T) + (1 + 2\varepsilon)\|H(x_S - x_{S^*})\|_T + \|x_{S^*} - x^*\|_T \\
&= \|g\|_T + O(\varepsilon)\|x^* - x_S\|_T + (1 + O(\varepsilon))(\|H(x - x_S)\|_T + \|H(x_S - x_{S^*})\|_T) + \|x_{S^*} - x^*\|_T \\
&\leq \|g\|_T + O(\varepsilon)\|x^* - x_S\|_T + (1 + O(\varepsilon))(\|H(x - x_S)\|_T + \|H(x_S - x_{S^*})\|_T) + (1 + \varepsilon)\mathcal{N}_1 \\
&\leq \|g\|_T + O(\varepsilon)\|x^* - x_S\|_T + (1 + O(\varepsilon))\sqrt{2}\sqrt{\|H(x - x_S)\|_T^2 + \|H(x_S - x_{S^*})\|_T^2} + (1 + \varepsilon)\mathcal{N}_1 \\
&\leq \|g\|_T + O(\varepsilon)\|x^* - x_S\|_T + (1 + O(\varepsilon))(1 + O(\sqrt{\varepsilon_2}))\sqrt{2}\|x - x_{S^*}\|_T + (1 + \varepsilon)\mathcal{N}_1 \\
&\leq \|g\|_T + O(\varepsilon)((1 + O(\sqrt{\varepsilon_2}))\|g\|_T + (2 + O(\sqrt{\varepsilon_2} + \varepsilon))\mathcal{N}_1) \\
&\quad + (1 + O(\varepsilon))(1 + O(\sqrt{\varepsilon_2}))\sqrt{2}\|x - x_{S^*}\|_T + (1 + \varepsilon)\mathcal{N}_1 \\
&\leq \|g\|_T + O(\varepsilon)((1 + O(\sqrt{\varepsilon_2}))\|g\|_T + (2 + O(\sqrt{\varepsilon_2} + \varepsilon))\mathcal{N}_1) \\
&\quad + (\sqrt{2} + O(\varepsilon + \sqrt{\varepsilon_2}))(\|g\|_T + (1 + \varepsilon)\mathcal{N}_1) + (1 + \varepsilon)\mathcal{N}_1 \\
&\leq (1 + \sqrt{2} + O(\sqrt{\varepsilon}))\|g\|_T + O(\sqrt{\delta})\|x^*\|_T,
\end{aligned}
$$

where the first step follows from Lemma H.23, the second step follows from triangle inequality, the third step follows from $x_S - x_{S^*}$ being $k$-Fourier-sparse and Property VI of Lemma H.7, the forth step change the order of the terms, the fifth step follows from Claim H.12, the sixth step follows from $\|H(x - x_S)\|_T + \|H(x_S - x_{S^*})\|_T \leq \sqrt{2}\sqrt{\|H(x - x_S)\|_T^2 + \|H(x_S - x_{S^*})\|_T^2}$, the seventh step follows from Lemma H.21, the eighth step follows from Eq. (49), the ninth step follows from Eq. (50), the last step follows from $\varepsilon = \varepsilon_0 = \varepsilon_1 = \varepsilon_2$. $\square$

The following lemma shows that the recoverable signal $x_S(t)$'s energy is close to the observation signal $x(t)$.

**Lemma H.39** (Recoverable signal's energy). *Let $x^*(t) = \sum_{j=1}^{k} v_j e^{2\pi \mathbf{i} f_j t}$ and $x(t) = x^*(t) + g(t)$ be our observable signal. Let $\mathcal{N}_1^2 := \varepsilon_1(\|g(t)\|_T^2 + \delta \|x^*(t)\|_T^2)$. Let $C_1, \cdots, C_l$ are the $\mathcal{N}_1$-heavy clusters from Definition H.11. Let $S^*$ denotes the set of frequencies $f^* \in \{f_j\}_{j \in [k]}$ such that, $f^* \in C_i$ for some $i \in [l]$, and*

$$\int_{C_i} |\widehat{x^* \cdot H}(f)|^2 \mathrm{d}f \geq T\mathcal{N}_1^2/k,$$

*Let $S$ denotes the set of frequencies $f^* \in S^*$ such that, $f^* \in C_j$ for some $j \in [l]$, and*

$$\int_{C_j} |\widehat{x \cdot H}(f)|^2 \mathrm{d}f \geq \varepsilon_2 T\mathcal{N}_1^2/k,$$

*Then, we have that,*

$$\|x_S\|_T \lesssim \|g\|_T + \|x^*\|_T$$

*Proof.* We have that,

$$
\begin{aligned}
\|x_S\|_T &\leq \|x_{S^*} - x^*\|_T + \|x_S - x_{S^*}\|_T + \|x^*\|_T \\
&\lesssim \|x_{S^*} - x^*\|_T + \|x - x_{S^*}\|_T + \|x^*\|_T \\
&\lesssim \|x_{S^*} - x^*\|_T + \|x - x^*\|_T + \|x^*\|_T \\
&\leq \|g\|_T + \|x^*\|_T,
\end{aligned}
$$

where the first step follows from triangle inequality, the second step follows from Corollary H.22, the third step follows from triangle inequality, the forth step follows from Claim H.12. □

## H.9  High SNR and recoverable signals

**Lemma H.40** (High SNR and recoverable approximation error guarantee). *Let $x^*(t) = \sum_{j=1}^{k} v_j e^{2\pi \mathbf{i} f_j t}$ and $x(t) = x^*(t) + g(t)$ be our observable signal. Let $\mathcal{N}_1^2 := \varepsilon_1(\|g(t)\|_T^2 + \delta \|x^*(t)\|_T^2)$. Let $C_1, \cdots, C_l$ are the $\mathcal{N}_1$-heavy clusters from Definition H.11. Let $S^*$ denotes the set of frequencies $f^* \in \{f_j\}_{j \in [k]}$ such that, $f^* \in C_i$ for some $i \in [l]$, and*

$$\int_{C_i} |\widehat{x^* \cdot H}(f)|^2 \mathrm{d}f \geq T\mathcal{N}_1^2/k,$$

*Let $S$ denotes the set of frequencies $f^* \in S^*$ such that, $f^* \in C_j$ for some $j \in [l]$, and*

$$\int_{C_j} |\widehat{x \cdot H}(f)|^2 \mathrm{d}f \geq \varepsilon_2 T\mathcal{N}_1^2/k,$$

*And $S_f$ is defined in Definition H.32. Then, we have that,*

$$\|x_{S_f} - x_S\|_T \leq (1 + O(\varepsilon)) \cdot \|g(t)\|_T \tag{51}$$

*Proof.* We have that

$$S_f \subseteq S.$$

And then for any $f \in S \setminus S_f, j = h_{\sigma,b}(f)$, we have that

$$\|(g \cdot H(t)) * G_{\sigma,b}^{(j)}(t)\|_T^2 \geq (1 - c \cdot \varepsilon)\|(x^* \cdot H(t)) * G_{\sigma,b}^{(j)}(t)\|_T^2$$

where the first step follows from Definition H.32, the second step is from simple algebra.

Let $\mathcal{T} = S \setminus S_f$. And for any $j \in [B]$, if $j \in [B] \setminus S_g$, $\mathcal{T}_j = \{i \in S | h_{\sigma,b}(f_i) = j\}$. Otherwise, $\mathcal{T}_j = \emptyset$. Moreover, we have that for any $f \in \mathrm{supp}(\widehat{x}_{\mathcal{T}_j} * \widehat{H})$,

$$\widehat{G}_{\sigma,b}^{(j)}(f) \geq 1 - \frac{\delta}{k} \tag{52}$$

From Property VI of Lemma H.7, we have that

$$\int_{-\infty}^{+\infty} |x^*(t) \cdot H(t)|^2 \mathrm{d}t \in [1 - \varepsilon, 1] \cdot \int_{-\infty}^{+\infty} |x^*(t)|^2 \mathrm{d}t. \tag{53}$$

By Lemma H.29, we know that $(x^* \cdot H) * G_{\sigma,b}^{(j)}$ always satisfies Property II (in Definition H.27):

$$
\begin{aligned}
&T \|x^*(t) H(t) * G_{\sigma,b}^{(j)}(t)\|_T^2 \\
&= \int_0^T |x^*(t) H(t) * G_{\sigma,b}^{(j)}(t)|^2 \mathrm{d}t \\
&\geq (1 - \varepsilon_0) \int_{-\infty}^{+\infty} |x^*(t) H(t) * G_{\sigma,b}^{(j)}(t)|^2 \mathrm{d}t \\
&= (1 - \varepsilon_0) \int_{-\infty}^{+\infty} |(\widehat{x}^*(f) * \widehat{H}(f)) * \widehat{G}_{\sigma,b}^{(j)}(f)|^2 \mathrm{d}f \\
&= (1 - \varepsilon_0) \left( \int_{-\infty}^{+\infty} |(\widehat{x}^*(f) * \widehat{H}(f)) \cdot \widehat{G}_{\sigma,b}^{(j)}(f)|^2 \mathrm{d}f + \int_{-\infty}^{+\infty} |(\widehat{x}^*(f) * \widehat{H}(f)) \cdot \widehat{G}_{\sigma,b}^{(j)}(f)|^2 \mathrm{d}f \right) \\
&\geq (1 - \varepsilon_0) \cdot \int_{-\infty}^{+\infty} |(\widehat{x}^*(f) * \widehat{H}(f)) \cdot \widehat{G}_{\sigma,b}^{(j)}(f)|^2 \mathrm{d}f \\
&\geq (1 - \varepsilon_0) \cdot \int_{-\infty}^{+\infty} |(\widehat{x}^*(f) * \widehat{H}(f))|^2 \mathrm{d}f \tag{54}
\end{aligned}
$$

where the first step follows from the definition of the norm, the second step is from Lemma H.29, the third step is due to Parseval's Theorem, the forth step is based on the Large Offset event not happening, the fifth step is based on simple algebra, the last step is because of Lemma H.29.

We also have that

$$
\begin{aligned}
&T \|x_{S_f}(t) - x_S(t)\|_T^2 \\
&= T \|x_{\mathcal{T}}\|_T^2 \\
&\leq (T/(1 - \varepsilon)^2) \cdot \|x_{\mathcal{T}}(t) \cdot H(t)\|_T^2 \\
&= \frac{1}{1 - \varepsilon^2} \cdot \int_0^T |x_{\mathcal{T}}(t) \cdot H(t)|^2 \mathrm{d}t \\
&\leq \frac{1}{1 - \varepsilon^2} \cdot \int_{-\infty}^{\infty} |x_{\mathcal{T}}(t) \cdot H(t)|^2 \mathrm{d}t \\
&= \frac{1}{1 - \varepsilon^2} \cdot \int_{-\infty}^{\infty} |\widehat{x}_{\mathcal{T}}(f) * \widehat{H}(f)|^2 \mathrm{d}f \\
&= \frac{1}{1 - \varepsilon^2} \cdot \sum_{j=1}^{B} \int_{-\infty}^{\infty} |\widehat{x}_{\mathcal{T}_j}(f) * \widehat{H}(f)|^2 \mathrm{d}f \\
&\leq \frac{k^2}{(1 - \varepsilon)^2 (k - \delta)^2} \cdot \sum_{j \in B \setminus S_g} T \|(x^*(t) \cdot H(t)) * G_{\sigma,b}^{(j)}(t)\|_T^2 \\
&\leq \frac{k^2}{(1 - c\varepsilon)(1 - \varepsilon)^2 (k - \delta)^2} \cdot \sum_{j \in B \setminus S_g} T \|(g(t) \cdot H(t)) * G_{\sigma,b}^{(j)}(t)\|_T^2 \tag{55}
\end{aligned}
$$

where the first step follows from Definition of $\mathcal{T}$, the second step follows from Eq. (53), the third step is based on definition of norm, the forth step follows from simple algebra, the fifth step follows from Parseval's Theorem, the six step is due to Large Offset event not happening, the seventh step is due to Lemma H.29, the eighth step follows from Eq. (51).

In the following, we have that

$$\sum_{j \in [B]} T \cdot \|(g(t) \cdot H(t)) * G_{\sigma,b}^{(j)}(t)\|_T^2$$

$$\leq \sum_{j\in[B]} \int_0^T |(g^*(t) \cdot H(t)) * G_{\sigma,b}^{(j)}(t)|^2 \mathrm{d}t$$

$$\leq \sum_{j\in[B]} \int_{-\infty}^{\infty} |(g^*(t) \cdot H(t)) * G_{\sigma,b}^{(j)}(t)|^2 \mathrm{d}t$$

$$= \sum_{j\in[B]} \int_{-\infty}^{\infty} |(\widehat{g}(f) * \widehat{H}(f)) \cdot \widehat{G}_{\sigma,b}^{(j)}(f)|^2 \mathrm{d}f$$

$$\leq \frac{k^2}{(k-\delta)^2} \int_{-\infty}^{\infty} |\widehat{g}(f) * \widehat{H}(f)|^2 \mathrm{d}f$$

$$= \frac{k^2}{(k-\delta)^2} \cdot \int_{-\infty}^{\infty} |g(t) \cdot H(t)|^2 \mathrm{d}t$$

$$= \frac{k^2}{(k-\delta)^2} \cdot \int_0^T |g(t) \cdot H(t)|^2 \mathrm{d}t$$

$$\leq \frac{k^2}{(k-\delta)^2} \cdot \int_0^T |g(t)|^2 \mathrm{d}t$$

$$= T \frac{k^2}{(k-\delta)^2} \|g(t)\|_T^2 \tag{56}$$

where the first step is due to the definition of norm, the second step follows from $g(t) = 0$ when $t \notin [0,T]$, the third step follows from Parseval's Theorem, the forth step is because of Lemma H.29, the fifth step is from Parseval's Theorem, the sixth step is based on $g(t) = 0$ when $t \notin [0,T]$, the seventh step is from $|H(t)|^2 \leq 1$, the last step is from the definition of norm. We have that

$$T\|x_{S_f}(t) - x_S(t)\|_T^2$$

$$\leq \frac{k^2}{(1-c\varepsilon)(1-\varepsilon)^2(k-\delta)^2} \cdot \sum_{j\in B\setminus S_g} T\|(g^*(t) \cdot H(t)) * G_{\sigma,b}^{(j)}(t))\|_T^2$$

$$\leq \frac{k^2}{(1-c\varepsilon)(1-\varepsilon)^2(k-\delta)^2} \sum_{j\in[B]} T\|(g^*(t) \cdot H(t)) * G_{\sigma,b}^{(j)}(t))\|_T^2$$

$$\leq \frac{k^4}{(1-c\varepsilon)(1-\varepsilon)^2(k-\delta)^4} T\|g(t)\|_T^2$$

$$\leq (1+O(\varepsilon))T\|g(t)\|_T^2$$

where the first step follows from Eq. (55), the second step follows from simple algebra, the third step is due to Eq.(54), the forth step is because of the reason that $\delta$ is much smaller than $\varepsilon$ and $\varepsilon < 1$. $\quad\square$

**Lemma H.41** (High SNR signal's energy). *Let $x^*(t) = \sum_{j=1}^k v_j e^{2\pi \mathbf{i} f_j t}$ and $x(t) = x^*(t) + g(t)$ be our observable signal. Let $\mathcal{N}_1^2 := \varepsilon_1(\|g(t)\|_T^2 + \delta\|x^*(t)\|_T^2)$. Let $C_1, \cdots, C_l$ are the $\mathcal{N}_1$-heavy clusters from Definition H.11. Let $S^*$ denotes the set of frequencies $f^* \in \{f_j\}_{j\in[k]}$ such that, $f^* \in C_i$ for some $i \in [l]$, and*

$$\int_{C_i} |\widehat{x^* \cdot H}(f)|^2 \mathrm{d}f \geq T\mathcal{N}_1^2/k,$$

*Let $S$ denotes the set of frequencies $f^* \in S^*$ such that, $f^* \in C_j$ for some $j \in [l]$, and*

$$\int_{C_j} |\widehat{x \cdot H}(f)|^2 \mathrm{d}f \geq \varepsilon_2 T\mathcal{N}_1^2/k,$$

*Let $S_f$ be defined in Definition H.32. Then, we have that,*

$$\|x_{S_f}\|_T \leq (1+O(\varepsilon))\|g\|_T + \|x^*\|_T$$

*Proof.* We have that,

$$
\begin{aligned}
\|x_{S_f}\|_T &\leq \|x_{S_f} - x_S\|_T + \|x_{S^*} - x^*\|_T + \|x_S - x_{S^*}\|_T + \|x^*\|_T \\
&\lesssim \|x_{S_f} - x_S\|_T + \|x_{S^*} - x^*\|_T + \|x - x_{S^*}\|_T + \|x^*\|_T \\
&\lesssim \|x_{S_f} - x_S\|_T + \|x_{S^*} - x^*\|_T + \|x - x^*\|_T + \|x^*\|_T \\
&\leq \|x_{S_f} - x_S\|_T + \|g\|_T + \|x^*\|_T \\
&\leq (1 + O(\varepsilon))\|g\|_T + \|x^*\|_T,
\end{aligned}
$$

where the first step follows from triangle inequality, the second step follows from Corollary H.22, the third step follows from triangle inequality, the forth step follows from Claim H.12, where the last step follows from Lemma H.40. □

## H.10 $(3 + \sqrt{2} + \varepsilon)$-approximate algorithm

In this section, we prove the main result: a $(3 + \sqrt{2} + \varepsilon)$-approximate Fourier interpolation algorithm, which significantly improves the accuracy of Chen et al. (2016)'s result.

**Theorem H.42** (Fourier interpolation with $(3+\sqrt{2}+\varepsilon)$-approximation error)**.** *Let $x(t) = x^*(t)+g(t)$, where $x^*$ is $k$-Fourier-sparse signal with frequencies in $[-F, F]$. Given samples of $x$ over $[0,T]$ we can output $y(t)$ such that with probability at least $1 - 2^{-\Omega(k)}$,*

$$
\|y - x^*\|_T \leq (3 + \sqrt{2} + \varepsilon)\|g\|_T + \delta\|x^*\|_T.
$$

*Our algorithm uses $\mathrm{poly}(k, \varepsilon^{-1}, \log(1/\delta)) \log(FT)$ samples and $\mathrm{poly}(k, \varepsilon^{-1}, \log(1/\delta)) \cdot \log^2(FT)$ time. The output $y$ is $\mathrm{poly}(k, \varepsilon^{-1}, \log(1/\delta))$-Fourier-sparse signal.*

*Proof.* Let $\mathcal{N}_2^2 := \varepsilon_1\varepsilon_2(\|g(t)\|_T^2 + \delta\|x^*(t)\|_T^2)$, $\mathcal{N}_1^2 := \varepsilon_1(\|g(t)\|_T^2 + \delta\|x^*(t)\|_T^2)$ be the heavy cluster parameter.

First, by Lemma H.12, there is a set of frequencies $S^* \subset [k]$ and $x_{S^*}(t) = \sum_{j \in S^*} v_j e^{2\pi \mathbf{i} f_j t}$ such that

$$
\|x_{S^*} - x^*\|_T^2 \leq (1 + \varepsilon)\mathcal{N}_1^2. \tag{57}
$$

Furthermore, each $f_j$ with $j \in S^*$ belongs to an $\mathcal{N}_1$-heavy cluster $C_j$ with respect to the filter function $H$ defined in Definition H.6.

By Definition H.11 of heavy cluster, it holds that

$$
\int_{C_j} |\widehat{H \cdot x^*}(f)|^2 \mathrm{d}f \geq T\mathcal{N}_1^2/k.
$$

By Definition H.11, we also have $|C_j| \leq k \cdot \Delta_h$, where $\Delta_h$ is the bandwidth of $\widehat{H}$.

Let $\Delta \in \mathbb{R}_+$, and $\Delta > k \cdot \Delta_h$, which implies that $C_j \subseteq [f_j - \Delta, f_j + \Delta]$. Thus, we have

$$
\int_{f_j-\Delta}^{f_j+\Delta} |\widehat{H \cdot x^*}(f)|^2 \mathrm{d}f \geq T\mathcal{N}_1^2/k.
$$

By Corollary H.22, there is a set of frequencies $S \subset S^*$ and $x_S(t) = \sum_{j \in S} v_j e^{2\pi \mathbf{i} f_j t}$ such that

$$
\|x_S - x_{S^*}\|_T^2 \leq (1 + O(\sqrt{\varepsilon_2}))\|x - x_{S^*}\|_T^2.
$$

Let $g' = x - x_{S^*}$.

In the following part, we will only focus on recovering the high SNR frequency. Let $S_f$ be defined in Definition H.32. It's to know $S_f \subset S$ By applying Theorem H.37, there is an algorithm that outputs a set of frequencies $L \subset \mathbb{R}$ such that, $|L| = O(k/(\varepsilon_0\varepsilon_1\varepsilon_2))$, and with probability at least $1 - 2^{-\Omega(k)}$, for any $f_j$ with $j \in S_f$, there is a $\widetilde{f} \in L$ such that,

$$
|f_j - \widetilde{f}| \lesssim \Delta\sqrt{\Delta T}.
$$

We define a map $p : \mathbb{R} \to L$ as follows:

$$p(f) := \arg\min_{\widetilde{f} \in L} |f - \widetilde{f}| \quad \forall f \in \mathbb{R}.$$

Then, $x_S(t)$ can be expressed as

$$
\begin{aligned}
x_{S_f}(t) &= \sum_{j \in S_f} v_j e^{2\pi \mathbf{i} f_j t} \\
&= \sum_{j \in S_f} v_j e^{2\pi \mathbf{i} \cdot p(f_j) t} \cdot e^{2\pi \mathbf{i} \cdot (f_j - p(f_j)) t} \\
&= \sum_{\widetilde{f} \in L} e^{2\pi \mathbf{i} \widetilde{f} t} \cdot \sum_{j \in S_f : \, p(f_j) = \widetilde{f}} v_j e^{2\pi \mathbf{i} (f_j - \widetilde{f}) t},
\end{aligned}
$$

where the first step follows from the definition of $x_S(t)$, the last step follows from interchanging the summations.

For each $\widetilde{f}_i \in L$, by Corollary H.2 with $x^* = x_S, \Delta = \Delta\sqrt{\Delta T}$, we have that there exist degree $d = O(T\Delta\sqrt{\Delta T} + k^3 \log k + k \log 1/\delta)$ polynomials $P_i(t)$ corresponding to $\widetilde{f}_i \in L$ such that,

$$\|x_{S_f}(t) - \sum_{\widetilde{f}_i \in L} e^{2\pi \mathbf{i} \widetilde{f}_i t} P_i(t)\|_T \leq \sqrt{\delta} \|x_{S_f}(t)\|_T \tag{58}$$

Define the following function family:

$$\mathcal{F} := \mathrm{span}\Big\{ e^{2\pi \mathbf{i} \widetilde{f} t} \cdot t^j \mid \forall \widetilde{f} \in L, j \in \{0, 1, \ldots, d\} \Big\}.$$

Note that $\sum_{\widetilde{f}_i \in L} e^{2\pi \mathbf{i} \widetilde{f}_i t} P_i(t) \in \mathcal{F}$.

By Claim H.16, for function family $\mathcal{F}$, $K_{\mathrm{Uniform}[0,T]} = O((|L|d)^4 \log^3(|L|d))$.

By Lemma H.18, we have that, choosing a set $W$ of $O(\varepsilon^{-1} K_{\mathrm{Uniform}[0,T]} \log(|L|d/\rho))$ i.i.d. samples uniformly at random over duration $[0, T]$ is a $(\varepsilon, \rho)$-WBSP.

By Lemma H.19, there is an algorithm that runs in $O(\varepsilon^{-1}|W|(|L|d)^{\omega-1} \log(1/\rho))$-time using samples in $W$, and outputs $y'(t) \in \mathcal{F}$ such that, with probability $1 - \rho$,

$$\Big\| y'(t) - \sum_{\widetilde{f}_i \in L} e^{2\pi \mathbf{i} \widetilde{f}_i t} P_i(t) \Big\|_T \leq (1 + \varepsilon) \Big\| x(t) - \sum_{\widetilde{f}_i \in L} e^{2\pi \mathbf{i} \widetilde{f}_i t} P_i(t) \Big\|_T \tag{59}$$

Then by Lemma H.3, we have that there is a $(kd)$-Fourier-sparse signal $y(t)$, such that

$$\|y - y'\|_T \leq \delta' \tag{60}$$

where $\delta' > 0$ is any positive real number, thus, $y$ can be arbitrarily close to $y'$.

Moreover, the sparsity of $y(t)$ is

$$kd = kO(T\Delta\sqrt{\Delta T} + k^3 \log k + k \log 1/\delta) = \mathrm{poly}(k, \varepsilon^{-1}, \log(1/\delta)).$$

Therefore, the total approximation error can be upper bounded as follows:

$$\|y - x^*\|_T$$

$$\leq \|y - y'\|_T + \Big\| y' - \sum_{\widetilde{f}_i \in L} e^{2\pi \mathbf{i} \widetilde{f}_i t} P_i(t) \Big\|_T + \Big\| \sum_{\widetilde{f}_i \in L} e^{2\pi \mathbf{i} \widetilde{f}_i t} P_i(t) - x^* \Big\|_T \quad \text{(Triangle inequality)}$$

$$\leq (1 + 0.1\varepsilon) \Big\| y' - \sum_{\widetilde{f}_i \in L} e^{2\pi \mathbf{i} \widetilde{f}_i t} P_i(t) \Big\|_T + \Big\| \sum_{\widetilde{f}_i \in L} e^{2\pi \mathbf{i} \widetilde{f}_i t} P_i(t) - x^* \Big\|_T \quad \text{(Eq. (60))}$$

$$\leq (1 + 2\varepsilon) \Big\| x - \sum_{\widetilde{f}_i \in L} e^{2\pi \mathbf{i} \widetilde{f}_i t} P_i(t) \Big\|_T + \Big\| \sum_{\widetilde{f}_i \in L} e^{2\pi \mathbf{i} \widetilde{f}_i t} P_i(t) - x^* \Big\|_T \quad \text{(Eq. (59))}$$

$$\leq (1+2\varepsilon)(\|x - x_{S_f}\|_T + \|x_{S_f} - x^*\|_T) + 2(1+2\varepsilon)\|\sum_{\widetilde{f}_i \in L} e^{2\pi \mathbf{i}\widetilde{f}_i t} P_i(t) - x_{S_f}\|_T$$
(Triangle Inequality)

$$\leq (1+2\varepsilon)(\|x - x_S\|_T + 2\|x_{S_f} - x_S\|_T + \|x_S - x^*\|_T) + 2(1+2\varepsilon)\|\sum_{\widetilde{f}_i \in L} e^{2\pi \mathbf{i}\widetilde{f}_i t} P_i(t) - x_{S_f}\|_T$$
(Triangle Inequality)

$$\leq (1+2\varepsilon)(\|x - x_S\|_T + \|x_S - x^*\|_T) + O(\sqrt{\delta})\|x_{S_f}(t)\|_T + 2(1+2\varepsilon)\|x_{S_f} - x_S\|_T$$
(Eq. (58))

$$\leq (1+2\varepsilon)(1 + \sqrt{2} + O(\sqrt{\varepsilon}))\|g\|_T + O(\sqrt{\delta})\|x^*\|_T + O(\sqrt{\delta})\|x_{S_f}(t)\|_T + 2(1+2\varepsilon)\|x_{S_f} - x_S\|_T$$
(Lemma H.38)

$$\leq (1+2\varepsilon)(1 + \sqrt{2} + O(\sqrt{\varepsilon}))\|g\|_T + O(\sqrt{\delta})\|x^*\|_T + O(\sqrt{\delta})(\|g\|_T + \|x^*\|_T) + 2(1+2\varepsilon)(1 + O(\varepsilon))\|g(t)\|_T$$
(Lemma H.39)

$$\leq (3 + \sqrt{2} + O(\sqrt{\varepsilon}))\|g\|_T + O(\sqrt{\delta})\|x^*\|_T$$

By re-scaling $\varepsilon$ and $\delta$, we prove the theorem.

□

# I  Improving Band-Limited Interpolation Precision in a Smaller Range

In this section, we show that the approximation error of the Fourier interpolation algorithm developed in Section H can be further improved, if we only care about the signal in a shorter time duration $[0, (1-c)T]$ for $c \in (0,1)$. The main result of this section is Theorem I.4.

## I.1  Control noise

**Lemma I.1.** *Let* $x^*(t) = \sum_{j=1}^{k} v_j e^{2\pi \mathbf{i} f_j t}$ *and* $x(t) = x^*(t) + g(t)$ *be our observable signal. Let* $\mathcal{N}_1^2 := \varepsilon_1(\|g(t)\|_T^2 + \delta\|x^*(t)\|_T^2)$. *Let* $C_1, \cdots, C_l$ *are the* $\mathcal{N}_1$-*heavy clusters from Definition H.11. Let* $S^*$ *denotes the set of frequencies* $f^* \in \{f_j\}_{j \in [k]}$ *such that,* $f^* \in C_i$ *for some* $i \in [l]$, *and*

$$\int_{C_i} |\widehat{x^* \cdot H}(f)|^2 \mathrm{d}f \geq T\mathcal{N}_1^2/k,$$

*Let* $S$ *denotes the set of frequencies* $f^* \in S^*$ *such that,* $f^* \in C_j$ *for some* $j \in [l]$, *and*

$$\int_{C_j} |\widehat{x \cdot H}(f)|^2 \mathrm{d}f \geq \varepsilon_2 T\mathcal{N}_1^2/k,$$

*Then, we have that,*

$$\|x - x_S\|_{T'} + \|x_S - x^*\|_{T'} \leq (\sqrt{2} + O(\sqrt{\varepsilon} + c))\|g\|_T + O(\sqrt{\delta})\|x^*\|_T.$$

*Proof.* Following from the fact that $\sqrt{1+\varepsilon} = 1 + O(\varepsilon)$ for $\varepsilon < 1$, we have

$$\mathcal{N}_1 = \sqrt{\varepsilon_1(\|g\|_T^2 + \delta\|x^*\|_T^2)} \leq \sqrt{\varepsilon_1}\|g\|_T + \sqrt{\delta\varepsilon_1}\|x^*\|_T.$$

We have that

$$\begin{aligned}
\|x - x_{S^*}\|_T &\leq \|x - x^*\|_T + \|x^* - x_{S^*}\|_T \\
&\leq \|g\|_T + \|x^* - x_{S^*}\|_T \\
&\leq \|g\|_T + (1+\varepsilon)\mathcal{N}_1, \quad\quad\quad (61)
\end{aligned}$$

where the first step follows from triangle inequality, the second step follows the definition of $g$, the third step follows from Claim H.12.

Therefore,

$$\|x - x_S\|_{T'} + \|x_S - x^*\|_{T'}$$
$$\leq \|x - x_S\|_{T'} + \|x_S - x_{S^*}\|_{T'} + \|x_{S^*} - x^*\|_{T'}$$
$$\leq \|x - x_S\|_{T'} + \|x_S - x_{S^*}\|_{T'} + (1 + 2c)\|x_{S^*} - x^*\|_{T'}$$
$$\leq (1 + 2\delta)\|H(x - x_S)\|_{T'} + (1 + 2\delta)\|H(x_S - x_{S^*})\|_{T'} + (1 + 2c)\|x_{S^*} - x^*\|_T$$
$$\leq (1 + O(\delta))(1 + 2c)(\|H(x - x_S)\|_T + \|H(x_S - x_{S^*})\|_T) + (1 + \varepsilon)(1 + O(c))\mathcal{N}_1$$
$$\leq (1 + O(\delta))(1 + 2c)\sqrt{2}\sqrt{\|H(x - x_S)\|_T^2 + \|H(x_S - x_{S^*})\|_T^2} + (1 + \varepsilon)(1 + O(c))\mathcal{N}_1$$
$$\leq (1 + O(\delta))(1 + O(\sqrt{\varepsilon_2}))(1 + O(c))\sqrt{2}\|x - x_{S^*}\|_T + (1 + \varepsilon)(1 + O(c))\mathcal{N}_1$$
$$\leq (\sqrt{2} + O(\delta + \sqrt{\varepsilon_2} + c))(\|g\|_T + (1 + \varepsilon)\mathcal{N}_1) + (1 + \varepsilon)(1 + O(c))\mathcal{N}_1$$
$$\leq (\sqrt{2} + O(\sqrt{\varepsilon} + c))\|g\|_T + O(\sqrt{\delta})\|x^*\|_T,$$

where the first step follows from triangle inequality, the second step follows from for any function $x : \mathbb{R} \to \mathbb{C}$, $(1 - c)\|x\|_{T'} \leq \|x\|_T$, the third step follows from Property I of Lemma H.7 and $(1 - c)/2 < (\frac{1}{2} - \frac{2}{s_1})s_3$, the forth step follows from Claim H.12, the fifth step follows from $\|H(x - x_S)\|_T + \|H(x_S - x_{S^*})\|_T \leq \sqrt{2}\sqrt{\|H(x - x_S)\|_T^2 + \|H(x_S - x_{S^*})\|_T^2}$, the sixth step follows from Lemma H.21, the seventh step follows from Eq. (49), the last step follows from $\varepsilon = \varepsilon_0 = \varepsilon_1 = \varepsilon_2$. $\qquad\qquad\square$

**Parameters setting**   By Section C.3 in Chen et al. (2016), we choose parameters for filter function $(H(t), \widehat{H}(f))$ as follows:

- By Eq. (19) in the proof of Property VI of filter function $(H(t), \widehat{H}(f))$, we need $(1 - s_3(1 - \frac{2}{s_1})) \cdot \widetilde{O}(k^4) \leq \varepsilon$, thus we have that $\min(\frac{1}{1 - s_3}, s_1) \geq \widetilde{O}(k^4)/\varepsilon$.

- In the proof of Property V of filter function $(H(t), \widehat{H}(f))$, we set $\ell \gtrsim k \log(k/\delta)$.

- In the proof of Lemma I.1, we set $(1 - c)/2 < (\frac{1}{2} - \frac{2}{s_1})s_3$. Thus, we have that $\min(s_3, 1 - \frac{4}{s_1}) \geq 1 - \frac{c}{2}$ or equivalently $\min(\frac{1}{1 - s_3}, s_1/4) \geq \frac{2}{c}$.

- $\Delta_h$ is determined by the parameters of filter $(H(t), \widehat{H}(f))$ in Eq. (20): $\Delta_h \approx \frac{s_1 \ell}{s_3 T}$. Combining the setting of $s_1$, $s_3$ $\ell$, we should set $\Delta_h \gtrsim \max(\widetilde{O}(k^5 \log(1/\delta))/(\varepsilon T), O(k \log(k/\delta)/(cT)))$.

## I.2   $(\sqrt{2} + \varepsilon)$-approximation ratio

**Corollary I.2** (Corollary of Theorem H.37)**.** *Let* $x^*(t) = \sum_{j=1}^k v_j e^{2\pi \mathbf{i} f_j t}$ *and* $x(t) = x^*(t) + g(t)$ *be our observable signal where* $\|g(t)\|_T^2 \leq c_0\|x^*(t)\|_T^2$ *for a sufficiently small constant* $c_0$. *Then Procedure* FREQUENCYRECOVERYKCLUSTER *returns a set* $L$ *of* $O(k/(\varepsilon_0 \varepsilon_1 \varepsilon_2))$ *frequencies that covers all* $\mathcal{N}_2$-*heavy clusters of* $x^*$, *which uses* $\mathrm{poly}(k, c^{-1}, \varepsilon^{-1}, \varepsilon_0^{-1}, \varepsilon_1^{-1}, \varepsilon_2^{-1}, \log(1/\delta)) \log(FT)$ *samples and* $\mathrm{poly}(k, c^{-1}, \varepsilon^{-1}, \varepsilon_0^{-1}, \varepsilon_1^{-1}, \varepsilon_2^{-1}, \log(1/\delta)) \log^2(FT)$ *time.*

*In particular, for* $\Delta_0 = c^{-1}\varepsilon^{-1}\mathrm{poly}(k, \log(1/\delta))/T$ *and* $\mathcal{N}_2^2 := \varepsilon_1 \varepsilon_2(\|g(t)\|_T^2 + \delta\|x^*(t)\|_T^2)$, *with probability* $1 - 2^{-\Omega(k)}$, *for any* $f^*$ *with*

$$\int_{f^* - \Delta}^{f^* + \Delta} |\widehat{x \cdot H}(f)|^2 \mathrm{d}f \geq T\mathcal{N}_2^2/k, \qquad (62)$$

*there exists an* $\widetilde{f} \in L$ *satisfying*

$$|f^* - \widetilde{f}| \lesssim \Delta_0 \sqrt{\Delta_0 T}.$$

**Remark I.3.** *The proof is similar with the proof of Theorem H.37.*

**Theorem I.4** (($\sqrt{2} + \varepsilon$)-approximate Fourier interpolation algorithm with shrinking range)**.** *Let* $x(t) = x^*(t) + g(t)$, *where* $x^*$ *is* $k$-*Fourier-sparse signal with frequencies in* $[-F, F]$. *Let* $T' =$

$T(1 - c)$. *Given samples of $x$ over $[0, T]$, we can output $y(t)$ such that with probability at least $1 - 2^{-\Omega(k)}$,*

$$\|y - x^*\|_{T'} \leq (\sqrt{2} + \varepsilon + c)\|g\|_T + \delta\|x^*\|_T.$$

*Our algorithm uses $\mathrm{poly}(k, \varepsilon^{-1}, c^{-1}, \log(1/\delta)) \log(FT)$ samples and $\mathrm{poly}(k, \varepsilon^{-1}, c^{-1}, \log(1/\delta)) \cdot \log^2(FT)$ time. The output $y$ is $\mathrm{poly}(k, \varepsilon^{-1}, c^{-1}, \log(1/\delta))$-Fourier-sparse signal.*

*Proof.* Let $\mathcal{N}_1^2 := \varepsilon_1(\|g(t)\|_T^2 + \delta\|x^*(t)\|_T^2)$ be the heavy cluster parameter.

First, by Lemma H.12, there is a set of frequencies $S^* \subset [k]$ and $x_{S^*}(t) = \sum_{j \in S^*} v_j e^{2\pi \mathbf{i} f_j t}$ such that

$$\|x_{S^*} - x^*\|_T^2 \leq (1 + \varepsilon)\mathcal{N}_1^2. \tag{63}$$

Furthermore, each $f_j$ with $j \in S$ belongs to an $\mathcal{N}_1$-heavy cluster $C_j$ with respect to the filter function $H$ defined in Definition H.6.

By Definition H.11 of heavy cluster, it holds that

$$\int_{C_j} |\widehat{H \cdot x^*}(f)|^2 \mathrm{d}f \geq T\mathcal{N}_1^2/k.$$

By Definition H.11, we also have $|C_j| \leq k \cdot \Delta_h$, where $\Delta_h$ is the bandwidth of $\widehat{H}$.

Let $\Delta \in \mathbb{R}_+$, and $\Delta > k \cdot \Delta_h$, which implies that $C_j \subseteq [f_j - \Delta, f_j + \Delta]$. Thus, we have

$$\int_{f_j - \Delta}^{f_j + \Delta} |\widehat{H \cdot x^*}(f)|^2 \mathrm{d}f \geq T\mathcal{N}_1^2/k.$$

By Corollary H.22, there is a set of frequencies $S \subset S^*$ and $x_S(t) = \sum_{j \in S} v_j e^{2\pi \mathbf{i} f_j t}$ such that

$$\|x_S - x_{S^*}\|_T^2 \leq (1 + O(\sqrt{\varepsilon_2}))\|x - x_{S^*}\|_T^2.$$

Let $g' = x - x_{S^*}$.

Now it is enough to recover only $x_S$, instead of $x^*$.

By applying Theorem I.2, there is an algorithm that outputs a set of frequencies $L \subset \mathbb{R}$ such that, $|L| = O(k/(\varepsilon_0 \varepsilon_1 \varepsilon_2))$, and with probability at least $1 - 2^{-\Omega(k)}$, for any $f_j$ with $j \in S$, there is a $\widetilde{f} \in L$ such that,

$$|f_j - \widetilde{f}| \lesssim \Delta\sqrt{\Delta T}.$$

We define a map $p : \mathbb{R} \to L$ as follows:

$$p(f) := \arg\min_{\widetilde{f} \in L} |f - \widetilde{f}| \quad \forall f \in \mathbb{R}.$$

Then, $x_S(t)$ can be expressed as

$$
\begin{aligned}
x_S(t) &= \sum_{j \in S} v_j e^{2\pi \mathbf{i} f_j t} \\
&= \sum_{j \in S} v_j e^{2\pi \mathbf{i} \cdot p(f_j) t} \cdot e^{2\pi \mathbf{i} \cdot (f_j - p(f_j)) t} \\
&= \sum_{\widetilde{f} \in L} e^{2\pi \mathbf{i} \widetilde{f} t} \cdot \sum_{j \in S:\, p(f_j) = \widetilde{f}} v_j e^{2\pi \mathbf{i}(f_j - \widetilde{f}) t},
\end{aligned}
$$

where the first step follows from the definition of $x_S(t)$, the last step follows from interchanging the summations.

For each $\widetilde{f}_i \in L$, by Corollary H.2 with $x^* = x_S, \Delta = \Delta\sqrt{\Delta T}$, we have that there exist degree $d = O(T\Delta\sqrt{\Delta T} + k^3 \log k + k \log 1/\delta)$ polynomials $P_i(t)$ corresponding to $\widetilde{f}_i \in L$ such that,

$$\left\|x_S(t) - \sum_{\widetilde{f}_i \in L} e^{2\pi \mathbf{i} \widetilde{f}_i t} P_i(t)\right\|_T \leq \delta\|x_S(t)\|_T \tag{64}$$

Define the following function family:

$$\mathcal{F} := \text{span}\left\{ e^{2\pi \mathbf{i}\widetilde{f}t} \cdot t^j \mid \forall \widetilde{f} \in L, j \in \{0, 1, \ldots, d\}\right\}.$$

Note that $\sum_{\widetilde{f}_i \in L} e^{2\pi \mathbf{i}\widetilde{f}_i t} P_i(t) \in \mathcal{F}$.

By Claim H.16, for function family $\mathcal{F}$, $K_{\text{Uniform}[cT/2, T(1-c/2)]} = O((|L|d)^4 \log^3(|L|d))$.

By Lemma H.18, we have that, choosing a set $W$ of $O(\varepsilon^{-1} K_{\text{Uniform}[cT/2, T(1-c/2)]} \log(|L|d/\rho))$ i.i.d. samples uniformly at random over duration $[0, T]$ is a $(\varepsilon, \rho)$-WBSP.

By Lemma H.19, there is an algorithm that runs in $O(\varepsilon^{-1}|W|(|L|d)^{\omega-1}\log(1/\rho))$-time using samples in $W$, and outputs $y'(t) \in \mathcal{F}$ such that, with probability $1 - \rho$,

$$\|y'(t) - \sum_{\widetilde{f}_i \in L} e^{2\pi \mathbf{i}\widetilde{f}_i t} P_i(t)\|_{T'} \leq (1+\varepsilon)\|x(t) - \sum_{\widetilde{f}_i \in L} e^{2\pi \mathbf{i}\widetilde{f}_i t} P_i(t)\|_{T'} \tag{65}$$

Then by Lemma H.3, we have that there is a $O(kd)$-Fourier-sparse signal $y(t)$, such that

$$\|y(t) - y'(t)\|_{T'} \leq \delta' \tag{66}$$

where $\delta' > 0$ is any positive real number. Thus, $y$ can be arbitrarily close to $y'$.

Moreover, the sparsity of $y(t)$ is $kd = kO(T\Delta\sqrt{\Delta T} + k^3 \log k + k \log 1/\delta) = \text{poly}(k, \varepsilon^{-1}, c^{-1}, \log(1/\delta))$.

Therefore, the total approximation error can be upper bounded as follows:

$$\|y - x^*\|_{T'}$$
$$\leq \|y - y'\|_{T'} + \left\|y - \sum_{\widetilde{f}_i \in L} e^{2\pi \mathbf{i}\widetilde{f}_i t} P_i(t)\right\|_{T'} + \left\|\sum_{\widetilde{f}_i \in L} e^{2\pi \mathbf{i}\widetilde{f}_i t} P_i(t) - x^*\right\|_{T'}$$
$$\leq (1 + 0.1\varepsilon)\left\|y - \sum_{\widetilde{f}_i \in L} e^{2\pi \mathbf{i}\widetilde{f}_i t} P_i(t)\right\|_{T'} + \left\|\sum_{\widetilde{f}_i \in L} e^{2\pi \mathbf{i}\widetilde{f}_i t} P_i(t) - x^*\right\|_{T'}$$
$$\leq (1 + 2\varepsilon)\left\|x - \sum_{\widetilde{f}_i \in L} e^{2\pi \mathbf{i}\widetilde{f}_i t} P_i(t)\right\|_{T'} + \left\|\sum_{\widetilde{f}_i \in L} e^{2\pi \mathbf{i}\widetilde{f}_i t} P_i(t) - x^*\right\|_{T'}$$
$$\leq (1 + 2\varepsilon)(\left\|x - x_S\right\|_{T'} + \left\|x_S - x^*\right\|_{T'}) + 2(1+\varepsilon)\|x_S - \sum_{\widetilde{f}_i \in L} e^{2\pi \mathbf{i}\widetilde{f}_i t} P_i(t)\|_{T'}$$
$$\leq (1 + 2\varepsilon)(\left\|x - x_S\right\|_{T'} + \left\|x_S - x^*\right\|_{T'}) + O(\delta)\|x_S(t)\|_T$$
$$\leq (1 + 2\varepsilon)(\sqrt{2} + O(\sqrt{\varepsilon} + c))\|g\|_T + O(\sqrt{\delta})\|x^*\|_T + O(\delta)\|x_S(t)\|_T$$
$$\leq (1 + 2\varepsilon)(\sqrt{2} + O(\sqrt{\varepsilon} + c))\|g\|_T + O(\sqrt{\delta})\|x^*\|_T + O(\delta)(\|g\|_T + \|x^*\|_T)$$
$$\leq (\sqrt{2} + O(\sqrt{\varepsilon} + c))\|g\|_T + O(\sqrt{\delta})\|x^*\|_T,$$

where the first step follows from triangle inequality, the second step follows from Eq. (66), the third step follows from Eq. (65), the forth step follows from Triangle Inequality again, the fifth step follows from Eq. (64), the sixth step follows from Lemma I.1, the seventh step follows from Lemma H.39, and the last step is straightforward.

By re-scaling $\varepsilon$ and $\delta$, we prove the theorem.

$\square$

# J  Broader Impact

By cutting the approximation constant from $\sim 100$ to $3 + \sqrt{2}$, our methods could materially shorten scan times in MRI, reduce power consumption in compressive sensing devices, and improve fidelity in spectrum-sparse communication systems, thus benefiting healthcare, environmental monitoring,

and data transmission. At the same time, higher-quality reconstructions from fewer samples may amplify surveillance capabilities or aid in generating convincingly doctored audio/video; responsible adoption therefore demands privacy safeguards, transparent validation on non-ideal data, and ethical oversight whenever the technology is applied to sensitive domains.

