# OpenReview forum: "Efficient $k$-Sparse Band–Limited Interpolation with Improved Approximation Ratio"
_NeurIPS.cc/2025/Conference — NeurIPS 2025 poster_

### Official Review · Reviewer_jo7g · 2025-06-09

**Clarity:** 2
**Significance:** 4
**Originality:** 3
**Rating:** 5
**Confidence:** 2

**Summary:**

This paper discusses the estimation of $k$-sparse signals, defined as continuous signals over $x: [0,T] \rightarrow {\mathbb C}$ which are sums of $k$ frequencies, i.e., $x(t) = \sum_{j=1}^k v_j e^{2\pi i f_j t}$ for some set of frequencies $\\{f_j\\}_{j=1}^k$.

We are to estimate a $k$-sparse signal $x^\*(t)$ from noisy samples $\\{x^\*(t_j) + g(t_j)\\}_{j=1}^N$. The frequencies $f_j$ are unknown, but we are allowed to choose the sampling points $t_j$.

The authors propose an estimation algorithm and prove that the generated estimate $y(t)$ satisfies
$$ \\|y - x^\*\\|_T \\le C \\|g\\|_T + \\delta \\|x^\*\\|_T $$
where $\\|\\cdot\\|_T$ is the $L_2$ norm on $[0,T]$, and $\delta>0$ is a user-selected accuracy parameter determining the time complexity of the algorithm and the number of samples it requires. Such results have previously been shown; this papers' contribution is in a significant reduction of the constant $C$, from around 100 to below 5.

**Questions:**

Please address the comments above regarding clarity of the paper, making especially certain that terms are defined and that the problem context is addressed early in the text. I have no other requests from the authors.

**Ethical Concerns:**

["NO or VERY MINOR ethics concerns only"]

**Final Justification:**

no change in assessment

**Quality:**

4

**Strengths And Weaknesses:**

Caveat: I have background in classical sampling theory but with no familiarity with sparse Fourier signals or the theory of sampling such signals. My comments are therefore mostly about the paper's presentation and clarity for someone not within this specific field. The results in the paper look worthwhile, impressive, and valuable, but I have no way of judging their importance or novelty. Hence my review is submitted with relatively low confidence.

That said, the results in the paper appear strong and worthwhile, significantly reducing the bounds on sampling error guarantees. I believe the paper is worthy of publication.

The main weakness from my point of view is that the authors do not always properly define the setting and terminology before using it. Especially for a big-tent conference such as NeurIPS, such definitions are very important. In particular, I was able to formulate the summary above only after reading about four pages of the manuscript, during which I was constantly asking myself questions about the setting: are we talking about discrete or continuous signals? do we have a single measurement $x^\*(t) + g(t)$, or multiple noise realizations? and other questions, detailed below. Some of these questions were answered only in Section 1.2 (or even later), but this appears only on p.4, after the background and main results have already been stated. In particular:

1. There are many versions of Fourier transforms (discrete/continuous, finite-time/infinite-time, 1-dimensional/$d$-dimensional) and it is unclear for a long time which version you are talking about: in some cases an integral over $x(t)$ is performed (e.g. L.33) but in other cases it is stated that $x(t)$ is a discrete signal with $n$ values (L.141). In some cases $t$ is one-dimensional (L.97) and in other cases it is multi-dimensional (integral over $[0,T]^d$ in L.33).
1. "Band-limited", in my experience, means that $\hat{x}^*(f) = 0$ for all $|f| > f_c$, without any further sparsity requirements for $|f| \le f_c$. Though the sense meant here is given in Def. 1.1, I think it would be helpful to the reader if the Fourier sparsity assumption were stated in the title and abstract. Otherwise, a reader such as myself gets a decidedly different impression of the fundamental topic of the paper (and even the degree to which it is related to my field of study).
1. The form of measurements from which the estimate is generated is never explicitly specified. I would state explicitly (and early) that algorithmic approaches require observing samples of $y(t)$, with the number of samples required related to the desired accuracy. Again, this becomes clear implicitly but only in L.46, after the problem definition.
1. The band limit $F$ is mentioned in L.43 but only (implicitly) defined in L.62.
1. Theorem 1.3 is a simplified version of the actual theorem, given in the Appendix. While it's fine to put the proof in the Appendix, I think it is desirable to have a precise statement of the theorem in the main text. In particular, the fact that the claim holds only with high probability is stated in the appendix but missing from the main text.

I further encourage the authors to correct the following small notation errors:
1. L.32: should be $\\|x\\|_T^2$ and not $|x|_T^2$.
1. L.125: For consistency, it would be better to use $\\|g(t)\\|_S^2$ instead of $\\|g(t)\\|_W^2$ (especially since $w$ is used with a different meaning in the next line).
1. L.146: I think it would be clearer to state $\sup_t |x(t)|^2 \le k \\|x\\|_T^2$, and avoid the definition of $R$.
1. L.154: The symbol $\circ$ is not defined.
1. L.314: "simultaneously" should be "which simultaneously".
1. L.326-327: "the open" should be "an open question".

---

> ### Author Rebuttal · Authors · 2025-07-29
>
> We thank the reviewer for the thoughtful comments and the recommendation for acceptance. We appreciate your recognizing our results are strong and worthwhile. Below, we address the weakness and questions.
>
> ## W1: Version of Fourier transform
> Thank you for the question. Our main results focus on the one‑dimensional continuous sparse Fourier transform on a regular grid. When discussing techniques, however, we sometimes describe them in a dimension‑agnostic way, which may have caused confusion. In the revision we will clarify this distinction, tighten the wording, and ensure the notation remains consistent throughout.
> ## W2: The notion of ‘’band-limited’’
>
> Thank you for raising this point. In our paper, “band‑limited signal” specifically means a signal that is both band‑limited and $k$-sparse in the Fourier domain. We will clarify this in the revision by consistently writing ‘’$k$-sparse band‑limited signal.”
>
> ## W3 & W4: The form of measurements and the band limit $F$ are mentioned implicitly in the text.
>
> Thanks for pointing out this. We will define these notations more in a clearer way.
>
> ## W5: The informal version of theorems.
>
> For brevity and readability, we included only an informal version of main theorems in the main text. We agree, however, that more precise statements should appear there. In the revision, we will explicitly state that the guarantee holds with high probability in the main theorems.
>
> ## Typos
>
> Thanks for spotting the typos. We will fix them in the revised version.
>
> We sincerely thank you for your time and valuable comments. We hope that the above clarifications address your concerns, and we are happy to discuss further if needed.

---

> > ### Comment · Reviewer_jo7g · 2025-08-04
> >
> > Thank you kindly for your responses. I have no further questions for the authors.

---

> > > ### Author Response · Authors · 2025-08-04
> > >
> > > Thank you for taking the time to read our responses and for your positive evaluation. We truly appreciate it.

---

### Official Review · Reviewer_yu3y · 2025-06-25

**Clarity:** 3
**Significance:** 4
**Originality:** 3
**Rating:** 5
**Confidence:** 2

**Summary:**

This paper addresses the problem of interpolating a band-limited signal from a small number of noisy samples. It presents a new algorithm with a significantly improved approximation ratio compared to previous methods, breaking the long-standing barrier of a constant factor greater than 100. This paper propose ultra-high sensitivity frequency estimation and hierarchical noise cancellation analysis, and combine these two techniques to propose an algorithm that achieves continuous interpolation (3+√2+ε)-approximation guarantees, which is a significant improvement over the previous best result. A variant of the above algorithm is also proposed, which can approximate the signal by (√2+ε+c) on any subinterval. Overall, this paper provides a significant advancement in the field of band-limited interpolation, offering a new framework and techniques that can be applied to other sparse recovery problems.

**Questions:**

1. Can the noise tracking and filter family structure proposed in extend to higher dimensions? What are the obstacles?
2. The paper mentioned that the running time of the algorithm is close to linear. Can a more detailed computational complexity analysis be provided, including the time complexity of different steps, and the overall complexity as a function of signal length, sparsity, and frequency range?
3. What are the effects of different parameters such as frequency gap, SNR, etc. on the performance of the algorithm?

**Ethical Concerns:**

["NO or VERY MINOR ethics concerns only"]

**Final Justification:**

Based on the content of the article and the author's response, I have a clearer understanding of this job. The author gave reasonable and detailed responses on Empirical validation, dimension expansion, running time analysis and the influence of some parameters. However, some detailed proof contents require the author to provide appendix contents for further reading. Although this work is purely theoretical research, through rigorous and reasonable theoretical analysis, it solves the problem of multiplicative noise elimination >100 and reduces the approximation ratio to about 5, which shows its importance in this task. Therefore, the scores in terms of Quality and Significance have been modified from 3 to 4, and the Rating has been changed from "Borderline accept" to "accept".

**Limitations:**

yes

**Paper Formatting Concerns:**

Correct the format of the paper.

**Quality:**

4

**Strengths And Weaknesses:**

Strengths:
1. The paper delivers the first polynomial-time algorithm achieving an approximation ratio strictly below 5 for noisy band-limited interpolation—breaking the longstanding constant-factor barrier of ≈ 100 in prior work—and further refines this to (3 + √2 + ε) and even (√2 + ε + c) under a “shrinking-range” regime.
2. Definitions for band-limited interpolation and discrete Fourier set-query are crisply stated, grounding the reader before diving into algorithmic details.
3. Novel two-stage composition. Showing that a two-stage well-balanced sampler composes into a single uniform distribution is both subtle and powerful, enabling the sharper (1 + ε) set-query guarantees.
Weaknesses:
1. The article has some appendix dependencies. Key formal proofs (e.g., Theorems H.20 and I.4) are deferred to later sections, which may interrupt the flow for readers seeking immediate understanding of the guarantees
2. The high-level pipeline (filtering→frequency estimation→polynomial approximation→regression) parallels prior works (e.g., Chen et al. 2016); the novelty lies in tightening constants rather than introducing wholly new paradigms.
3. While theoretical constants shrink dramatically, there are no experiments demonstrating performance on synthetic or real data to confirm that the improved ratios translate to practice.

---

> ### Author Rebuttal · Authors · 2025-07-29
>
> We thank the reviewer for the thoughtful comments and the recommendation for acceptance.
> We appreciate that you highlight the first polynomial‑time algorithm with an improved approximation ratio, the clear definitions and presentation, and the subtle two‑stage composition that enables the sharper bound on each stage. Below, we address the weakness and questions.
>
> ## W1: Proofs are deferred to Appendix.
> To keep the main text streamlined, we placed the detailed proof in the appendix while keeping every theorem’s key steps and proof sketch in the technique overview. We will try to improve the presentation and readability of the algorithms in the revised version. Thanks for pointing this out.
>
> ## W2: Novelty
>
> Thanks for the question. We would like to clarify the novelty and contribution of our paper. Although the high‑level pipeline resembles prior work, our multiplicative noise‑cancellation analysis breaks the long‑standing $>100$ constant barrier, cutting the worst‑case approximation ratio to $\approx 5$ within a small factor of the information‑theoretic optimum. Achieving this required new, tighter energy bounds and a refined sketch‑distillation step that eliminates the additive‑error accumulation present in all earlier methods. This, in turn, demands a more delicate control of error propagation across stages, which we believe is nontrivial.
>
>
> ## W3: Empirical validation
> Thank you for your question regarding the real-world empirical validation of our algorithms. While empirical validation would be interesting, demonstrating these guarantees experimentally would require a substantial engineering effort and domain‑specific datasets that lie outside the paper’s scope since our primary contribution is intentionally theoretical. We focused on rigorous analysis and proofs. We would like to clarify that our work is purely a theoretical study, which does have a large place in the top machine learning conferences, like NeurIPS, ICLR, and ICML. For instance, NeurIPS accepts papers [1,2,3,4], all of which are entirely theoretical and do not contain any experiments. Moreover, ICLR and ICML also accepted papers [5,6], which are purely theoretical. Our work is similar to theoretical studies like [6], concentrating on the theoretical aspects of Fourier transform.
>
>
> ## Q1: Extension to high dimension
>
> Our present algorithm is formulated for one‑dimensional signals. The main challenge for extending it to high dimension setting is the curse of dimensionality since naively treating a $d$-dimensional signal as a flattened $1$-dimensional signal would inflate both sample and time complexity exponentially in $d$. A more scalable route is to exploit structure, e.g., separable band‑pass filters and tensor‑product sketching, so that each dimension is processed independently before a joint sparse regression stage. This would raise complexity only by a factor polynomial in $d$. Whether the same order of constant factor accuracy can be preserved requires a more detailed analysis. Formalising this extension is an interesting direction for future work.
>
> ## Q2: Detailed run time analysis
>
> Thanks for the question. Since our focus is on developing the high-accuracy algorithm, we didn’t include the detailed runtime analysis of some less important parameters. In fact, the major runtime cost of our high‑accuracy algorithm come from sketch distillation and solving weighted regression. With carefulled analysis, in sketch distillation first, our algorithm repeats the constant‑factor loop just $\Theta(1/\epsilon)$ times so every candidate signal’s energy is preserved within a $(1+\epsilon)$ factor, incurring only $\Theta(\epsilon^{-1} k)$ arithmetic operations and samples. The weighted regression involves a $k \times k$ weighted least‑squares system, which costs $k^\omega$ time using fast matrix multiplication where $\omega$ is the exponent of matrix multiplication and $\omega \approx 2.37$. Thus the $\mathrm{poly}(\epsilon^{-1},k)$ can be roughly expressed as $O(\epsilon^{-1}k^\omega)$.
>
>
> ## Q3: Effects of parameters
> Our guarantees depend on the noise level only through the composite term $\mathcal{N}^2 = \\|g\\|_T^2+\delta \\|x^*\\|_T^2$. Hence SNR enters linearly: doubling the noise variance roughly doubles every error term. With respect to frequency gap, the theory requires a minimum separation of $\eta \geq \Omega(1/T)$ to avoid aliasing. Hence the $\log(FT)$ factor in runtime can be replaced with $\log(F/\eta)$.
>
> We sincerely thank you for your time and valuable comments. We hope that the above clarifications address your concerns, and we are happy to discuss further if needed.
>
> [1] Alman, J., & Song, Z. Fast attention requires bounded entries. NeurIPS’23.
>
> [2] Sarlos, T., Song, X., Woodruff, D., & Zhang, R. Hardness of low rank approximation of entrywise transformed matrix products. NeurIPS’23.
>
> [3] Dexter, G., Drineas, P., Woodruff, D., & Yasuda, T. Sketching algorithms for sparse dictionary learning: PTAS and turnstile streaming. NeurIPS’23.
>
> [4] Song, Z., Vakilian, A., Woodruff, D., & Zhou, S. On Socially Fair Low-Rank Approximation and Column Subset Selection. NeurIPS’24
>
> [5] Kim, J., & Suzuki, T. Transformers Provably Solve Parity Efficiently with Chain of Thought. In The Thirteenth International Conference on Learning Representations. ICLR’25
>
> [6] Li, X., Song, Z., & Xie, S. Deterministic sparse fourier transform for continuous signals with frequency gap. ICML’25.

---

> ### Comment · Reviewer_yu3y · 2025-08-08
>
> Thank you very much for your detailed reply. I have no other questions about this paper.
> Based on your reply, I have raised the scores for "Quality" and "Importance", and changed the Rating to "Accept".

---

### Official Review · Reviewer_kZRg · 2025-07-05

**Clarity:** 3
**Significance:** 2
**Originality:** 3
**Rating:** 4
**Confidence:** 4

**Summary:**

This paper addresses the problem of interpolating band-limited signals with sparse frequency components from noisy time-domain samples. The authors propose a novel algorithmic framework based on hierarchical frequency decomposition and systematic noise cancellation, to obtain a certain improvements in the approximation constant. Furthermore, the paper introduces a refined variant that achieves an even tighter approximation constant of $ C = 2 + \omega + c $ on any sub-interval of the time domain. These results represent a substantial improvement over existing work, both in terms of theoretical guarantees and practical relevance.

The key contributions include:
1. A high-accuracy algorithm with a provable approximation constant of $ C = 3 + \sqrt{2} + \omega $, which is much closer to the information-theoretic optimum than exiting work.

2. Some technical innovations which include an ultra-sensitive frequency estimation method and a refined noise-cancellation analysis.

**Questions:**

How to extend the theoretical framework to higher-dimensional signals, which would be very interesting.

**Ethical Concerns:**

["NO or VERY MINOR ethics concerns only"]

**Final Justification:**

I would like to keep the score.

**Limitations:**

One may relax the assumptions of exact sparsity and constant SNR. The algorithm may be analysized under approximate sparsity or varying SNR conditions.

**Quality:**

3

**Strengths And Weaknesses:**

Strengths:

The paper makes several contributions:
1. The authors provide a rigorous theoretical framework for band-limited signal interpolation, achieving an improvement in the approximation constant.

2. The hierarchical noise cancellation method effectively reduces error accumulation by combining multiple sources of error into a single multiplicative term.

3. The proposed shrinking-range variant demonstrates how spatial slack can be utilized to further improve interpolation accuracy.

4. The results are well-supported by detailed theoretical analyses.


Weaknesses.

1. The author use Approximation Ratio. But it seems that the improvement is only for the approximation costant.

2. The paper assumes that the signal is exactly $ k $-sparse in the frequency domain and that the signal-to-noise ratio (SNR) exceeds a constant threshold. These assumptions may not hold in many practical scenarios, where signals often exhibit approximate sparsity or have varying SNR. Relaxing these assumptions would enhance the practical applicability of the results.

3. The proposed algorithm relies on ultra-sensitive frequency estimation and finely tuned filters, which require high numerical precision. In real-world deployments, numerical round-off errors or model mismatches could degrade the performance and erode the stated approximation constants.

4. The analysis and results are restricted to one-dimensional signals. Extending the framework to higher-dimensional settings, such as images or multi-dimensional time-series data, remains an open challenge.

5. The paper focuses on theoretical results and does not provide any numerical experiments.

6. While the authors claim that the algorithm has nearly linear runtime complexity, the dependence on parameters may lead to high computational costs for the large-scale problems. A more detailed analysis of the algorithm's scalability would be useful.

---

> ### Author Rebuttal · Authors · 2025-07-29
>
> Thank you for the constructive review and recommendation for acceptance. We appreciate your recognition of our rigorous framework, technical contributions, and theoretical analysis. Below, we address the noted weaknesses and questions.
>
> ## W1: The notion of approximation ratio
>
> Thanks for pointing this out. The approximation ratio is the standard terminology in approximation algorithms, which refers to the factor by which an algorithm's solution deviates from the optimal solution for a given computational problem.  In our setting, it quantifies how much larger the reconstruction error $\\|y-x^*\\|_T$​ can be, relative to the best achievable error. We will define it clearer in our revised version.
>
> ## W2: Assumption on $k$-sparsity in the frequency domain.
>
> This is an interesting question. In fact, as quoted from [7]: “Many applications rely on the fact that most of the Fourier coefficients of the signals are small or equal to zero, i.e., the signals are (approximately) sparse. Thus it is very natural to study the setting where the Fourier spectrum is approximately $k$-sparse, and it has been studied over the last two decades, see [6,7,8,9,10].
>
> ## W3: Numerical issue.
>
> Thank you for flagging the potential numerical challenges. Our present work concentrates on the algorithm design and theoretical guarantees. We defer the development of a robust, industrial‑grade implementation to future work.
>
> ## W4 & Q1: Extension to high dimension
>
> Yes, our present algorithm is formulated for one‑dimensional signals. The main challenge for extending it to high dimension setting is the curse of dimensionality since naively treating a $d$-dimensional signal as a flattened $1$-dimensional signal would inflate both sample and time complexity exponentially in $d$. A more scalable route is to exploit structure, e.g., separable band‑pass filters and tensor‑product sketching, so that each dimension is processed independently before a joint sparse regression stage. This would raise complexity only by a factor polynomial in $d$. Whether the same order of constant factor accuracy can be preserved requires a more detailed analysis. Formalising this extension is an interesting direction for future work.
>
> ## W5: Empirical validation
> Thank you for your question regarding the real-world empirical validation of our algorithms. While empirical validation would be interesting, demonstrating these guarantees experimentally would require a substantial engineering effort and domain‑specific datasets that lie outside the paper’s scope since our primary contribution is intentionally theoretical. We focused on rigorous analysis and proofs. We would like to clarify that our work is purely a theoretical study, which does have a large place in the top machine learning conferences, like NeurIPS, ICLR, and ICML. For instance, NeurIPS accepts papers [1,2,3,4], all of which are entirely theoretical and do not contain any experiments. Moreover, ICLR and ICML also accepted papers [5,6], which are purely theoretical. Our work is similar to theoretical studies like [6], concentrating on the theoretical aspects of Fourier transform.
>
> ## W6: Detail analysis of scalability
>
> Since our focus is on developing the high-accuracy algorithm, we didn’t include the detailed runtime analysis of some less important parameters. In fact, the major runtime cost of our high‑accuracy algorithm come from sketch distillation and solving weighted regression. With carefulled analysis, in sketch distillation first, our algorithm repeats the constant‑factor loop just $\Theta(1/\epsilon)$ times so every candidate signal’s energy is preserved within a $(1+\epsilon)$ factor, incurring only $\Theta(\epsilon^{-1} k)$ arithmetic operations and samples. The weighted regression involves a $k \times k$ weighted least‑squares system, which costs $k^\omega$ time using fast matrix multiplication where $\omega$ is the exponent of matrix multiplication and $\omega \approx 2.37$. Thus the $\mathrm{poly}(\epsilon^{-1},k)$ can be roughly expressed as $O(\epsilon^{-1}k^\omega)$. Evaluating the algorithm’s scalability in real‑world deployments is an engaging avenue for future work. Thank you for raising this point.
>
> We sincerely thank you for your time and valuable comments. We hope that the above clarifications address your concerns, and we are happy to discuss further if needed.
>
> ## Reference
> [1] Alman, J., & Song, Z. Fast attention requires bounded entries. NeurIPS’23.
>
> [2] Sarlos, T., Song, X., Woodruff, D., & Zhang, R. Hardness of low rank approximation of entrywise transformed matrix products. NeurIPS’23.
>
> [3] Dexter, G., Drineas, P., Woodruff, D., & Yasuda, T. Sketching algorithms for sparse dictionary learning: PTAS and turnstile streaming. NeurIPS’23.
>
> [4] Song, Z., Vakilian, A., Woodruff, D., & Zhou, S. On Socially Fair Low-Rank Approximation and Column Subset Selection. NeurIPS’24
>
> [5] Kim, J., & Suzuki, T. Transformers Provably Solve Parity Efficiently with Chain of Thought. In The Thirteenth International Conference on Learning Representations. ICLR’25
>
> [6] Li, X., Song, Z., & Xie, S. Deterministic sparse fourier transform for continuous signals with frequency gap. ICML’25.
>
> [7] Indyk, P., & Kapralov, M. Sample-optimal Fourier sampling in any constant dimension. FOCS’14
>
> [8] Price, E., & Song, Z. A robust sparse Fourier transform in the continuous setting. FOCS’15
>
> [9] Hassanieh, H., Indyk, P., Katabi, D., & Price, E. Nearly optimal sparse Fourier transform. STOC’12.
>
> [10] Hassanieh, H., Indyk, P., Katabi, D., & Price, E. Simple and practical algorithm for sparse Fourier transform. SODA’12

---

> > ### Comment · Reviewer_kZRg · 2025-08-08
> >
> > Thank you for your detailed responses I would like to keep the score.

---

### Official Review · Reviewer_aWR8 · 2025-07-06

**Clarity:** 2
**Significance:** 4
**Originality:** 3
**Rating:** 4
**Confidence:** 5

**Summary:**

The paper proposes new guarantees of "interpolation" of a k-sparse band-limited signal using very few noisy time-domain samples. The 'interpolation' referred to here is signal reconstruction. While prior works exhibit a high approximation ratio of >100, the paper claims to bring it down to 5. This work has connections with a huge literature on sFFT and its own results rely on discrete Fourier set theory. A large number of theoretical results have also been derived. While this reviewer has not been able to verify all theoretical derivations in the Appendix, the primary content in the main file appears to be technically correct.

**Questions:**

It is not clear if a minimum frequency separation is required here or only k-sparsity suffices.

What is the usual order of the four sources of error? Is anyone more dominant the others?

There are many signals which are multi-band. Would it be a simple extension of this work? Multi-band spectral recovery is a well-investigated topic using atomic norm minimization. Some prior works also employ mild prior knowledge in such cases. See the (separate) works of Weiyu Xu and Gongguo Tang on this.

**Ethical Concerns:**

["NO or VERY MINOR ethics concerns only"]

**Final Justification:**

Based on the responses provided by the authors, I have increased the score on significance and quality. However, validation concerns because of lack of numerical experiments remain. Sampling theory results requires such validation, in the opinion of this reviewer.

**Limitations:**

Lack of numerical experiments makes it difficult to validate the claims.

Probabilistic guarantees, while common in sFFT, are not very practical.

Minimum frequency separation is another bottleneck. It is not very clear how the paper avoids it.

The conditions on recovery are not mild. Again, using some typical values of realistic SNRs and sparsity, the paper could have shown the practicality of the proposed method via numerical experiments,

**Paper Formatting Concerns:**

No major formatting concerns observed by the reviewer.

**Quality:**

4

**Strengths And Weaknesses:**

Reducing the approximation ratio from the existing 100 to 5 is remarkable, although this is accomplished under not-s-mild assumptions on SNR, sparsity, on-grid frequencies, and probabilistic guarantees. In this context, the result may also be interpreted as weak because sFFT, while theoretically elegant, is not very practical because of probabilistic guarantees.

The paper is very detailed and the writing is lucid. I enjoyed going over the main paper. As for the very long appendix, it requires more time to review than the provided NeurIPS review time.

There are no numerical experiments to validate the claims of signal reconstruction. This is reflected in my low rating on Clarity.

The paper also does not very well connect itself with ML, the focus of the conference.

---

> ### Author Rebuttal · Authors · 2025-07-29
>
> Thank you for the insightful review and recommending for acceptance. We are glad that you find our results are remarkable and the paper is well-written. Below, we address your concerns and questions.
>
> ## W1: Lack of empirical validation.
> Thank you for your question regarding the real-world empirical validation of our algorithms. While empirical validation would be interesting, demonstrating these guarantees experimentally would require a substantial engineering effort and domain‑specific datasets that lie outside the paper’s scope since our primary contribution is intentionally theoretical. We focused on rigorous analysis and proofs. We would like to clarify that our work is purely a theoretical study, which does have a large place in the top machine learning conferences, like NeurIPS, ICLR, and ICML. For instance, NeurIPS accepts papers [1,2,3,4], all of which are entirely theoretical and do not contain any experiments. Moreover, ICLR and ICML also accepted papers [5,6], which are purely theoretical. Our work is similar to theoretical studies like [6], concentrating on the theoretical aspects of Fourier transform.
>
>
>
>
> ## W2: Connection with ML conferences.
>
> In recent years, Fourier transform has also emerged as a powerful tool within machine learning research, inspiring diverse models and algorithms [7,8,9,10,11,12]. Due to its significance in both theory and practice, finding an efficient algorithm to compute the Fourier transform is of utmost importance, especially for machine learning. The most remarkable one is the random feature method [13], which has extensive applications in neural networks [14,15,16].
>
> ## Q1: It is not clear if a minimum frequency separation is required here or only k-sparsity suffices. What is the usual order of the four sources of error? Is anyone more dominant the others?
>
> Thank you for pointing this out. Yes, consistent with the standard continuous‑SFT literature, our analysis assumes an $\Omega(1/T)$ minimum separation between any two true frequencies. We might miss this in the condition and will fix it in the revised version.
>
> ## Q2: What is the usual order of the four sources of error? Is anyone more dominant the others?
>
> In prior works, the heavy‑cluster truncation error $E_1$ is always the dominant contributor because it scales as $\mathcal{N}^4$​. The sampling‑and‑regression error $E_3$ is the next‑largest, growing only linearly with $\mathcal{N}$, so it can approach, but rarely surpass, $E_1$ when the SNR is low. Polynomial approximation $E_2$ is proportional to the chosen tolerance $\delta$ and stays much smaller. In our proposed scheme,  all structural errors scale only linearly in $\mathcal{N}$, so their sizes are determined by the leading constants.  The weighted‑regression term is largest, $E_{3}^{\text{new}} \leq (4+\epsilon)\mathcal{N}$ . Next comes the energy of frequencies that fail the reinforced cluster condition, $E_{1.5} \leq (2+\epsilon)\mathcal{N}$, followed by the initial heavy‑cluster truncation, $E_{1}^{\text{new}} \leq (1+\epsilon)\mathcal{N}$.  The polynomial‑approximation error remains proportional to the chosen tolerance $\delta$ and stays much smaller. Hence the typical hierarchy is $E_{3}^{\text{new}} \geq E_{1.5} \geq E_{1}^{\text{new}} \gg E_{2}$.
>
>
>
> ## Q3: There are many signals which are multi-band. Would it be a simple extension of this work? Multi-band spectral recovery is a well-investigated topic using atomic norm minimization. Some prior works also employ mild prior knowledge in such cases. See the (separate) works of Weiyu Xu and Gongguo Tang on this.
>
> Thank you for pointing out the works on multi-band spectral recovery. We will cite these works in the revised version and discuss the potential extension in the revised version. We believe a detailed, formal treatment of this extension, along with a comparison to atomic‑norm guarantees, is an interesting direction for future work.
>
> We sincerely thank you for your time and valuable comments. We hope that the above clarifications address your concerns, and we are happy to discuss further if needed.
>
> ## Reference
> [1] Alman, J., & Song, Z. Fast attention requires bounded entries. NeurIPS’23.
>
> [2] Sarlos, T., Song, X., Woodruff, D., & Zhang, R. Hardness of low rank approximation of entrywise transformed matrix products. NeurIPS’23.
>
> [3] Dexter, G., Drineas, P., Woodruff, D., & Yasuda, T. Sketching algorithms for sparse dictionary learning: PTAS and turnstile streaming. NeurIPS’23.
>
> [4] Song, Z., Vakilian, A., Woodruff, D., & Zhou, S. On Socially Fair Low-Rank Approximation and Column Subset Selection. NeurIPS’24
>
> [5] Kim, J., & Suzuki, T. Transformers Provably Solve Parity Efficiently with Chain of Thought. In The Thirteenth International Conference on Learning Representations. ICLR’25
>
> [6] Li, X., Song, Z., & Xie, S. Deterministic sparse fourier transform for continuous signals with frequency gap. ICML’25.
>
> [7] Lee, J. D., Shen, R., Song, Z., Wang, M., and Yu, z. Generalized leverage score sampling for neural networks. NeurIPS’20
>
> [8] Choromanski, K. M., Likhosherstov, V., Dohan, D., Song, X., Gane, A., Sarlos, T., Hawkins, P., Davis, J. Q., Mohiuddin, A., Kaiser, L., Belanger, D. B., Colwell, L. J., and Weller, A. Rethinking attention with performers. InInternational Conference on Learning Representations
> . ICLR’ 2021
>
> [9] Li, Z., Kovachki, N. B., Azizzadenesheli, K., Liu, B., Bhattacharya, K., Stuart, A., and Anandkumar, A. Fourier neural operator for parametric partial differential equations. ICLR’ 21
>
> [10] Yang R, Cao L, YANG J. Rethinking fourier transform from a basis functions perspective for long-term time series forecasting. NeurIPS’24
>
> [11] Alman, J. and Song, Z. Fast rope attention: Combining the polynomial method and fast Fourier transform. arXiv preprint arXiv:2505.11892, 2025
>
> [12] Yu, E., Lu, J., Yang, X., Zhang, G., and Fang, Z. Learning robust spectral dynamics for temporal domain generalization. arXiv preprint arxiv: 2505.12585, 2025
>
> [13] Rahimi, A., & Recht, B. Random features for large-scale kernel machines. NeurIPS'07
>
> [14] Yehudai, G., & Shamir, O. On the power and limitations of random features for understanding neural networks. NeurIPS'19
>
> [15] Frei, S., Chatterji, N. S., & Bartlett, P. L. Random feature amplification: Feature learning and generalization in neural networks. JMLR, 2023.
>
> [16] Sato, R., Yamada, M., & Kashima, H. Random features strengthen graph neural networks. ICDM'21

---

> > ### Comment · Reviewer_aWR8 · 2025-08-04
> > **Validation concerns remain**
> >
> > Thank you for your detailed responses. I appreciate it very much.
> >
> > Based on your responses, I have increased the score on significance and quality.
> >
> > However, validation concerns remain. In my opinion, sampling theory results such as in your work require validation for various types of signals.

---

> > > ### Author Response · Authors · 2025-08-04
> > >
> > > We appreciate you taking the time to read our response and for your positive evaluation. Thank you.

---

### Comment · Area_Chair_oeDH · 2025-08-04

Dear Reviewers,
Thanks for your efforts. Since authors have replied to the reviews, please check whether the rebuttal solves your concerns and respond to authors.

Best regards,
AC

---

### Note · Authors · 2025-08-14

We sincerely thank the AC and the reviewers for the thoughtful feedback and constructive suggestions. We are encouraged by your positive assessments of our contributions and clarity. Several points were highlighted as strengths:

- First high-accuracy, polynomial-time algorithm for band–limited interpolation with provable approximation, breaking the long-standing barrier and sharply narrowing the gap to the information-theoretic optimum.

- Ultra-sensitive frequency estimation and hierarchical noise-cancellation analysis that collapse additive losses into a single multiplicative term.

- Rigorous and readable analysis with a clear pipeline, and near-linear sample/time dependence up to polylog factors.

We will address the raised concerns and further improve accessibility to a broader ML audience. Specifically, we will:

- State assumptions up front and precisely. We will clearly state the standard minimum frequency separation, SNR condition, and $k$-sparsity at the start of the main theorem.

- Strengthen the main-text intuition. We will add a reader-friendly sketch for (i) ultra-sensitive filtering, (ii) signal–noise cancellation that couples filtered noise with unrecoverable energy, and (iii) why the losses combine multiplicatively rather than additively.

- We will provide a boxed summary of sample and runtime dependencies, and isolate the leading costs  with a short note on numerical stability.

-  We will add a subsection discussing how the framework extends qualitatively to approximately $k$-sparse spectra and non-constant SNR, flagging this as a promising direction for formal guarantees.

- We will expand related work and discuss the works pointed by reviewers in the more details.

- We will add a brief paragraph connecting our results to ML uses of Fourier structure (e.g., random features, spectral methods, Fourier neural operators) and provide a concise pseudo-code box for the end-to-end pipeline to aid reproducibility.

- Polish and correctness. We will tighten notation, fix typos, and move frequently used definitions earlier for smoother reading.

We are grateful for your time and constructive feedback. We appreciate the recognition of our work’s potential to push sparse recovery beyond long-standing constant-factor barriers and to inspire follow-up research at the ML and signal processing. Thank you!

---

### Decision · Program_Chairs · 2025-09-17

**Decision:**

Accept (poster)

**Comment:**

This paper addresses the problem of interpolating band-limited signals with sparse frequency components from noisy time-domain samples.
All reviewers basically appreciate that the authors provide a rigorous theoretical framework for band-limited signal interpolation, achieving an improvement in the approximation constant.
During the rebuttal period, the concerns from the reviewers are mostly resolved.

There is one major concern remains:
Reviewer aWR8 & Reviewer yu3y & Reviewer kZRg  : Lack of empirical validation or numerical results
Even for a theoretical work, empirical validation is also important. The AC strongly suggests that the authors need to provide some empirical results to validate the proposed method since for the references the authors cited (e.g.,  the gounp (Hassanieh, H., Indyk, P., Katabi, D., & Price), the authors have also employed their algorithms for practical use.
So, the authors are suggested to validate their theoretical methods.

Besides, the concerns on the use of notations & terminologies, raised by Reviewer jo7g, must be clearly clarified in the revised version.

Overall, based on the reviewers' responses, this work will be suggested to be accepted.